# Structures of proteinase 3 and the CD177 receptor complex reveal a major autoantibody epitope

Céline Zheng-Gérard [1,2,10], Jana Joha [1,2,10], Maria Carrasquero[1,2], Kamel El Omari [3], Edward Lowe[1,2], Shirish Dubey [4,5], Simon J Draper [2,6], Yu-Chi Chang[7], Hsi-Hsien Lin [7,8✉], Alan D Salama [9✉], Kirsty McHugh [1,6✉] & Elena Seiradake [1,2✉]

## Abstract

**Granulomatosis with polyangiitis is a life-threatening systemic vasculitis, characterised by anti-neutrophil cytoplasmic auto-antibodies (ANCA) most commonly against proteinase 3 (PR3), a protease expressed intracellularly and on the surface of neutrophils. Most cell surface PR3 is bound to the receptor CD177; however, the molecular mechanism of the interactions is not well understood. Here, we present crystal structures of CD177 in complex with PR3 and unliganded CD177. We describe a mainly hydrophobic binding interface between PR3 and CD177, involving the first two Ly6/uPAR (LU) domains of CD177. These form a globular structure which is connected to downstream domains via a flexible linker. Using a panel of PR3-ANCA-positive patient samples, we show that a significant proportion of ANCAs target the CD177-binding site of PR3 in these samples. Structure-guided mutation of the CD177-binding site on PR3 is effective in reducing PR3-ANCA binding. The results demonstrate that the CD177-binding surface of PR3 harbours a major PR3-ANCA epitope, and that the extent of binding to this surface varies between different patients.**

**Keywords** PR3; CD177; ANCA; Granulomatosis With Polyangiitis (GPA); ANCA-associated Vasculitis
**Subject Categories** Immunology; Molecular Biology of Disease; Structural Biology

## Introduction

Granulomatosis with polyangiitis (GPA) is a devastating form of ANCA-associated vasculitis (AAV) which causes necrosis of small and medium blood vessels and affects ~2-150 per million individuals (Kitching et al, 2020; Robson et al, 2022). GPA is commonly associated with the presence of anti-neutrophil cytoplasmic antibodies (ANCA) that predominantly target proteinase 3 (PR3-ANCA). The varying levels of extracellular PR3 found in different individuals correlate with a risk of developing autoantibodies against PR3, with higher levels presenting a higher risk for developing PR3-ANCA and GPA (Witko-Sarsat et al, 1999; Rarok et al, 2002; Yang et al, 2004). Occasionally, PR3-ANCA are also detected in other conditions, for example, during inflammation, and in drug-induced vasculitis (Csernok et al, 2010; McAdoo et al, 2011). PR3 is a neutrophil serine protease that is mostly found in intracellular azurophilic granules of resting neutrophils. Upon neutrophil activation, mature PR3 is released into the extracellular space, where it plays key roles in cleaving protein substrates and mediating inflammatory signalling cascades (Robache-Gallea et al, 1995; Coeshott et al, 1999; Csernok et al, 2006; Chu et al, 2022). In neutrophils, PR3 is first synthesised as a pre-pro-enzyme with a signal sequence of 25 residues that is cleaved, resulting in an inactive pro-enzyme. Subsequent removal of an N-terminal pro-dipeptide is catalysed by the cysteine protease dipeptidyl peptidase I (DPPI or cathepsin C) before or during transport to neutrophil granules (Adkison et al, 2002), resulting in active mature PR3. As found for other neutrophil serine proteases, PR3 also undergoes C-terminal processing (Salvesen and Enghild, 1990; Rao et al, 1996), which may play a role in proper trafficking of the enzymes to granule compartments (Benson et al, 2003; Horwitz et al, 2004). Structural analysis of recombinant mature human PR3 from insect cell cultures revealed a chymotrypsin-like fold, each consisting of two 6-stranded β-barrel domains, and a central active site catalytic triad (H71-D118-S203) (Fujinaga et al, 1996). More recently, mammalian HEK293 cultures were used to produce functional PR3 for biophysical analysis (Jerke et al, 2017; Marino et al, 2022).

In contrast to many other neutrophil serine proteases, low levels of PR3 are detected on the surface of some populations of unstimulated neutrophils (Csernok et al, 1994; Halbwachs-Mecarelli et al, 1995; Campbell et al, 2000), resulting in constitutively cell-surface-expressed, membrane-associated PR3 (mPR3). In CD177-positive individuals,

[1]Department of Biochemistry, University of Oxford, South Parks Road, Oxford OX1 3QU, UK. [2]Kavli Institute for Nanoscience Discovery, Dorothy Crowfoot Hodgkin Building, University of Oxford, South Parks Road, Oxford OX1 3QU, UK. [3]Diamond Light Source Limited, Harwell Science and Innovation Campus, Didcot, UK. [4]Department of Rheumatology, Oxford University Hospitals NHS FT, Oxford OX3 7LD, UK. [5]Nuffield Department of Orthopaedics, Rheumatology and Musculoskeletal Sciences, University of Oxford, Oxford OX3 7HE, UK. [6]Department of Paediatrics, University of Oxford, South Parks Road, Oxford OX1 3QU, UK. [7]Department of Microbiology and Immunology, College of Medicine, Chang Gung University, Taoyuan 333, Taiwan. [8]Department of Anatomic Pathology, Chang Gung Memorial Hospital at Linkou, Taoyuan 333, Taiwan. [9]UCL Centre for Kidney and Bladder Health, Royal Free Hospital, London, UK. [10]These authors contributed equally: Céline Zheng-Gérard, Jana Joha. ✉E-mail: hhlin@mail.cgu.edu.tw; a.salama@ucl.ac.uk; kirsty.mchugh@paediatrics.ox.ac.uk; elena.seiradake@bioch.ox.ac.uk

the majority of extracellular PR3 is presented on the neutrophil cell surface in complex with its receptor CD177 (also named HNA-2a or NB1) (Bauer et al, 2007; Von Vietinghoff et al, 2007; Abdgawad et al, 2010), although PR3 can also bind directly to cell membrane components (Hu et al, 2009; Goldmann et al, 1999; Hajjar et al, 2008; Broemstrup and Reuter, 2010; Kantari et al, 2011). Human CD177 is a ~ 50 kDa glycosylphosphatidylinositol (GPI)-anchored surface glycoprotein (Kissel et al, 2001). It is a member of the lymphocyte antigen-6 (Ly6)/urokinase plasminogen activator receptor (uPAR) family, which is characterised by the presence of Ly6/uPAR (LU) domain(s). CD177 contains four predicted LU domains, followed by a linker and a GPI-anchor motif. A typical LU domain consists of 60–90 amino acids and is stabilised by five disulfide bridges. The fold resembles the shape of three outstretched fingers of a hand, and so it has been described as a 'three-fingered protein fold'. Surface plasmon resonance (SPR) spectroscopy using neutrophil-derived PR3 (nPR3) and a recombinant PR3 showed high-affinity binding to CD177 ($K_D = 4.1$ nM) (Jerke et al, 2017). Pioneering experiments suggested that a 'hydrophobic patch' on human PR3, which protects from apoptotic cell clearance (Kantari et al, 2011; Gabillet et al, 2012), is centred around the residues F180, F181, I221, W222, L228, and F229. This patch is not conserved in other primates and is thought to bind to hydrophobic surfaces on CD177 (Korkmaz et al, 2008). These findings are consistent with the fact that gibbon PR3 does not interact with human CD177-expressing cells (Korkmaz et al, 2008). However, subsequent work showed that mutation of I221 and W222 to alanine reduced, but did not abolish, PR3 binding to CD177 (Jerke et al, 2017). Thus, the precise mechanism of PR3-CD177 interactions in vivo still remains to be shown.

In healthy individuals, the PR3-CD177 complex presents a key checkpoint in signalling cascades that control neutrophil activation in response to inflammatory signals. It forms a higher-order complex with two G-protein-coupled receptors, GPR97 and PAR2, and the Fc receptor CD16b (Chu et al, 2022). Robust PR3 enzymatic activity requires interaction with GPR97, which thereby promotes cleavage and activation of PAR2, triggering neutrophil activation (Chu et al, 2022). Not all neutrophils, however, express CD177, resulting in two distinct CD177[pos]/PR3[high] and CD177[neg]/PR3[low] neutrophil subpopulations, with PR3[high] and PR3[low] here referring to extracellular membrane-bound PR3 concentrations (Abdgawad et al, 2010; Von Vietinghoff et al, 2007). The ratio of CD177[pos]/PR3[high] versus CD177[neg]/PR3[low] neutrophils varies between individuals, is genetically determined, and stable over time (Schreiber et al, 2003). PR3-ANCA activate CD177[pos]/PR3[high] neutrophils more strongly compared to CD177[neg]/PR3[low] neutrophil populations (Schreiber et al, 2004).

The mechanisms of how PR3-ANCA interact with target PR3 are not fully understood, but there is evidence that some PR3-ANCA-binding epitopes are more pathogenic than others (Daouk et al, 1995; Van Der Geld et al, 2002; Dolman et al, 1993). Pioneering work in mapping PR3 epitopes used synthetic over-lapping peptides (Williams et al, 1994; Griffith et al, 2001), recombinant chimeric recombinant PR3 (Silva et al, 2010; Kuhl et al, 2010; Van Der Geld et al, 1999), and competition experiments with mouse monoclonal antibodies (mAbs) or a patient-derived PR3-ANCA (Kuhl et al, 2010; Silva et al, 2010; Van Der Geld et al, 1999). The latter, together with grafting experiments, revealed four major conformational epitopes on the PR3 surface targeted by anti-PR3 antibodies. An anti-CD177 Fab, which blocks PR3-CD177

binding, was shown to remove CD177-bound PR3 from neutrophil surfaces, presumably by competing for binding (Marino et al, 2022). The presence of this anti-CD177 Fab also reduced PR3-ANCA-induced activation of CD177[pos]/PR3[high] neutrophils, down to the lower level measured for CD177[neg]/PR3[low] neutrophils (Marino et al, 2022).

Immunoassays are the primary screening method for patients suspected of AAV. The presence of PR3-ANCA in these assays, in association with the correct clinical features and histology, confirms the diagnosis of GPA (Bossuyt et al, 2017). While PR3-ANCA are frequently present in GPA, their concentrations fail to consistently predict disease activity and relapse propensity (Cornec et al, 2016; Boomsma et al, 2000; Thai et al, 2014; Kemna et al, 2015; Nowack et al, 2001; Van Dam et al, 2021). Persistent PR3-ANCA levels have also been detected in remission and in asymptomatic patients, suggesting that not all PR3-ANCA populations trigger disease equally (Cui et al, 2010; Lurati-Ruiz and Spertini, 2005; Thai et al, 2014). However, persistent PR3-ANCA positivity in GPA is associated with a greater risk of relapse (Van Dam et al, 2021). There is currently no specific cure for GPA, and standard treatment regimens rely on the use of various immunosuppressants and immunomodulating drugs, such as cyclophosphamide, rituximab, methotrexate, azathioprine and myco-phenolate mofetil (Reggiani et al, 2024), which can have adverse effects. Long-term follow-up is essential as there is a significant risk of disease relapse (Hellmich et al, 2023). The success of a complement 5a receptor antagonist, Avacopan, in phase 3 trials has facilitated quicker reduction in corticosteroids (Jayne et al, 2021). Taken together, current drug regimens result in considerable morbidity and contribute to mortality, underpinning the urgent need to better understand the molecular mechanisms at play in order to develop better treatments.

Here, we present the crystal structures of a PR3-CD177 complex and of an unliganded CD177 construct containing all four LU domains. The study was possible using a site-directed mutagenesis screen to identify stable PR3 variants that are readily expressed and purified from HEK293 cultures. We describe a mainly hydrophobic binding interface between PR3 and CD177, involving the first two LU domains. The crystal structure shows that CD177 is composed of two subdomains, each containing two tightly packed LU domains, that are connected by a flexible linker. We employ a panel of PR3-ANCA-positive GPA patient samples to show that the recombinant PR3 is effective in PR3-ANCA detection assays. We discover that most of the PR3-ANCA-positive samples tested here target the CD177-binding site of PR3 and display reduced binding to PR3 protein, where the CD177-binding site is masked. Altogether, those results suggest that the CD177-binding surface of PR3 is a major autoantibody epitope, targeted in most GPA patients examined in this study.

## Results

### Crystal structure of CD177 LU1-LU4

We expressed and purified recombinant human CD177 ectodomain (CD177[ecto], residues 1–408) from HEK293 cultures and crystallised the protein using the vapour-diffusion method. Crystals of CD177[ecto] diffracted to 2.7 Å and 2.87 Å resolution at X-ray wavelengths of 1.0718 Å and 2.7552 Å, respectively (Table 1). The structure was solved using a combination of sulphur-SAD phasing

**Table 1. Data collection and refinement statistics for CD177ᵉᶜᵗᵒ.**

| | Native | S-SAD |
|---|---|---|
| **Data collection** | | |
| Source | I03 (DLS) | I23 (DLS) |
| Wavelength λ (Å) | 1.0718 | 2.7552 |
| Number of crystals | 1 | 3 |
| Resolution range (Å) | 96.92-2.70 (2.83-2.70) | 96.46-2.87 (3.01-2.87) |
| Space group | H 3 | H 3 |
| Cell dimensions | | |
| a, b, c (Å) | 192.990, 192.990, 71.808 | 192.930, 192.930, 71.980 |
| α, β, γ (°) | 90, 90, 120 | 90, 90, 120 |
| Unique reflections | 27,651 (3693) | 22,589 (3071) |
| Multiplicity | 7.0 (6.8) | 100.8 (56.5) |
| Completeness (%) | 100 (100) | 98.9 (92.4) |
| Wilson B (Å²) | 76.3 | 85.22 |
| $R_{meas}$ (%) | 22.5 (316.7) | 21.2 (679.9) |
| $R_{pim}$ (%) | 8.3 (120) | 2.9 (118.0) |
| $CC_{1/2}$ (%) | 99.4 (36.3) | 100 (56.7) |
| Average I/σ (I) | 4.2 (0.5) | 22.2 (1.1) |
| Anomalous completeness (%) | | 98.4 (89.1) |
| **Refinement** | | |
| Resolution range (Å) | 66.06-2.7 | |
| Reflections (work/free set) | 26068/1296 | |
| $R_{work}/R_{free}$ (%) | 22.36/25.85 | |
| **Number of atoms** | | |
| Protein | 5592 | |
| Ligand | 68 | |
| **Mean B value (Å²) (overall)** | 101.9 | |
| Protein | 101.6 | |
| Ligand | 130.6 | |
| RMSD bonds (Å) | 0.007 | |
| RMSD angles (°) | 2.043 | |
| **Ramachandran** | | |
| Favoured (%) | 96.39 | |
| Allowed (%) | 3.61 | |
| Outliers (%) | 0 | |
| Molprobity score/ percentile | 1.71 | |

Values in parentheses are for the highest resolution shell. $R_{meas}$ for the multiplicity-corrected merging R factor, $R_{pim}$ for the precision-indicating merging R factor and $CC_{1/2}$ for the correlation coefficients between random half datasets. RMSD is the acronym for root-mean-square deviation from ideal geometry.

and molecular replacement, and confirmed the presence of four LU domains (LU1-LU4) in CD177ᵉᶜᵗᵒ (Figs. 1A–F and EV1A,B; PDB: 9IGP). Each LU domain forms the characteristic 'three-finger' fold in which the six β-strands are stabilised by three (LU1 and LU3) or four (LU2 and LU4) disulphide bridges (Figs. 1E,F and EV1A). Unexpectedly, the LU domains are arranged into two compact subdomains, here termed subdomain I and II, which are connected

by a 20-residue linker. Within each subdomain, two LU domains (LU1 + LU2, LU3 + LU4) pack closely against each other in a head-to-tail manner, forming a globular unit that is stabilised by a network of hydrophobic interactions (Fig. 1B–F). Superposition of the two CD177ᵉᶜᵗᵒ copy subunits found within the asymmetric unit of the crystal shows that the relative orientations of subdomain I to II differ in the two copies, suggesting that the linker between the two subdomains is flexible (Fig. EV1C).

## PR3 binds subdomain I of CD177

In agreement with previously published studies (Jerke et al, 2017), we found that native-sequence recombinant human PR3 tends to oligomerise into non-homogenous particles when purified and concentrated, hampering biophysical studies using the protein. We also faced similar challenges with a recombinant PR3 variant, which lacks the N-terminal pro-dipeptide (ΔA26E27) and contains a mutation in the catalytic site (S203A) (Sun et al, 1998; Specks et al, 1996; Capizzi et al, 2003; Finkielman et al, 2007; Lee et al, 2005). This tendency for aggregation has been attributed to the presence of a conserved 'hydrophobic patch' (Fig. EV1D,E) that forms part of the surface of PR3 and contains predominantly hydrophobic and positively charged residues (Jerke et al, 2017; Hajjar et al, 2010; Broemstrup and Reuter, 2010; Kantari et al, 2011). Using an established protein engineering approach that introduces an N-linked glycosylation site (Akkermans et al, 2022; del Toro et al, 2020), we mutated three residues within the PR3 hydrophobic surface patch: I221N-W222G-G223T (Fig. EV1F). A previously published mutant in which two of these residues were mutated to alanine residues (I221A-W222A) had also been reported to improve protein solubility (Jerke et al, 2017). Further, we removed the N-terminal pro-dipeptide and mutated the catalytic site, S203A, to produce a glycosylated PR3 variant (PR3ʳᵉᶜ, residues 1–256) that is readily expressed at high levels and yields stable and monodisperse protein samples (Fig. EV1G–I). Surface plasmon resonance (SPR) and SEC-MALS experiments confirmed that PR3ʳᵉᶜ binds CD177ᵉᶜᵗᵒ with high affinity ($K_D = 10.8 \pm 0.2$ nM), forming a 1:1 complex (Fig. EV2). $k_{on}$ and $k_{off}$ values were $1.11 \times 10^6 \pm 5.78 \times 10^3$ M$^{-1}$ s$^{-1}$ and $0.00817 \pm 0.00080$ s$^{-1}$, respectively, for PR3ʳᵉᶜ binding to CD177ᵉᶜᵗᵒ. Domain deletion experiments showed that subdomain I of CD177 (CD177ˢᵘᵇᴵ, residues 1-206) is sufficient and necessary for binding and that subdomain II does not bind PR3ʳᵉᶜ (Fig. EV2F).

To produce PR3-CD177 complex crystals, we expressed CD177ˢᵘᵇᴵ in HEK293 cultures and crystallised it with PR3ʳᵉᶜ. The resulting crystals diffracted to 1.5 Å resolution and the complex structure was solved using molecular replacement (Figs. 2A–C and EV3A; Table 2; PDB: 9IGO). The data show that PR3ʳᵉᶜ binds CD177 at the linker that connects LU1 to LU2 in subdomain I, burying a total of ~655 Å² surface area. The site is distal to the linker leading into LU3, and therefore in agreement with only subdomain I being involved in PR3 binding (Figs. 2A–C and EV2F). The structure of PR3ʳᵉᶜ in this model is similar to that of unliganded PR3 (Fujinaga et al, 1996) with a low backbone C-alpha root-mean square deviation ($C_\alpha$ RMSD) of only 0.538 Å for 214 aligned atoms, and most differences are found in peripheral regions such as the CD177-binding loops (Fig. EV3B). These are contributed mainly by the linkers found between β-strands 9–10 and 11–12, plus additional contacts contributed by the loops between β-strands 6–7 and 8–9 (β-strand numbering as in (Hajjar et al,

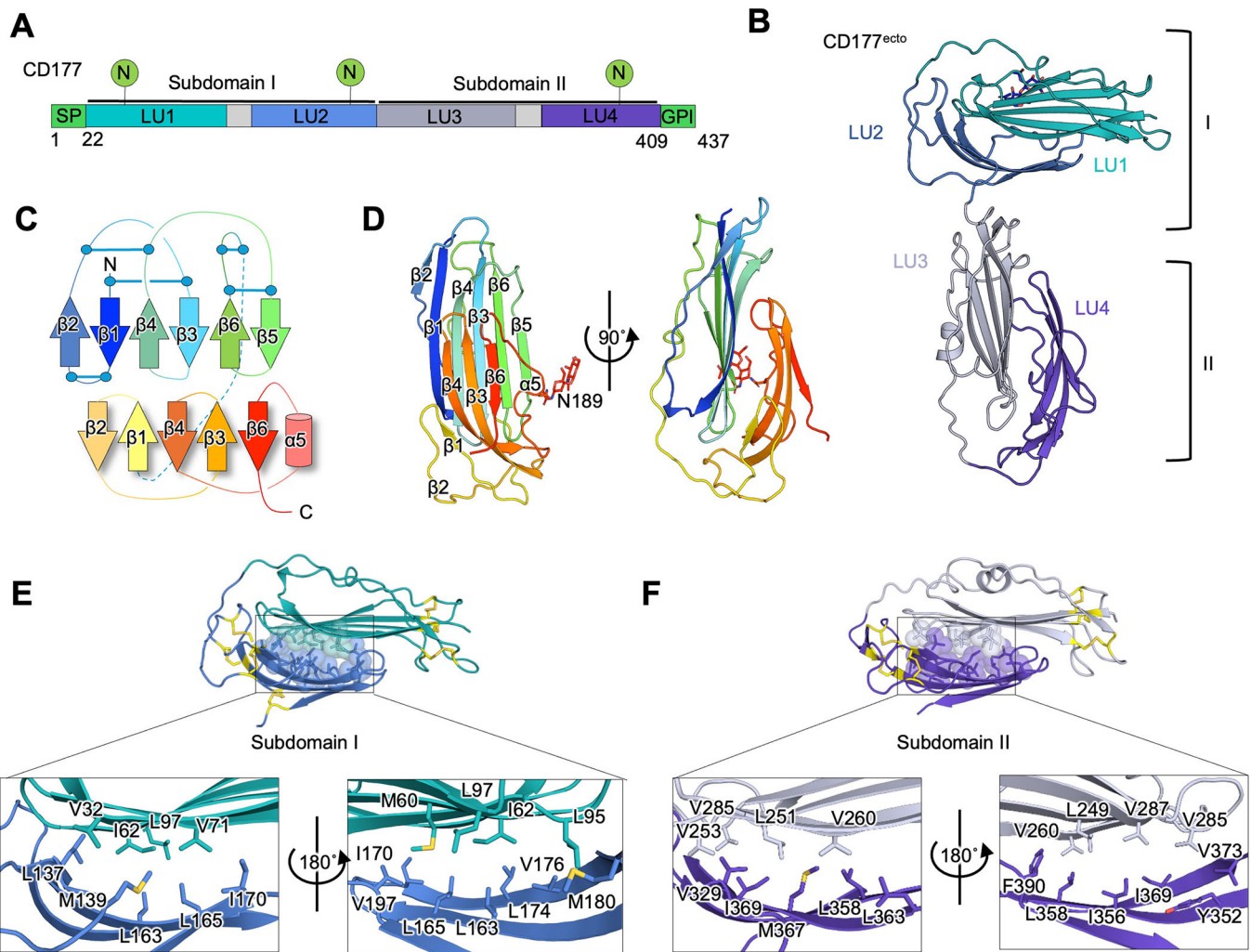

**Figure 1. Structure of CD177 LU1-LU4.**

(A) Domain overview of human CD177. (B) X-ray crystallography structure of CD177^ecto encompassing LU domains 1–4. LU1 + LU2 and LU3 + 4 form separate subdomains - PDB: 9IGP. (C) Schematic diagram of the CD177 subdomain architecture. (D) Ribbon representation of CD177 subdomain I (LU1 and LU2), coloured according to the rainbow from blue (N-terminus) to red (C-terminus). β-strands of each LU domain are numbered 1–6. β6 of LU1 (green) extends to form a loop, which packs against LU1 so that LU1 and LU2 are arranged in a head-to-tail conformation. (E) Views of the interface formed by CD177 LU1 and LU2. (F) As in (E), but showing LU3 and LU4.

2010), Fig. EV3B). The centre of the binding surface contains hydrophobic residues (F180, F181, F190 and L228 from PR3, and L117, P118 and W120 from CD177) (Fig. 2D–F). These residues are located within the previously described 'hydrophobic patch' of PR3 (Korkmaz et al, 2008) (Fig. EV3C). Peripheral interactions involve H148, V178, P192, F229, P230, R193 and R227 from PR3, and V114, P122 from CD177. Hydrogen bonds are formed between R193 and L228 from PR3, with L119 and N115 of CD177, respectively (Fig. 2E,F). The data show that CD177 binding does not occlude the PR3 catalytic site, possibly leaving it available for substrates. The modified residues in PR3^rec (I221N, W222G, G223T) are not within the binding surface of CD177, explaining why PR3^rec binds CD177 effectively (Fig. 2A–C). Glycans were modelled into relevant density within the map at predicted N-glycosylation sites (N129, N174, N221 for PR3, and N189 for CD177). We were able to resolve the PR3^rec C-terminus up until residue K253, suggesting that at least part of the C-terminal pro-peptide is intact (Fig. EV3D).

Based on the structural data, we engineered a non-CD177-binding mutant of PR3^rec (PR3^nonCD177) by introducing a N-linked glycosylation site in the binding site (T179N-F180G-F181T; Fig. 3A). In SPR experiments, PR3^nonCD177 does not bind CD177 (Fig. 3B). We also compared the binding of PR3^rec, and PR3^nonCD177 to CD177 expressed on HEK293 cells using fluorescence-activated cell sorting (FACS), confirming our SPR results (Fig. 3C). AlphaFold (Jumper et al, 2021) predicts that the fold of the PR3^nonCD177 mutant is similar to nPR3 and PR3^rec (Fig. EV3E).

## PR3^rec is effective in capturing PR3-ANCA in ELISA assays

The presence of serum PR3-ANCA is a key biomarker for GPA (Jennette and Falk, 2014; Nakazawa et al, 2019; Pfister et al, 2004). Alongside immunofluorescence protocols, standardised ELISA assays are used in clinical settings to detect and quantify PR3-

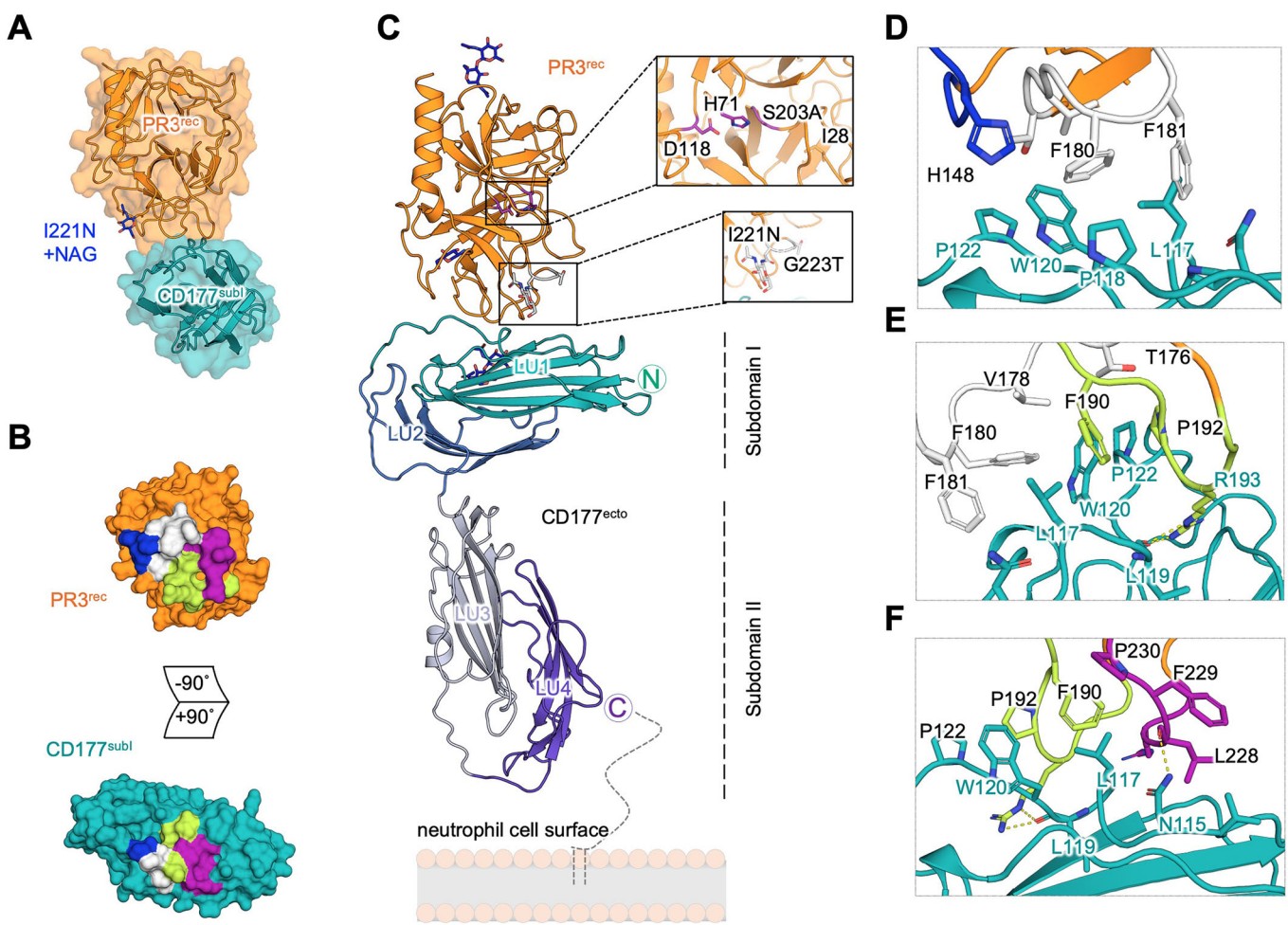

**Figure 2. Structure of PR3-CD177 complex.**

(A) Crystal structure of the PR3rec-CD177subl complex - PDB: 9IGO. The glycan introduced to produce PR3rec is highlighted in blue. The structure of CD177subl is shown below. (B) 'Open book' surface views of the PR3rec-binding surface on CD177subl. Distinct areas of the binding interface are coloured separately. (C) A composite model of CD177ecto and PR3rec was produced by overlaying the structures of the PR3rec-CD177subl complex and CD177ecto. Residues mutated to produce PR3rec are shown in stick representation in insets. (D–F) Close-up views of the PR3rec-CD177subl interface. Colours as in (B).

ANCA in patient serum samples. While these ELISA assays utilise recombinant PR3 peptides, some also rely on nPR3, which requires costly protein production and is subject to batch-to-batch variations. Our FACS experiments suggested that nPR3 and PR3rec bind to CD177 at similar levels (Fig. 3C). Here, we establish a standardised ELISA assay that uses variants of PR3rec and CD177ecto as ligands, resulting in a sensitive test which provides additional epitope information. We report all results of ELISA experiments as arbitrary binding units (PR3 AU). In a proof of principle experiment, we immobilised equal amounts of the following ELISA ligands, nPR3, PR3rec, PR3nonCD177 or PR3rec+CD177ecto and CD177ecto (where CD177 is TwinStrep-tagged) (Figs. EV2E and EV4A), and used a PR3-ANCA standard sample, which was composed of equal amounts of crude double-spun plasma derived from three PR3-ANCA positive patients (P02 + P05 + P08) at 1:1600 (Fig. EV4A) and 1:50 dilution (Fig. EV4B). These ELISA experiments demonstrated that PR3rec ligand produces a quicker and stronger positive signal compared to commercially available

nPR3, suggesting that it is more effective at detecting patient PR3-ANCA (Fig. EV4B). To confirm that similar levels of PR3rec variants were immobilised in these experiments, we used a human recombinant monoclonal anti-His specifically targeting the poly-histidine tag present on both proteins. The observed binding signal indeed confirmed that equal amounts of PR3rec and PR3nonCD177 were immobilised (Fig. 3E). We also used a mouse anti-Strep monoclonal antibody to confirm that equal amounts of CD177ecto were immobilised (Fig. 3E). Taken together, these data show that PR3rec is readily immobilised, is recognised by relevant monoclonal antibodies, and acts as a high-affinity ligand in PR3-ANCA detection assays. In subsequent ELISA experiments, we used human recombinant anti-His for standardisation, allowing us to compare PR3-ANCA binding to different His-tagged PR3rec variants (Fig. 4A). We used either crude plasma samples or purified immunoglobulin G (IgG) preparations from a panel of eight clinically confirmed GPA (PR3-ANCA positive) patient samples, P01-P08. A disease control sample (PCtrl) was derived

**Table 2. Data collection and refinement statistics for PR3-CD177$^{\text{subI}}$.**

| Data collection | |
|---|---|
| Source | PX-1 (SOLEIL) |
| Wavelength λ (Å) | 0.9786 |
| Number of crystals | 1 |
| Resolution range (Å) | 60.37–1.50 (1.53–1.50) |
| Space group | P21 |
| Cell dimensions | |
| a, b, c (Å) | 39.760, 120.735, 61.983 |
| α, β, γ (°) | 90, 96.992, 90 |
| Unique reflections | 92506 (4607) |
| Multiplicity | 6.6 (6.7) |
| Completeness (%) | 99.9 (99.9) |
| Wilson B (Å$^2$) | 19.2 |
| $R_{\text{meas}}$ (%) | 10.1 (205.2) |
| $R_{\text{pim}}$ (%) | 3.9 (78.7) |
| $CC_{1/2}$ (%) | 99.8 (65.2) |
| Average I/σ (I) | 8.6 (0.7) |
| **Refinement** | |
| Resolution range (Å) | 43.09-1.50 |
| Reflections (work / free set) | 87744/4684 |
| $R_{\text{work}}$ / $R_{\text{free}}$ (%) | 14.54/18.39 |
| **Number of atoms** | |
| Protein | 3157 |
| Ligand | 109 |
| Solvent | 429 |
| **Mean B value (Å$^2$) (overall)** | 33.3 |
| Protein | 30.8 |
| Ligand | 55.0 |
| Solvent | 46.4 |
| RMSD bonds (Å) | 0.013 |
| RMSD angles (°) | 1.15 |
| **Ramachandran** | |
| Favoured (%) | 98.74 |
| Allowed (%) | 1.26 |
| Outliers (%) | 0 |
| Molprobity score/percentile | 1.712 |

Values in parentheses are for the highest resolution shell. $R_{\text{meas}}$ for the multiplicity-corrected merging R factor, $R_{\text{pim}}$ for the precision-indicating merging R factor and $CC_{1/2}$ for the correlation coefficients between random half datasets. RMSD stands for root-mean-square deviation from ideal geometry.

from a patient with myeloperoxidase-specific autoantibodies (MPO-ANCA) associated with vasculitis. A healthy control sample was also included (Figs. 4C and EV5A–C). ELISA experiments with these samples resulted in a range of positive responses for all PR3-ANCA positive samples, suggesting that the assay works as intended (Figs. 4B and EV5B,C). The binding pattern differed between plasma and IgG samples, suggesting that there could be

factors in plasma that interfere with the PR3-ANCA interaction. For comparison, we also used a commercially available 'cANCA proteinase' ELISA kit (Euroimmun) with 8-fold more concentrated plasma samples compared to our PR3$^{\text{rec}}$ ELISA experiments, to match the instructions specified by the manufacturer (Fig. EV5D–G). Similar to our PR3$^{\text{rec}}$ ELISA experiments, the change in binding patterns between plasma and IgG samples could also be observed with the commercial kit. We comment on the differences seen in 'Discussion'.

## Most GPA patient samples in this study harbour autoantibodies that target the CD177-binding site of PR3

Previous work using anti-CD177-derived Fab fragments targeting the PR3-binding surface of CD177 has suggested that removing CD177-bound PR3 from the surface of neutrophils could reduce PR3-ANCA-mediated activation of neutrophils (Marino et al, 2022). This work raised the question whether patients could also express human PR3-ANCA that interfere with PR3-CD177 interactions. We used our panel of plasma and purified IgG samples to address this using PR3$^{\text{rec}}$+CD177$^{\text{ecto}}$ complex as the ligand in ELISA experiments (Fig. 4A). Compared to ELISA assays with unliganded PR3$^{\text{rec}}$, we measured significantly reduced signals for five out of eight PR3-ANCA positive IgG samples (P01, P03, P04, P07, P08) when tested against PR3$^{\text{rec}}$+CD177$^{\text{ecto}}$ (Fig. 4B). The level of reduction varied between different patients. Experiments using plasma samples essentially recapitulate these results, with the exception of P01, P02 and P08.

An orthogonal way to probe antigen epitopes is to mutate their surfaces. The advantage of using mutations is that these cause an irreversible change to the epitope surface properties and thus cannot be 'outcompeted' by particularly strongly binding PR3-ANCA. We chose the PR3$^{\text{nonCD177}}$, which introduces an N-linked glycosylation site in the CD177-binding surface for our experiments. In agreement with the results using the PR3$^{\text{rec}}$+CD177$^{\text{ecto}}$ complex, we found a significant reduction of binding to PR3$^{\text{nonCD177}}$ compared to PR3$^{\text{rec}}$ for both IgG and plasma samples P02, P03, P04 and P07. In addition, P01, P05 and P08 displayed significantly reduced binding using the IgG samples (Fig. 4B). To further confirm those observations, we also tested our panel of plasma and purified IgG samples using PR3$^{\text{rec}}$+CD177$^{\text{subI}}$ complex as the ligand. A significant reduction in signal could be measured for three out of eight PR3-ANCA positive IgG samples (P02, P04 and P07), and two corresponding plasma samples (P04 and P07). Interestingly, the decrease in binding was sometimes less drastic for PR3$^{\text{rec}}$+CD177$^{\text{subI}}$ compared to PR3$^{\text{rec}}$+CD177$^{\text{ecto}}$. One explanation for this could be the steric hindrance introduced when using CD177$^{\text{ecto}}$ instead of CD177$^{\text{subI}}$.

Taken together, those results confirmed the existence of PR3-ANCA that targets the CD177-binding site on PR3, and that this interface harbours an epitope that is targeted by PR3-ANCA in most patients in this study (Fig. 5A).

## Discussion

The hydrophobic nature of wild-type PR3, which is thought to be due to the "hydrophobic patch" on the protein surface (Korkmaz et al, 2008), has presented a challenge for the production and

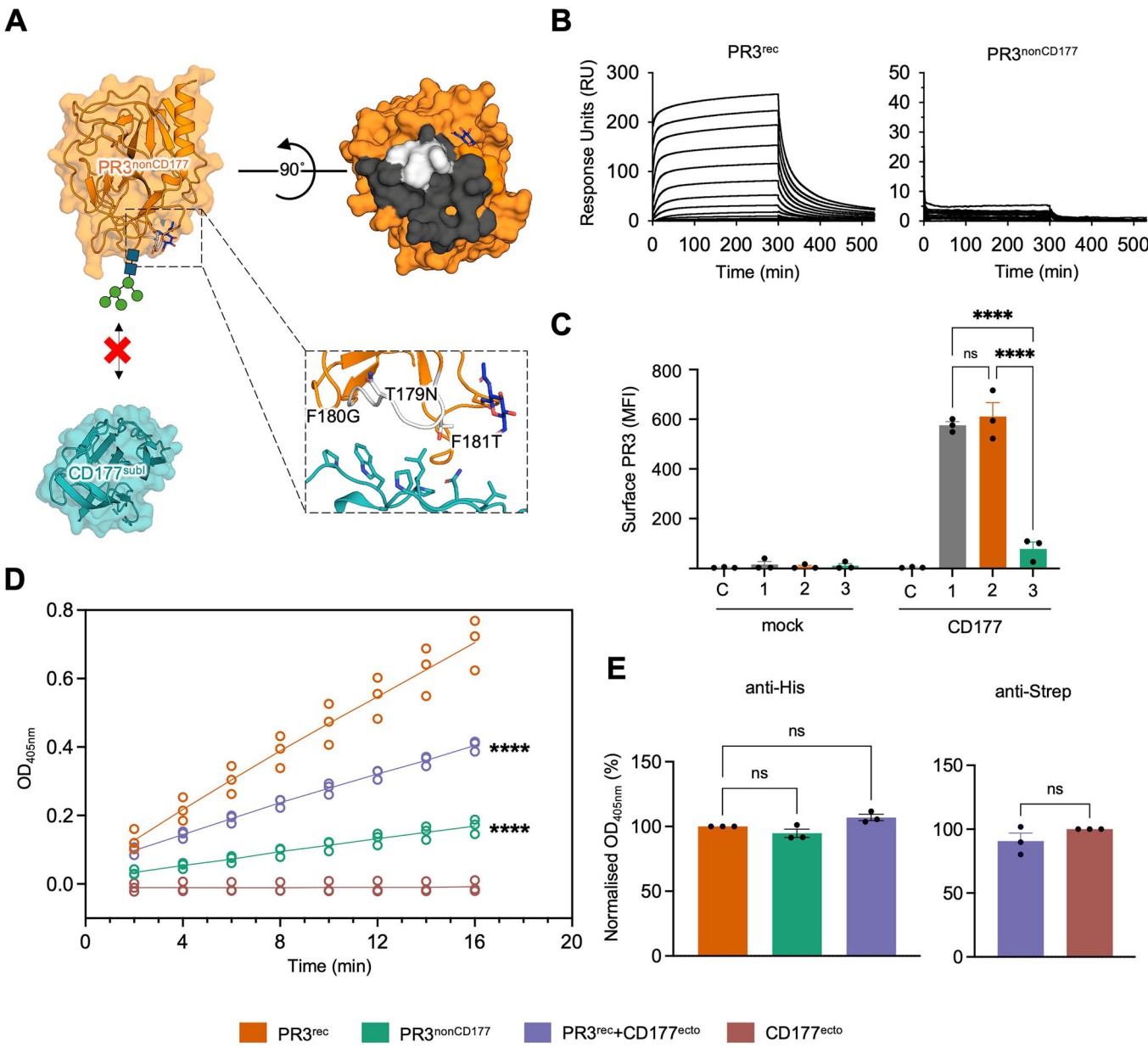

**Figure 3. High efficacy ELISA assays show epitope-specific binding.**

(A) T179N, F180G, F181T mutations were introduced in the CD177-binding site of PR3$^{nonCD177}$ to abolish CD177 binding. (B) SPR data show that CD177$^{ecto}$ analyte binds PR3$^{rec}$ ligand, but not PR3$^{nonCD177}$ ligand. A twofold dilution series of concentrations was injected, and the response over time is plotted. (C) FACS experiment using transfected HEK293 cells expressing full-length CD177 (or mock-transfected) shows binding to PR3 variants. Mock-transfected (Mock) and CD177-expressing (CD177) HEK293T cells were incubated without (C = control) or with various PR3 variants (1 = nPR3, 2 = PR3$^{rec}$, 3 = PR3$^{nonCD177}$) and then probed with a mouse anti-PR3 mAb, followed by a fluorescence-labelled secondary antibody. The extent of PR3-CD177 interaction was presented as the mean fluorescence intensity (MFI) of surface PR3 displayed on transfectant cells. $n = 3$. Two-way ANOVA analysis was carried out with the Tukey post hoc test. (D) Equal amounts of PR3$^{rec}$, PR3$^{nonCD177}$, PR3$^{rec}$+CD177$^{ecto}$ and CD177$^{ecto}$ were coated on ELISA plates for time course measurements using GPA patient plasma mixed in equal amounts (P02, P05, P08) tested at a 1:1600 dilution. Simple linear regression analysis was carried out. The binding of PR3$^{nonCD177}$ and PR3$^{rec}$+CD177$^{ecto}$ was compared to PR3$^{rec}$ binding. $n = 3$. (E) ELISA assays where nPR3, PR3$^{rec}$, PR3$^{nonCD177}$, PR3$^{rec}$+CD177$^{ecto}$ and CD177$^{ecto}$ proteins were tested for their recognition by the anti-His and anti-Strep mAb. PR3$^{rec}$ and PR3$^{nonCD177}$ are His-tagged, while CD177$^{ecto}$ is TwinStrep-tagged. Human anti-His detects recombinant His-tagged proteins (PR3$^{rec}$ and PR3$^{nonCD177}$) and mouse anti-Strep detects TwinStrep-tagged CD177$^{ecto}$. For anti-His binding, one-way ANOVA analysis was carried out with Dunnett's post hoc and for anti-Strep an unpaired, two-tailed $t$ test was carried out. $n = 3$. *$P \le 0.05$, **$P \le 0.01$, ***$P \le 0.001$, ****$P \le 0.0001$. Data are presented as mean ± S.E.M. For exact $P$ values, please refer to Dataset EV1. For SPR data, please refer to Dataset EV2. Source data are available online for this figure.

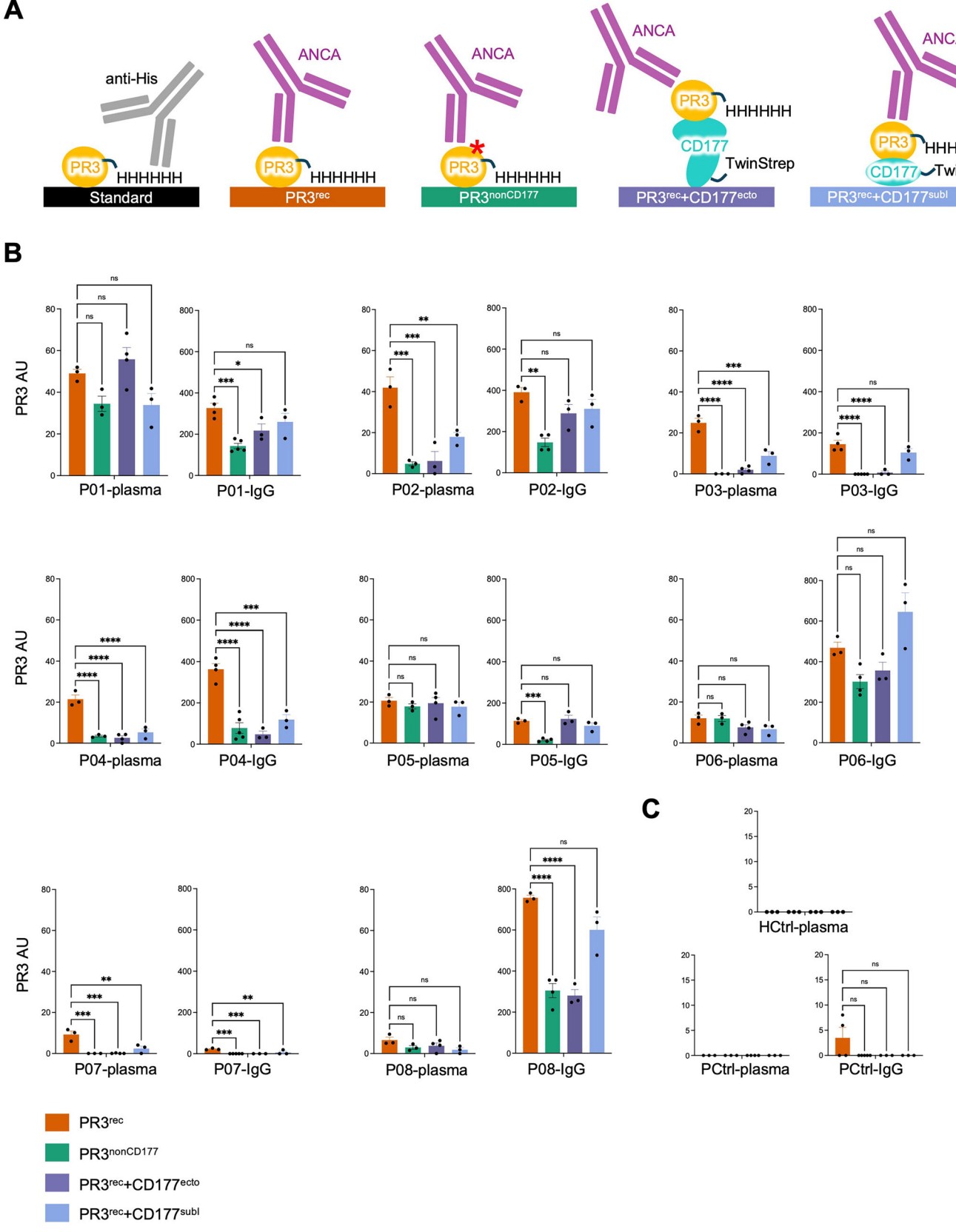

**Figure 4. ANCA binding differs for PR3rec, PR3nonCD177, PR3rec+CD177ecto and PR3rec+CD177subl (complex).**

(A) Explicative diagram of the PR3rec ELISA experiments presented in this study. Red asterisk represents the mutation on PR3nonCD177. (B) Standardised ELISA data using plasma and IgG samples from PR3-ANCA positive GPA patients (P01-P08). ANCA binding is reported in arbitrary units (PR3 AU). IgG samples were tested at 10 or 5 μg/mL. Responses were adjusted by the experiment dilution factor and then normalised by their purification dilution factor. One-way ANOVA analysis was carried out with Dunnett's post hoc test. $n = 3/4$. (C) Standardised ELISA control data using plasma and IgG samples from MPO-ANCA positive patient (PCtrl) and plasma from a healthy control sample (HCtrl). $n = 3/4$. *$P \leq 0.05$, **$P \leq 0.01$, ***$P \leq 0.001$, ****$P \leq 0.0001$. Data are presented as mean ± S.E.M. For exact $P$ values, please refer to Dataset EV1. Source data are available online for this figure.

purification of this protein for structural studies. Over the last 30 years, protocols were established for the expression of recombinant PR3 in yeast (Harmsen et al, 1997), Sf9 insect cells (Fujinaga et al, 1996), human mast cell line-1 HMC-1 (Sun et al, 1998; Specks et al, 1996; Van Der Geld et al, 2000) and HEK293 cells (Korkmaz et al, 2008; Van Der Geld et al, 2000; Jerke et al, 2017; Sun et al, 1998). Several of these purification trials have successfully used detergents such as 0.1% Triton X-100 (Halenbeck et al, 2003), 1% β-OG (Stummann and Wiik, 1997), 0.1% Tween-80 (Van Der Geld et al, 2002) or 0.02% DDM (Jerke et al, 2017) to mitigate protein aggregation. Mutagenesis of the "hydrophobic patch" residues I221 and W222 to alanine residues resulted in a monomeric construct with improved solubility (Jerke et al, 2017). Our work introduces mutations at the same site, but we opted for an N-linked glycosylation site in position 221 (PR3rec), which made production and purification from HEK293 cells and successful co-crystallisation with CD177 receptor without the use of detergent possible. The structure shows that the PR3-CD177-binding interface overlaps, at least in part, with the 'hydrophobic patch' described previously (Korkmaz et al, 2008; Jerke et al, 2017; Goldmann et al, 1999; Hajjar et al, 2008) (Fig. EV3C). Previous work suggested a negative effect of CD177 on PR3 activity: the addition of purified CD177 was shown to inhibit nPR3 activity in vitro, in neutrophil degranulation experiments, and at the surface of neutrophils or CD177-expressing HEK cells (Jerke et al, 2017). Here we show that CD177-binding does not directly occlude the PR3 active site, nor does it perturb the PR3 structure compared to unliganded PR3 (Fig. EV3B). Therefore, the mechanism of inhibition is not clear.

In analogy to MPO-ANCA, where epitope specificities differ between healthy individuals and vasculitis patients of different disease states (Chang et al, 1995; Roth et al, 2013; Selga et al, 2010; Granel et al, 2020), PR3-ANCA subpopulations target distinct epitopes on PR3 (Kuhl et al, 2010; Silva et al, 2010; Casal Moura et al, 2023). It is thought that these PR3-ANCA epitope specificities contribute to the complex pathologies seen in patients and to the poor prognostic value of total PR3-ANCA blood titres. Given that in most people, extracellular neutrophil PR3 largely exists within supramolecular complexes that include CD177 and other neutrophil surface proteins (Chu et al, 2022), we developed two approaches to investigate whether PR3-ANCA subpopulations target the newly described CD177-binding site: we used competition experiments with purified CD177 protein and a structure-based mutation in the CD177-binding site of PR3. In these experiments, we developed the use of anti-His for standardisation, rather than the antigen itself, to produce directly comparable readouts for different recombinant ligands. Comparisons demonstrated that PR3rec+CD177ecto, PR3rec+CD177subl and the PR3nonCD177 mutant lead to broadly similar effects, i.e. reduced binding of PR3-ANCA, both pointing to the conclusion that PR3-ANCA can

target the CD177-binding site on PR3. However, there are subtle differences between the results: for example, IgG sample P05 does not bind PR3nonCD177, but it still interacts with PR3rec+CD177ecto complex. A possible explanation is that this population of PR3-ANCA requires the CD177-binding site and that it outcompetes CD177 for binding to PR3. Alternatively, this sample could contain a PR3-ANCA subpopulation, which, although not inhibited by the mutation, requires the presence of CD177 to bind. In comparing different patient samples, we also found that some PR3-ANCA populations depended on access to the CD177-binding site for most of their binding (P03, P04 and P07) while others are less dependent on this site, such as P06. This variability suggests that different, potentially polyclonal antibody epitope specificities may predominate in different patients. The use of purified IgG samples is not practical in a day-to-day clinical setting; therefore, we also analysed matched plasma samples. Overall, the results using plasma correlate with those using purified IgGs. However, some differences were observed when, for example, P08 displayed a much lower response when using plasma samples instead of IgG preparations. A likely explanation is that plasma factors such as fibrinogen interfere with the immunoassay, as previously suggested for other ELISA experiments (Kwak and Lee, 2019). In future, these issues could potentially be circumvented by removing fibrinogen with protamine treatment or by using serum instead. Another explanation is that other ANCA isotypes, such as IgM and IgA, compete with IgG binding. IgM and IgA ANCA, alongside IgG ANCA, are present in systemic vasculitis (Esnault et al, 1992, 1993; Kelley et al, 2011; Jeffs et al, 2019; Jayne et al, 1989). These antibody isotypes could compete with each other for binding (Muthana et al, 2015). Furthermore, anti-idiotypic antibodies against ANCAs exist in healthy and disease populations (Jayne et al, 1993a, 1993b) and, as they are often IgM, could account for some differences seen between plasma and IgG preparations. Indeed, previous studies identified IgM anti-PR3 autoantibodies (Sibilia et al, 1997; Davis et al, 1998; Peen and Williams, 2000). Analysis of one autoantibody (WGH1) indicated that the CDR3 region was accessible for interaction with positively charged residues on PR3 (Davis et al, 1998). However, a more recent study identified 19 PR3-specific B cells expressing IgG, with them being predominantly IgG1 and exhibiting an enrichment for IgG4 (Kelly et al, 2024). Another study used a patient-derived monoclonal ANCA (moANCA518) and demonstrated that it bound selectively to Epitope 3 of PR3 (Casal Moura et al, 2023).

It is important to note that deviations from the wild-type PR3 sequence (be it through swaps, mutations and/or introductions of glycans) could result in local or allosteric conformational changes of PR3, which in turn could influence ANCA binding. This was highlighted by the mutation of the previously reported Epitope 3, which led to an unexpected increase in PR3-ANCA binding, possibly due to allosteric conformational changes and the exposure of a latent epitope (Casal Moura et al, 2023). Indeed, molecular

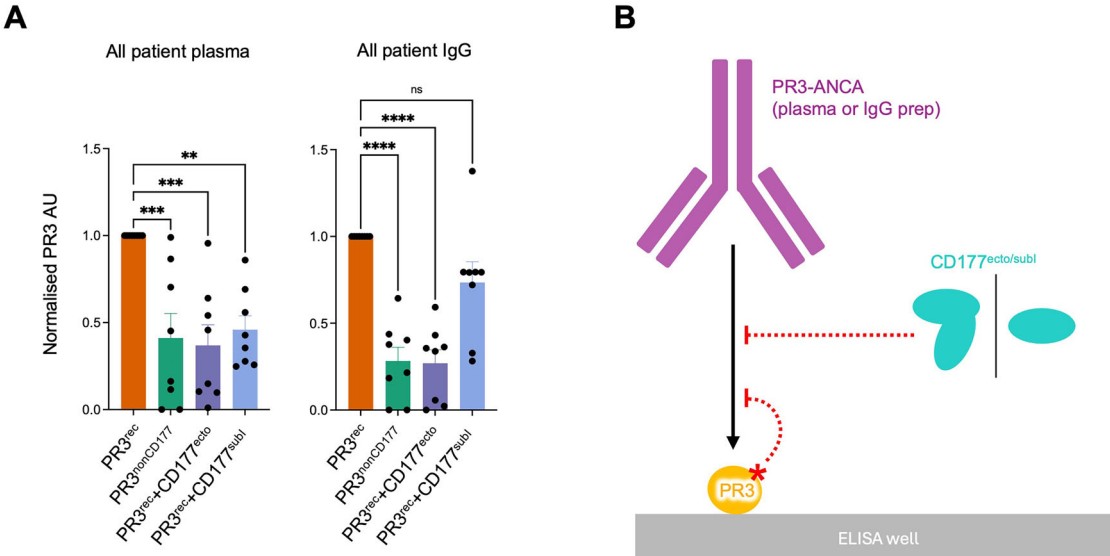

**Figure 5. Summary diagram of patient plasma and IgG ELISAs.**

(A) A summary plot of ELISA data from Fig. 4B comparing the binding of all ligands for plasma and IgG samples is shown. Data are normalised to PR3rec binding for each sample. One-way ANOVA analysis was carried out with Dunnett's post hoc test. $n = 8$. *$P \leq 0.05$, **$P \leq 0.01$, ***$P \leq 0.001$, ****$P \leq 0.0001$. For exact $P$ values, please refer to Dataset EV1. Source data are available online for this figure. (B) Summary diagram of ELISA experiments. The addition of CD177ecto/subl and PR3 with a mutation at the CD177-binding site (PR3nonCD177) interferes with PR3-ANCA binding.

dynamic simulations indicated that a distal region away from the mutation became more accessible for ANCA binding (Pang et al, 2019). Allosteric effects on PR3 activity have also been shown, for example, for mouse anti-PR3 MCPR3-7 (Hinkofer et al, 2013) and upon CD177 binding (Jerke et al, 2017). Our mutants were designed so as to minimise the predicted disturbance of the PR3 fold. Additional reassurance is given by the fact that the structure of PR3rec, as found in complex with CD177, is near-identical to the published structure of unliganded PR3 containing the native sequence (Fujinaga et al, 1996) (Fig. EV3B), and AlphaFold models suggest that PR3rec and PR3nonCD177 adopt essentially the same fold as native PR3 (Fig. EV3E). Whilst the PR3 fold is likely preserved in PR3rec, the mutations we introduced could potentially inhibit the binding of PR3-ANCA subpopulations targeting this surface. We show that PR3rec is more effective in detecting PR3-ANCA in representative PR3-ANCA-positive plasma samples compared to nPR3 (Fig. EV4B). However, one of the samples in our panel, P07, displayed relatively higher responses in the commercial PR3-ANCA ELISA screen that includes also nPR3 as ligand, compared to our PR3rec ELISA assay, where the signal is robust but weaker compared to other samples (Fig. EV5B–G). Conversely, IgG preparations from P01 produce no signal at all with the commercial screen, while it produces a robust signal in our PR3rec ELISA assay and were clinically confirmed (Fig. EV5B–G). Taken together, the results suggest that the accessibility of different PR3 epitopes varies between different experimental screens, with the PR3rec-based screen producing a robust signal with the sample set tested. Overall, the work demonstrates the need for a more standardised approach in assessing PR3-ANCA titres and epitopes, especially in the clinical context.

The idea of 'protective' ANCAs has been previously hypothesised by Kuhl et al (2010), coining the term 'harmless ANCAs'. This

hypothesis is further supported by a recent report showing that a human-derived monoclonal antibody, 4C3, which is predicted to target an epitope close to what we reveal here to be the CD177-binding site, reduces PR3-ANCA-mediated neutrophil activation in vitro (Granel et al, 2020). Mouse antibodies which compete with CD177 binding and thus reduce the cell-surface concentrations of neutrophil PR3 are also effective at reducing neutrophil activation (Marino et al, 2022). Should the idea of 'protective ANCA' hold true and coincide with those competing with CD177 for PR3 binding, then these autoantibodies could remove excess circulating soluble PR3 and neutrophil/CD177-bound PR3 (Fig. 5B), perhaps protecting patients from the effects of other, disease-causing PR3-ANCA subpopulations. It is thought that these PR3-ANCA may therefore fulfil a similar role to PR3's main and natural inhibitor, alpha-1 anti-trypsin (Korkmaz et al, 2008). The removal of excess PR3 could also inhibit PR3 binding to phosphatidylserine (PS) on circulating micro-vesicles (MVs) or apoptotic cells, thus preventing systemic dissemination of PR3 (reviewed in Martin and Witko-Sarsat, 2017). Should the hypothesis hold true, then the structural data detailing the PR3-CD177 interaction sites could underpin the development of structure-based drugs that interfere with PR3-CD177 interaction. It remains to be shown whether the 'non-interfering' ANCA epitopes, which do not lead to a clash with CD177-binding, are indeed the targets of the most harmful PR3-ANCAs, e.g. those that activate neutrophils. In agreement with previous work, we show that not all patient-derived PR3-ANCA interfere with CD177 binding (Kuhl et al, 2010). The use of different recombinant, engineered PR3 variants and associated factors, such as CD177ecto, could underpin the development of scalable, finely tuned epitope-screening platforms. Such platforms would allow monitoring of the diverse and evolving epitope landscapes during disease progression, as suggested by previous

antibody competition experiments (Rarok et al, 2003; Selga et al, 2010), ultimately offering prognostic clinical insights and supporting personalised care.

# Methods

### Reagents and tools table

| Reagent/resource | Reference or source | Identifier or catalogue number |
|---|---|---|
| **Experimental models** | | |
| HEK293T | ATCC | CRL-3216; RRID: CVCL_0063 |
| HEK293S GnTI⁻ | ATCC | CRL-3022; RRID: CVCL_A785 |
| Healthy control plasma | Cambridge Bioscience | Cat. # PLSSKF 2LHE10-F |
| **Recombinant DNA** | | |
| *hs*PR3 | cDNA from Mammalian Gene Collection | Clone 116835, GenBank BC096184.3 |
| *hs*CD177 | cDNA from Mammalian Gene Collection | Clone 34257, GenBank BC029167.1 |
| **Antibodies** | | |
| Monoclonal mouse anti-His antibody | Qiagen | Cat.#34660 |
| Monoclonal mouse anti-Strep antibody | IBA Lifesciences | Cat.#2-1507-001 |
| Polyclonal goat anti-mouse antibody conjugated to horseradish peroxidase | Sigma-Aldrich | Cat.#A0168 |
| Monoclonal human recombinant anti-His antibody | Abcam | Cat.# ab219465 |
| Polyclonal goat anti-human antibody conjugated to alkaline phosphatase | Sigma-Alrich | Cat. #A3187 |
| Monoclonal mouse anti-PR3 - clone PR3G-2 | Thermo Fisher Scientific | Cat. #MA1-40218 |
| **Oligonucleotides and other sequence-based reagents** | | |
| PCR primer: PR3$^{rec}$, forward: ATGGCTCACCGGCCCCCCAGCCCTGCCCTGGCGTCCGTGCTGCTGGCCTTGCTGCTGAGCGGTGCTGCCCGAGCTATCGTGGGCGGGCACGAGGCGCAGC | This study | |
| PCR primer: PR3$^{rec}$, reverse: GGGGCGGCCCTTGGCCTCCACACGG | This study | |
| PCR S203A mutation primer: PR3$^{rec}$, forward: CGTCCCTCGCCGCAAGGCCGGCATCTGCTTCGGAGACGCAGGTGGC | This study | |
| PCR S203A mutation primer: PR3$^{rec}$, reverse: CCTTGGATGATGCCATCACAGATCAGGGGCCACCTGCGTCTCCG | This study | |
| PR3$^{nonCD177}$ | Genscript | |

| Reagent/resource | Reference or source | Identifier or catalogue number |
|---|---|---|
| PCR NGT→IWG mutation primer on PR3$^{nonCD177}$: PR3$^{rec}$, forward: CCAGCCCAGGTCCTGCAGGAGCTCAATGTCACCGTGGTCACCTTCTTCTGCCGGCCAC | This study | |
| PCR NGT→IWG mutation primer on PR3$^{nonCD177}$: PR3$^{rec}$, reverse: CCTTGCGGCGAGGGACGAAAGTGCAAATGTTATGTGGCCGGCAGAAGAAGGTGACCACGG | This study | |
| PCR primer: CD177$^{ecto}$, CD177$^{subII}$, forward: ATGAGCCCGGTATTACTGCTGG | This study | |
| PCR primer: CD177$^{ecto}$, CD177$^{subII}$, reverse: ACCTCCCTCATGCTGAGAGGCAGG | This study | |
| PCR primer: CD177$^{subII}$, forward: ATGAGCCCGGTATTACTGCTGGCCCTCCTGGGGTTCATCCTCCCACTGCCAGGAGTGCAGGCGCTGACCTGTCATCGGGGGACCACC | This study | |
| PCR primer: CD177$^{subI}$, reverse: TTTCCTATTGCAGTTCTCAGTCATACC | This study | |
| pHLsec expression vectors | Addgene, Aricescu et al, 2006 | Cat#99845 |
| PCR primer: *hs*PR3, forward: | This study | |
| PCR primer: *hs*PR3, reverse: | This study | |
| PCR primer: *hs*CD177$^{ecto}$, *hs*CD177$^{subII}$, forward: | This study | |
| PCR primer: *hs*CD177$^{ecto}$, *hs*CD177$^{subII}$, reverse: | This study | |
| PCR primer: *hs*CD177$^{subI}$, Reverse: | This study | |
| pHLsec expression vectors | Addgene, Aricescu et al, 2006 | Cat#99845 |
| **Chemicals, enzymes and other reagents** | | |
| CloneAmpTM HiFi PCR Premix kit | Takara Bio | Cat.#639298 |
| Neutrophil PR3 | Athens research and technology | Cat.#16-14-161820 |
| Gel clean-up kit | Thermo Fischer Scientific | Cat.#12303368 |
| *Eco*RI | New England Biolabs | Cat.#0101S |
| *Age*I | New England Biolabs | Cat.#R0552S |
| *Kpn*I | New England Biolabs | Cat.#R0142M |
| *Xho*I | New England Biolabs | Cat.#R0146S |
| In-Fusion® HD Cloning kit | Takara Bio | Cat.#121416 |
| Stellar competent cells | Takara Bio | Cat.#636763 |
| lysogeny broth (LB) | Merck | Cat.# 1.10285.5000 |
| plasmid DNA miniprep kit | Thermo Fischer Scientific | Cat.#11932392 |

| Reagent/resource | Reference or source | Identifier or catalogue number |
|---|---|---|
| Subcloning Efficiency DH5α TM competent cells | Life Technologies | Cat.#18265017 |
| Dulbecco's Modified Eagle Medium (DMEM) | Gibco | Cat.#41966052 |
| Foetal Bovine Serum (FBS) | Gibco | Cat.#10437028 |
| L-Glutamine | Thermo Fischer Scientific | Cat.#25030024 |
| non-essential amino acids | Gibco | Cat.#11140035 |
| phosphate buffer saline (PBS) | Lonza | Cat.#LZBE17-516F |
| Trypsin- EDTA | Gibco | Cat.#25300054 |
| polyethylenimine (CAS #9002-98-6) | Sigma-Aldrich | Cat.#408727 |
| Opti-MEM | Gibco | Cat.#31985047 |
| plasmid DNA gigaprep kit | Invitrogen | Cat.#K210009XP |
| kifunensine (CAS #109944-15-2) | Cayman Chemical | Cat.#10009437 |
| NuPAGETM 4-12% Bis-Tris gel | Invitrogen | Cat.#NP0336 |
| Protein ladder | Invitrogen | Cat.#10747012 |
| Pre-stained protein ladder | Invitrogen | Cat.#10748010 |
| NuPAGE MES SDS running buffer | Invitrogen | Cat.#NP0002 |
| InstantBlue | Sigma-Aldrich | Cat.#ISB1L |
| 0.45 μm nitrocellulose membrane | Cytiva | Cat.#GE10600013 |
| NuPAGE transfer buffer | Invitrogen | Cat.#NP0006 |
| Tween-20 (CAS 9005-64-5) | Thermo Fisher Scientific | Cat.#10246910 |
| Skim milk powder | Thermo Fisher Scientific | Cat.#16694685 |
| Bovine serum albumin (BSA) | Sigma-Aldrich | Cat.#A7906 |
| Autoradiography films | Cytiva | Cat.#GE28-9068-35 |
| ECL detection reagents | Cytiva | Cat.#GERPN2106 |
| PNGase F | New England Biolabs | Cat.#P0704S |
| Endoglycosidase F1 | Gift from Jones Lab Grueninger-Leitch et al, 1996 | |
| Biotin | Sigma-Aldrich | Cat.#B4501 |
| FuGENE 6 | Promega | Cat.#E2691 |
| Amine coupling kit | Cytiva | Cat.#BR100050 |
| Nunc MaxiSorp™ flat-bottom plates | ThermoScientific | Cat.#735-0083 |
| Blocker™ Casein in PBS | ThermoScientific | Cat.#37528 |
| Dulbecco's PBS (DPBS) | Sigma | Cat.#D8537 |
| PBS tablets | Sigma-Aldrich | Cat.#P4417-100TAB |
| 5x Diethanolamine Buffer | Fisher | Cat.#34064 |
| 4-Nitrophenyl Phosphate Tablets (20 mg) | Sigma | Cat.#N2765 |
| Anti-human IgG (γ-chain specific)-alkaline phosphatase conjugate produced in goat | Sigma-Aldrich | Cat.#A3187 |
| Reagent Reservoirs (Costar 4870) | Fisher Scientific | Cat.#PMP-331-010C |
| 10X Buffer BXT; Strep-TactinXT Elution buffer | Fisher Scientific | Cat. #2-1042-025 |

| Reagent/resource | Reference or source | Identifier or catalogue number |
|---|---|---|
| Commercial PR3-ANCA (cANCA) ELISA kit | Euroimmun | Cat.#EA 1201-9601-2 G |
| **Software** | | |
| GraphPad Prism 9 | https://www.graphpad.com/ | |
| RStudio, version 2025.09.0 + 387 | https://www.rstudio.com/ | |
| BIAevaluation software 1.0 | Cytiva | |
| DIALS 3.8.0 via xia2 | Winter et al, 2018; Winter, 2010 | |
| XDS gen version: r0.5 | Kabsch, 2010 | |
| AIMLESS | Evans, 2011 | |
| Blend | Foadi et al, 2013 | |
| XSCALE version Jan 26, 2018 | Kabsch, 2010 | |
| SHELXC-D via hkl2map | Sheldrick, 2010; Pape and Schneider, 2004 | |
| Autosol | Terwilliger et al, 2009 | |
| Crank2 | Pannu et al, 2011 | |
| AutoBuild | Terwilliger et al, 2007 | |
| Phaser | McCoy et al, 2007 | |
| Bucaneer | Cowtan, 2006 | |
| autoBUSTER version 2.11.2 | Bricogne et al, 2011 | |
| Phenix.refine version 1.20.1_4487 | Afonine et al, 2012 | |
| Refmac5 version 5.8.0430 | Murshudov et al, 2011 | |
| Coot 0.9.8.96 | Emsley and Cowtan, 2004 | |
| Phenix interface | Liebschner et al, 2019 | |
| CCP4i2 interface | Potterton et al, 2018 | |
| Gen5.3.11 ELISA software | Agilent Technologies | |
| **Other** | | |
| T75 flask | Greiner Bio-One | Cat.#658175 |
| T175 flask | Greiner Bio-One | Cat.#660175 |
| Six-well plate | Greiner Bio-One | Cat.#657160 |
| Roller bottles | Greiner Bio-One | Cat.#681070 |
| HisTrap HP column | Cytiva | Cat.#GE17-5248-02 |
| StrepTrap XT column | Cytiva | Cat.# 29401323 |
| HiLoad® 16/600 Superdex® 200 column | Cytiva | Cat.#GE28-9893-35 |
| 3000 and 10,000 MWCO Amicon® Ultra centrifugal filters | Merck | Cat.#UFC800324, Cat. #UFC900324 |
| Biacore T200 instrument | GE Healthcare | |
| Series S Sensor Chip CM5 | Cytiva | Cat#29149603 |
| Mosquito® LV device | SPT Labtech | |
| SwissCi (MRC) 96-well 2-drop plates | SPT Labtech | |
| NanoDrop microvolume spectrophotometer | ThermoScientific | |

| Reagent/resource | Reference or source | Identifier or catalogue number |
|---|---|---|
| ELx800 | Biotek | |
| SpectraMax M3 | Molecular Devices | |
| FACSCalibur flow cytometer | BD Biosciences | |
| Eiger2 XE 16 M | Dectris | |

## Vectors and cloning

The cDNAs for PR3 (GenBank BC096184.3, UniProt P24158) and CD177 (GenBank BC029167.1, UniProt Q8N6Q3) were purchased from the Mammalian Gene Collection (MGC:116835 IMAGE:40003797 and MGC:34257 IMAGE:5182826, respectively). The PR3 cDNA contains a natural single-nucleotide polymorphism (SNP) at nucleotide 355 (G→A), resulting in V119I. The CD177 cDNA contains two SNPs at nucleotide 25 (G→C) and 1060 (G→A), resulting in A3P and A350T, respectively.

All PR3 variants have the ΔA26E27 deletion and S203A mutation, resulting in production of PR3 in its active conformation but catalytically inactive.

Constructs of PR3 (residues M1-P256 ΔA26E27) and CD177 (residues M1-G408 for CD177$^{ecto}$, M1-K206 for CD177$^{subI}$) were cloned into the EcoRI-KpnI cloning sites of vectors from the pHLsec family with their native secretion signal sequence and a C-terminal His$_6$ or TwinStrep tag (Aricescu et al, 2006). For surface plasmon resonance (SPR) experiments, constructs were cloned into the pHL-Avitag vector for fusion to a C-terminal GSS linker region, a biotin ligase (BirA) recognition site and His$_6$ tag.

## Protein expression and purification

PR3 variants and CD177$^{ecto}$ were expressed in a secreted form in N-acetylglucosaminyltransferase I-deficient HEK293S (HEK293S GnTI$^-$) or HEK293T cells using previously described protocols (Seiradake et al, 2014; Aricescu et al, 2006). Briefly, plasmid DNAs were transfected with PEI in a 2:1 ratio into 3 L of 90–100% confluent HEK293 cells. After 5 (HEK293T) or 10 days (HEK293S), cell culture medium containing the secreted proteins was clarified by centrifugation and filtration prior to diafiltration into PBS, 20 mM Tris pH 7.5 and 150 mM NaCl. The proteins were then purified by immobilised metal affinity chromatography (HiTrap HP, Cytiva or StrepTrap XT, Cytiva) and size-exclusion chromatography (Superdex 200 16/600, Cytiva) in 20 mM Tris pH 7.5 and 300 mM NaCl (SEC buffer). Protein purity was assessed by sodium dodecyl sulphate-polyacrylamide gel electrophoresis (SDS-PAGE), and fractions containing pure protein were pooled and concentrated using 3000 or 10,000 MWCO concentrators (Amicon Ultra Centrifugal Filters). Protein concentrations were measured in triplicate using a NanoDrop spectrophotometer device and the theoretical extinction coefficient calculated by the ProtParam tool on the ExPASy server (Gasteiger et al, 2005).

## PNGase F deglycosylation assay

Glycosylated proteins were incubated with PNGase F (NEB, Cat.#P0704S) in accordance with the protocol supplied with the

enzyme. In total, ~20 µg of protein were incubated with PNGase F in denaturing reaction conditions for analysis on SDS-PAGE gel, and in non-denaturing reaction conditions for SEC experiments.

## SEC-binding assay

In all, 4 nmol of CD177$^{ecto}$ was mixed with 8 nmol of PR3 variant in a maximum of 80 µl total volume. The two proteins were allowed to incubate at r.t. for at least 10 min before loading into a 100 µl tubing loop connected to a Superdex® 200 10/300 GL column previously equilibrated with SEC buffer. 250 µl fractions were collected and analysed by SDS-PAGE.

## Surface plasmon resonance

PR3 and CD177 constructs were biotinylated enzymatically in vitro at their C-terminal AviTag and coupled to a streptavidin-coated sensor chip CM5 (Cytiva).

Equilibrium binding experiments were performed at 25 °C using a Biacore T200 instrument (GE Healthcare) using PBS supplemented with 0.005% (v/v) polysorbate 20 (pH 7.5) or 20 mM Tris + 200 mM NaCl + 0.005% (v/v) polysorbate 20 (pH 7.5) as running buffers.

For dissociation constant $K_D$ calculations, concentration series of the analytes were prepared by twofold serial dilutions. Each concentration was injected for 200–300 s, followed by 300 s of dissociation time. The chip was regenerated with 2 M MgCl$_2$ followed by 300 s of stabilisation period between each injection. All the steps were carried out at a flow rate of 5 µl/min, in PBS supplemented with 0.05% (v/v) Tween-20 and 300 mM NaCl as running buffer.

Data were analysed using the BIAevaluation software 1.0, Prism 9 (GraphPad) and Rstudio. Indicative steady-state $K_D$ and RU$_{max}$ values were obtained by non-linear curve fitting of a 1:1 Langmuir interaction model (bound = (RU$_{max}$ × C)/($K_D$ + C), where C is analyte concentration calculated as monomer). For PR3$^{rec}$ binding to CD177$^{ecto}$ and CD177$^{subI}$ ligand, kinetic rate constants $k_{obs}$, $k_{off}$ were determined by the fitting association (RU$_{max}$ × (1 − exp(−$k_{obs}$ × t))) and dissociation phases (R$_0$ × exp(-$k_{off}$ × t)) with a 1:1 Langmuir binding model. $k_{on}$ was obtained by linear regression fit ($k_{obs} = k_{on}$ × C + $k_{off}$). For CD177$^{ecto}$ binding to PR3$^{rec}$ ligand, the same approach was applied at concentrations below 16 nM, where binding curves reached a clear plateau. At higher concentrations, an alternative model accounting for mass transport limitations was used (RU$_{max}$ × (1 − exp(−$k_{obs}$ × t)) + mt$_{rate}$ × t), where mt$_{rate}$ represents the linear increase due to mass transport. Final kinetic $K_D$ were calculated as $k_{off}/k_{on}$.

## Flow cytometry analysis of PR3-binding to CD177-expressing cells

Binding of native and recombinant PR3 proteins to surface CD177 overexpressed on HEK293T cells was analysed using a FACS-based binding assay as described previously (Chu et al, 2022). Briefly, mock-transfected and CD177-expressing HEK293T transfectants were incubated with native or recombinant PR3 proteins (1 µg/mL) diluted in blocking buffer (1% BSA, 5% normal goat serum/PBS) for 1 h at 4 °C. Following extensive washes in cold PBS, cells were incubated with the mouse anti-PR3 mAb (0.5 µg/mL) (clone PR3G-2, Thermo Fisher Scientific) for 1 h at 4 °C. Cells were extensively

washed and then incubated with fluorescence-labelled secondary antibody for 1 h at 4 °C. Finally, cells were washed in cold PBS and subjected to analysis by FACSCalibur flow cytometer (BD Biosciences).

## CD177$^{ecto}$ crystallisation and structure solution

CD177$^{ecto}$ crystals grew in 150 nL + 75 nL sitting nanodrops by the vapour-diffusion method at 18 °C in 20% w/v PEG 3350, 0.20 M ammonium nitrate. The crystals were harvested and cryo-protected in 25% glycerol before flash-cooling in liquid nitrogen. A native dataset was collected at a wavelength of 1.0718 Å at the Diamond Light Source (DLS) beamline I03. In all, 180° of data were collected with an exposure of 0.05 s per 0.2° rotation on an Eiger2 XE 16 M (Dectris). The data were integrated using DIALS (Winter et al, 2018) via the xia2 system (Winter, 2010) and scaled and merged using AIMLESS (Evans, 2011; Evans and Murshudov, 2013). Blend (Foadi et al, 2013) was used to merge two datasets together.

The S-SAD dataset was collected at a wavelength of 2.7552 Å at the DLS beamline I23 (Wagner et al, 2016). In all, 360° of data were collected with an exposure of 0.1 s per 0.1° rotation on a PILATUS 12 M detector (Dectris). Twelve datasets from three different crystals were individually integrated using XDS and merged using XSCALE (Kabsch, 2010). SHELXC-D within the hkl2map graphical interface (Pape and Schneider, 2004; Sheldrick, 2010) was used for substructure determination. AutoSol was used to obtain initial density-modified electron density maps (Terwilliger et al, 2009). A combination of automated model building using Crank2 (Pannu et al, 2011) and AutoBuild (Terwilliger et al, 2007) with manual building in COOT (Emsley and Cowtan, 2004) enabled to build the model step-by-step. The solution consisted of two molecules in the asymmetric unit, and several iterations of model building and AutoSol runs resulted in a nearly full model of CD177$^{ecto}$. A composite model was created by aligning the high-resolution model of CD177$^{subI}$ (see below) and the AlphaFold model of CD177$^{subII}$ (AF-G8N6Q3) onto the preliminary model of CD177$^{ecto}$, and was used as a search model for molecular replacement using the native dataset and PHASER (McCoy et al, 2007). All-atom refinement with autoBUSTER (Bricogne et al, 2011) and phenix.refine (Afonine et al, 2012) resulted in a full model containing residues L22-D397.

## PR3-CD177$^{subI}$ crystallisation and structure solution

PR3-CD177$^{subI}$ crystals grew in 100 nL + 100 nL sitting nanodrops by the vapour-diffusion method at 18 °C in 0.1 M BIS-TRIS pH 6.5, 45% v/v Polypropylene glycol P 400. The crystals were harvested and cryo-protected in 25% glycerol before flash-cooling in liquid nitrogen. A native dataset was collected at a wavelength of 0.9786 Å at the SOLEIL beamline PROXIMA-1. 360° of data were collected with an exposure of 0.01 s per 0.1° rotation on a Eiger2 XE 16 M (Dectris). The data were integrated using DIALS (Winter et al, 2018) via the xia2 system (Winter, 2010) and scaled and merged using AIMLESS (Evans, 2011; Evans and Murshudov, 2013).

A model of PR3 (PDB ID 1FUJ) as well as a partial model of CD177$^{subI}$ were used as search models for molecular replacement using PHASER (McCoy et al, 2007). The solution consisted of one molecule in the asymmetric unit and was rebuilt using Buccaneer (Cowtan, 2006). All-atom refinement with Refmac5 (Murshudov et al, 2011) (version 5.8.0430) resulted in a full model containing

residues I28-K253 and L22-C203 for PR3 and CD177$^{subI}$, respectively.

## Size-exclusion chromatography and multiangle light scattering

In total, 110 μl of protein at 1 mg/mL was applied onto a Superdex® 200 HR 10/30 column previously equilibrated with SEC buffer. The system was connected to a Wyatt Dawn® HELEOS-II 8-angle light scattering detector as well as a Wyatt Optilab® rEX refractive index monitor.

## Plasma samples

Study participants: Patients with GPA and MPA were identified from our clinical vasculitis database, with the disease classified according to the Chapel Hill Consensus Conference diagnostic criteria. Patient demographics, clinical characteristics and investigations, including ANCA reactivity, were documented from electronic records. Disease activity, scored by the Birmingham Vasculitis Activity Score (BVAS) and the Vasculitis Damage Index (VDI), was calculated at the time of sample collection. Frozen plasma from a healthy donor was purchased from Cambridge Bioscience.

## IgG purification from patients' plasma samples

Human anti-PR3 ANCA IgGs from the plasma exchange fluid of patients with PR3-ANCA were purified using Protein G agarose beads (Thermo Fisher) according to the manufacturer's instructions. Briefly, thawed plasma was incubated with Protein G beads overnight at 4 °C on a rotating agitator. Beads were washed with 10 column volumes of DPBS. Bound IgG was eluted with 0.1 M glycine-HCl, pH 2.8 in 1.5-mL reaction tubes containing neutralising solution (2 M Tris-HCl pH 7.5). The resulting IgG solution was dialysed using a dialysis cassette against PBS for 24-48 h at 4 °C before being applied to a 1 ml Detoxi-Gel™ endotoxin removing column (Thermo Fisher Scientific™) as per the manufacturer's instructions. The final amount of endotoxin-depleted IgG was determined using a NanoDrop spectrophotometer using the IgG setting.

## Enzyme-linked immunosorbent assays

Nunc MaxiSorpTM flat-bottom plates (Thermo Fisher Scientific) were coated with 2 μg/mL of antigen in Dulbecco's PBS (DPBS) overnight at 4 °C. Neutrophil PR3 was purchased from Athens Research and Technology. Plates were washed three times with PBS supplemented with 0.05% Tween-20 (PBST) and blocked with 1% casein in PBS buffer (Thermo Fisher Scientific) at room temperature for 30–60 min in order to reduce non-specific interactions.

To set-up the standardised PR3$^{rec}$ and PR3$^{nonCD177}$, we took advantage of both PR3 constructs possessing a His$_6$ tag and used human recombinant monoclonal 6X His tag® antibody (Abcam) as a reference to standardise and compare responses obtained for different PR3 constructs. The standard curve was generated from a twofold serial dilution of the human anti-His starting from a 1 mg/mL stock (1:1000–1:102,400). Each standard dilution was added in duplicate to the plate. For the standardised ELISA using plasma, patients' plasma samples were diluted to 1:800 and 1:1600,

and were added to the plate in triplicate. When using IgG samples, the samples were diluted to 10 or 5 µg/mL and added to the plate in triplicate. After the addition of primary antibody (samples + standard) plates were incubated overnight at 4 °C and then washed as before. Goat anti-human IgG (γ-chain specific) conjugated to alkaline phosphatase (Sigma-Aldrich) at 1:1000 dilution was added for 30 min at room temperature. Following a final wash, plates were developed by using 4-nitrophenylphosphate (Sigma-Aldrich) at 1 mg/mL in diethanolamine buffer as substrate. $OD_{405nm}$ was read on an ELx800 (Biotek) and SpectraMax M3 (Molecular Devices) microplate readers and data were analysed with Gen5.3.11 ELISA software. Each plate was assessed for its quality and had to pass the following parameters: (i) the $R^2$ value for the standard curve is >0.995, (ii) the average blank $OD_{405nm}$ is <0.15. Additionally, each sample was assessed for their quality and had to pass the following parameters: (i) No sample displays an $OD_{405nm}$ value > 1.2 at the timepoint selected for analysis, (ii) the coefficient of variability (CV %) is <20%. IgG sample responses were adjusted by the experiment dilution factor and then normalised by their purification dilution factor. Each patient's sample was independently measured at least three times. Plasma and IgG samples from a patient containing MPO-ANCAs were included as negative controls.

For the $PR3^{rec}+CD177^{ecto}$ and $PR3^{rec}+CD177^{subI}$ standardised ELISA, a $CD177^{ecto/subI}$ construct without a $His_6$ tag was used so that the standard curve would always be derived from $PR3$-$His_6$ binding. $PR3^{rec}$ and $CD177^{ecto/subI}$ (without $His_6$ tag) were mixed at 1:1 molar ratio before coating the plate. Concentration was measured before coating by calculating the average extinction coefficient for the complex.

For endpoint ELISAs, human recombinant anti-His mAb (Abcam) and Strep mAb (Iba) were used at 0.1 µg/mL. Goat anti-mouse/human IgG (Fc specific) conjugated to alkaline phosphatase (Sigma-Aldrich) at 1:1000 was used as a secondary antibody to detect bound mAb.

For the commercial ANCA ELISA, a cANCA proteinase ELISA kit was purchased from Euroimmun (EA 1201-9601-2 G). Plasma and purified IgG samples were analysed following the manufacturer's protocols.

## Statistical analysis

All results were analysed using GraphPad Prism (version 10.4.1) and expressed as means ± standard error of the mean (SEM) with the number of experimental replicates (*n*) provided. Differences between groups were determined by one-way or two-way ANOVA, or *t* test with appropriate post hoc tests as indicated. In all cases, a probability (*P*) value of <0.05 was accepted to reject the null hypothesis and considered significant.

## Ethics approval

This study involves human participants and was approved by the NHS Research Ethics Committee (05/Q0508/6 and 21/SC/0355) and was in accordance with the Declaration of Helsinki.

## Data availability

Structural data has been deposited at the RCSB Protein Data Bank (PDB). Accession numbers 9IGP and 9IGO.

The source data of this paper are collected in the following database record: biostudies:S-SCDT-10_1038-S44319-026-00716-5.

## Peer review information

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

## Acknowledgements

This work was carried out with the support of Diamond Light Source, instrument beamlines I23 and I03 (proposal mx18069) and SOLEIL, instrument beamline PROXIMA-1. CZG was supported by the Wellcome Trust DPhil in Structural Biology programme and the Oxford Cephalosporin Fund. Research in the ES lab was supported by the Wellcome Trust (202827/Z/16/Z and 226647/Z/22/Z) and the EMBO Young Investigator Programme. Research in the HHL lab was supported by Chang Gung Memorial Hospital-Linkou, Taiwan (CMRPD1M0033) and the National Science and Technology Council (NSTC), Taiwan (NSTC-113-2918-I-182-001 and NSTC-113-2320- B-182-009).

## Author contributions

**Céline Zheng-Gérard**: Conceptualisation; Resources; Data curation; Software; Formal analysis; Validation; Investigation; Visualisation; Methodology; Writing—original draft; Project administration; Writing—review and editing. **Jana Joha**: Conceptualisation; Resources; Data curation; Formal analysis; Validation; Investigation; Visualisation; Methodology; Writing—original draft; Project administration; Writing—review and editing. **Maria Carrasquero**: Resources; Data curation; Formal analysis; Supervision; Validation; Investigation;

Visualisation; Methodology; Writing—review and editing. **Kamel El Omari**: Resources; Formal analysis; Validation; Investigation; Methodology. **Edward Lowe**: Software; Supervision; Validation; Investigation; Methodology. **Shirish Dubey**: Data curation; Supervision; Validation; Writing—review and editing. **Simon J Draper**: Resources; Supervision; Methodology; Project administration. **Yu-Chi Chang**: Validation; Investigation; Visualisation; Methodology. **Hsi-Hsien Lin**: Conceptualisation; Resources; Data curation; Formal analysis; Supervision; Funding acquisition; Validation; Investigation; Visualisation; Methodology; Writing—review and editing. **Alan D Salama**: Conceptualisation; Resources; Supervision; Methodology; Writing—review and editing. **Kirsty McHugh**: Conceptualisation; Data curation; Formal analysis; Supervision; Validation; Investigation; Visualisation; Methodology; Writing—review and editing. **Elena Seiradake**: Conceptualisation; Resources; Data curation; Formal analysis; Supervision; Funding acquisition; Validation; Investigation; Visualisation; Methodology; Writing—original draft; Project administration; Writing—review and editing.

Source data underlying figure panels in this paper may have individual authorship assigned. Where available, figure panel/source data authorship is listed in the following database record: biostudies:S-SCDT-10_1038-S44319-026-00716-5.

## Disclosure and competing interests statement

The authors declare no competing interests.

# Expanded View Figures

**Figure EV1.  Structural and biochemical analysis.**

(**A**) Sequence alignment of LU1-4 domains of CD177 (D1-D4) and LU domains from other proteins as indicated. The positions of predicted β-strands and disulphide bridges are indicated. (**B**) Plots of sulphur anomalous signal metrics as a function of resolution obtained from XSCALE and SHELXC softwares. SigAno represents the estimated anomalous signal strength while <d″/sig> shows the normalised difference signal. See Dataset EV3. (**C**) Structural superposition of the two copies of CD177ecto found in the asymmetric unit of the crystal, shown in grey and cyan. (**D**) Sequence alignment of different PR3 orthologs of the hydrophobic (blue) and positively charged (orange) residues. (**E**) Cartoon representation of PR3 (PDB ID: 1FUJ). The catalytic triad, hydrophobic and positively charged residues in the hydrophobic surface patch are indicated by purple, blue and orange spheres, respectively. (**F**) Close-up view of mutated residues in the hydrophobic surface patch. (**G**) Data from recombinant PR3rec protein purification from HEK293 cultures. The Size-Exclusion Chromatography (SEC) purification profile is shown as well as its corresponding Coomassie-stained SDS-PAGE gel (top) and anti-His western blot (bottom) with the eluted fractions. The main protein peak is indicated by a red asterisk. Void volume fractions are indicated in grey. Impurity fractions are indicated by yellow, orange and red arrowheads. Presence of PR3 (~ 30 kDa) in the analysed samples is indicated by a black arrow. (**H**) SDS-PAGE gel of glycosylated PR3rec samples produced by HEK293T and N-acetylglucosaminyltransferase I-deficient HEK293S cells displaying the difference in glycosylation pattern and size between the two cell lines (lanes 1 and 2). Samples obtained after deglycosylation by the endoglycosidases Endo-F1 (lane 3) or PNgase F (lanes 4 and 5) confirm that PR3rec is indeed glycosylated. (**I**) SEC-binding profile and corresponding SDS-PAGE gel of PR3rec with CD177ecto. PR3rec and CD177ecto profiles are shown in teal and grey, respectively. The profile of a PR3rec-CD177ecto protein mixture in a 2:1 molar ratio is shown in purple. The protein peaks are indicated by a purple asterisk. Source data are available online for this figure.

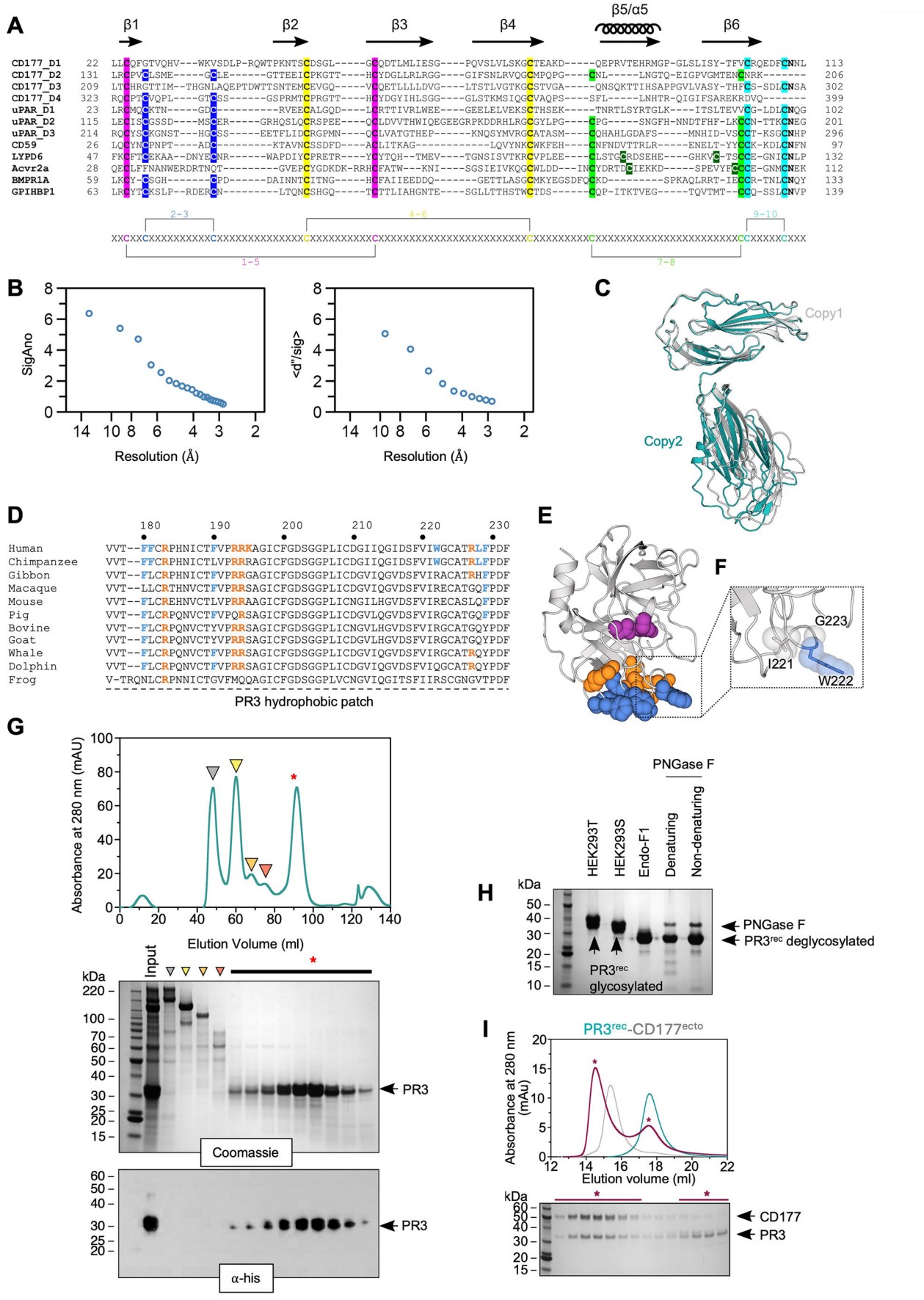

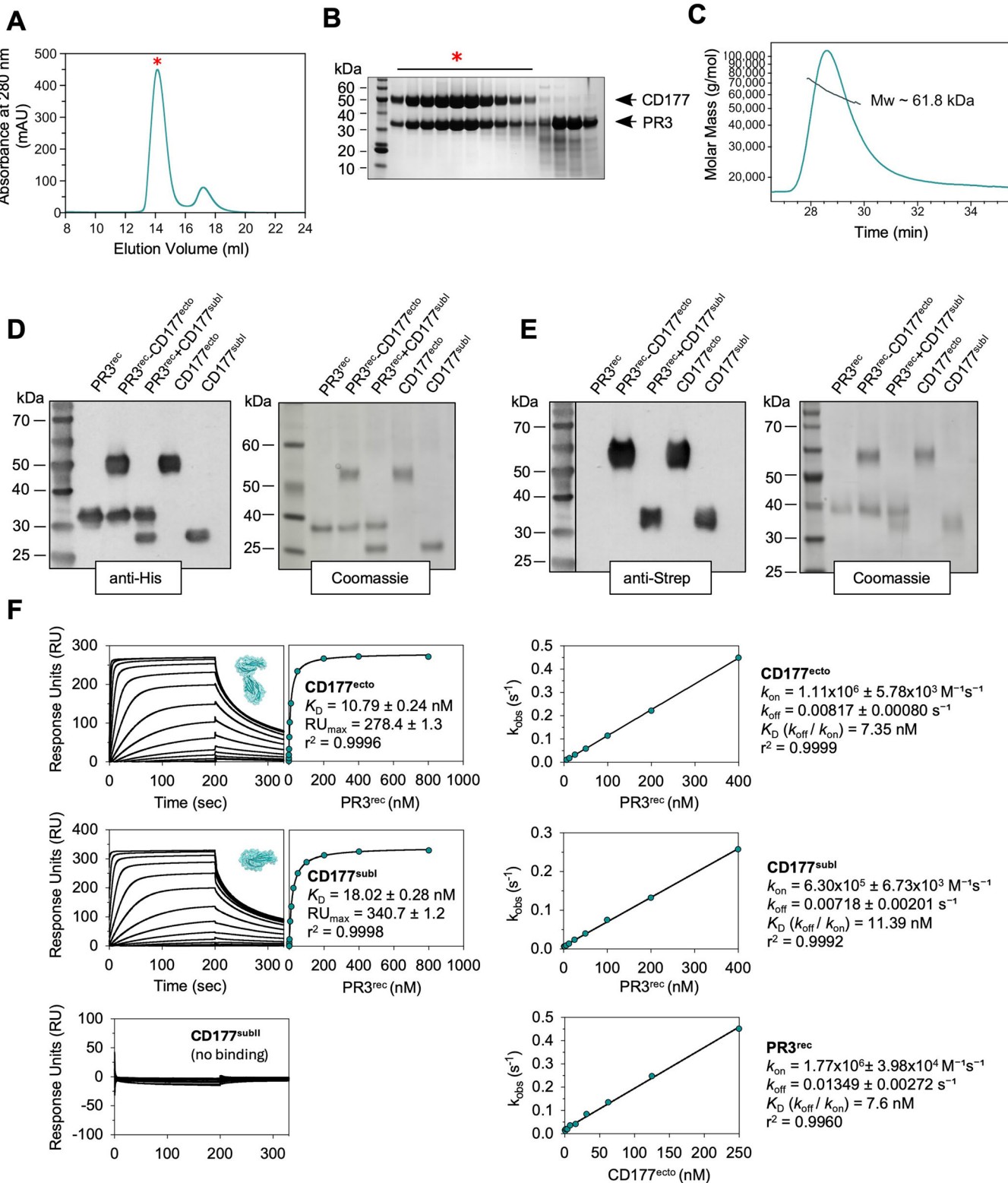

◀  **Figure EV2.  Purification and biophysical characterisation of PR3$^{rec}$-CD177$^{ecto}$ complex.**

(A) SEC profile of purified PR3$^{rec}$-CD177$^{ecto}$ complex. The main protein peak is indicated by a red asterisk. (B) Coomassie-stained SDS-PAGE gel showing fractions eluted from SEC (experiment in (A)). (C) Multiangle light scattering experiment of PR3$^{rec}$-CD177$^{ecto}$ confirms a 1:1 interaction in solution, with the expected size of ~60 kDa for the complex. (D) anti-His western blot (left) and reducing Coomassie-stained SDS-PAGE gel (right) of His-tagged PR3 and CD177 constructs. Same sample preps were used for both experiments. The PR3$^{rec}$-CD177$^{ecto}$ sample comes from the SEC-purified complex peak as shown in panel A. The PR3$^{rec}$+CD177$^{subl}$ sample comes from mixing the two proteins together in a 1:1 molar ratio. (E) anti-Strep western blot (left) and reducing Coomassie-stained SDS-PAGE gel (right) of His-tagged PR3$^{rec}$ and TwinStrep-tagged CD177 constructs. Same sample preps were used for both gels. A different batch of PR3$^{rec}$ was used for these blots. (F) SPR data (left panels) and corresponding fitted curves (centre and right panels) show the binding of Avi-tagged CD177$^{ecto}$, CD177$^{subl}$ and CD177$^{subll}$ ligands (0.05 to 800 nM) to PR3$^{rec}$ analyte, and of Avi-tagged PR3$^{rec}$ ligand (0.06 to 1000 nM) to CD177$^{ecto}$ analyte (see sensorgrams in Fig. 3B). The immobilised levels were 560, 340, 320 and 350 resonance units (RU) respectively. For PR3$^{rec}$ binding to CD177$^{ecto}$ and CD177$^{subl}$ ligand, $K_D$ were derived using both 1:1 equilibrium analysis (centre panel) and kinetic analysis (right panel). For CD177$^{ecto}$ binding to PR3$^{rec}$ ligand, kinetic analysis was performed with a 1:1 Langmuir model at concentrations below 16 nM, where binding curves reached clear plateau. At higher concentrations, an alternative model accounting for mass transport limitations was used. Final $K_D$ values were determined from $k_{ON}$ and $k_{OFF}$ rates obtained independently from fits to association and dissociation phases. For SPR data, please refer to Dataset EV2. Source data are available online for this figure.

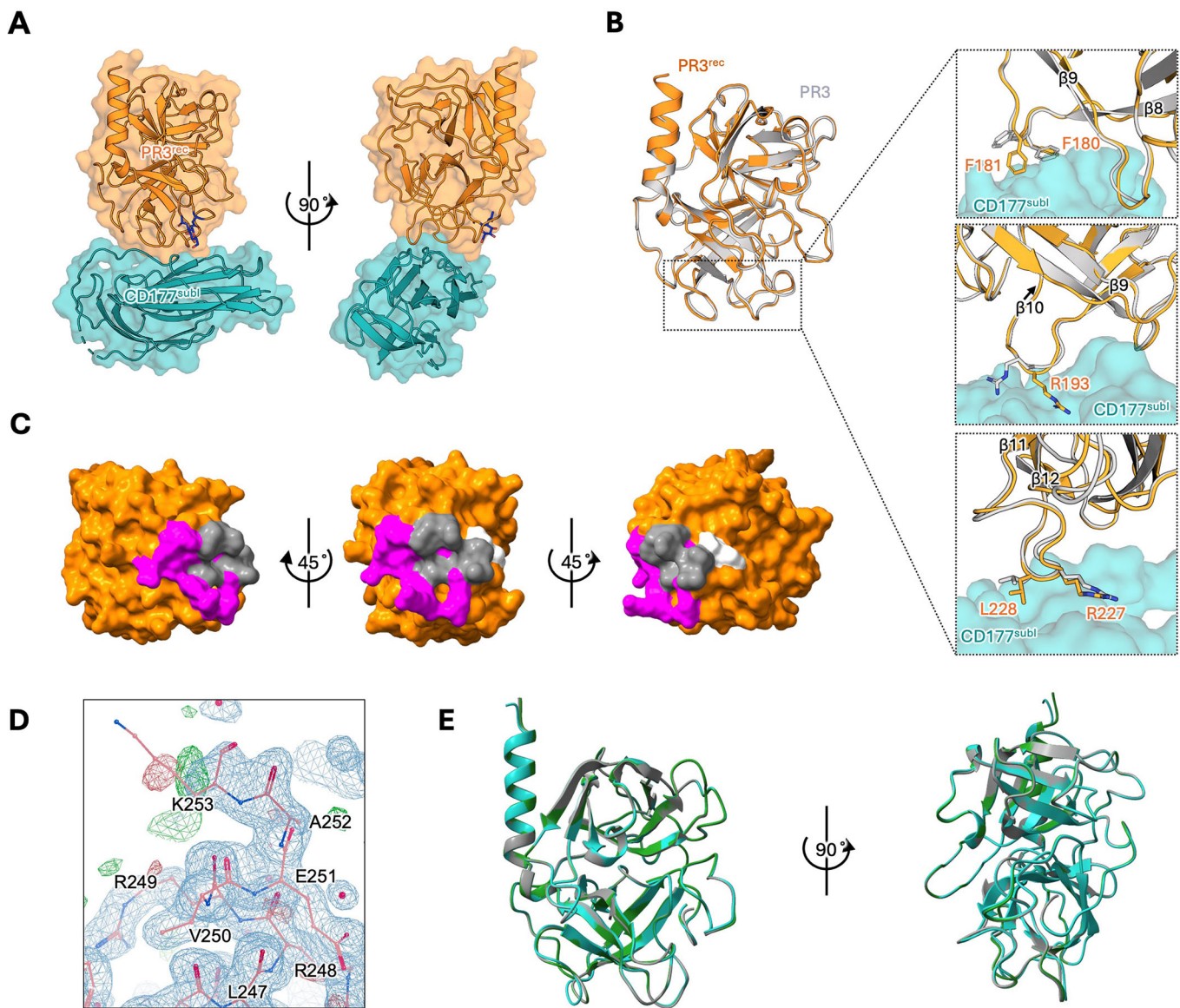

**Figure EV3. Structural views of PR3^rec and PR3^rec-CD177^subl.**

(A) Views of the PR3^rec-CD177^subl complex structure. (B) Superposition of the PR3^rec structure as found in complex with CD177^subl, and the previously published structure of PR3 (PDB: 1FUJ). Insets display close-ups on diverging side chains between PR3^rec and 1FUJ in the CD177-binding loops. (C) Views of PR3^rec with the CD177-binding region highlighted. Residues H148, G149, Q151, T176, V178, P192, R193, R227 and P230 are highlighted in magenta, residues I221N and W222G are highlighted in white and residues that overlap with the hydrophobic patch and CD177-binding region (F180, F181, F190, L228 and F229) are highlighted in dark grey. (D) The electron density maps calculated from data collected for the complex in panel A (2Fo-Fc at 1 sigma level in blue, Fo-Fc at −/+ 3 sigma level in green/red, respectively. The view is focused on PR3 K253. Protein model shown as sticks. (E) Superposition of AlphaFold predicted nPR3, PR3^rec and PR3^nonCD177 structures. Grey: nPR3, Green: PR3^rec, Teal: PR3^nonCD177.

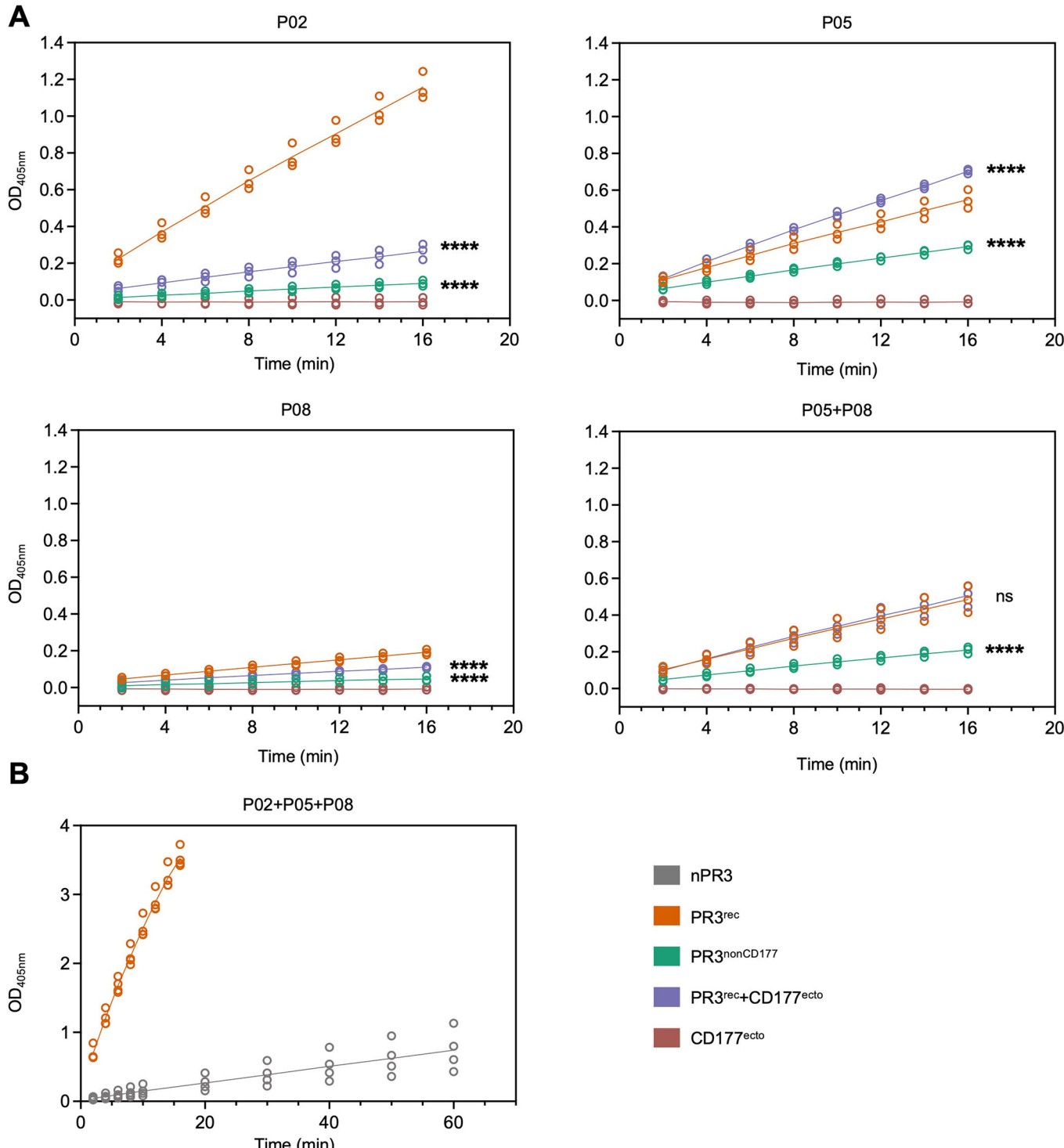

**Figure EV4.  Time-course ELISAs for individual plasma and plasma pool.**

(A) Equal amounts of PR3$^{rec}$, PR3$^{nonCD177}$, PR3$^{rec}$+CD177$^{ecto}$ and CD177$^{ecto}$ were coated on ELISA plates for time course measurements using individual and mixed GPA patient plasma at a 1:1600 dilution. Simple linear regression analysis was carried out. Binding of PR3$^{nonCD177}$ and PR3$^{rec}$+CD177$^{ecto}$ were compared to PR3$^{rec}$ binding. $n = 3$. (B) Equal amounts of nPR3 and PR3$^{rec}$ were coated on ELISA plates for time course measurements using GPA patient plasma mixed in equal amounts (P02 + P05 + P08) at a 1:50 dilution. The results suggest that PR3$^{rec}$ offers faster detection of robust signal compared to nPR3. $n = 3$. *$P \leq 0.05$, **$P \leq 0.01$, ***$P \leq 0.001$, ****$P \leq 0.0001$. Data are presented as mean ± S.E.M. For exact $P$ values, please refer to Dataset EV1. Source data are available online for this figure.

                                                    

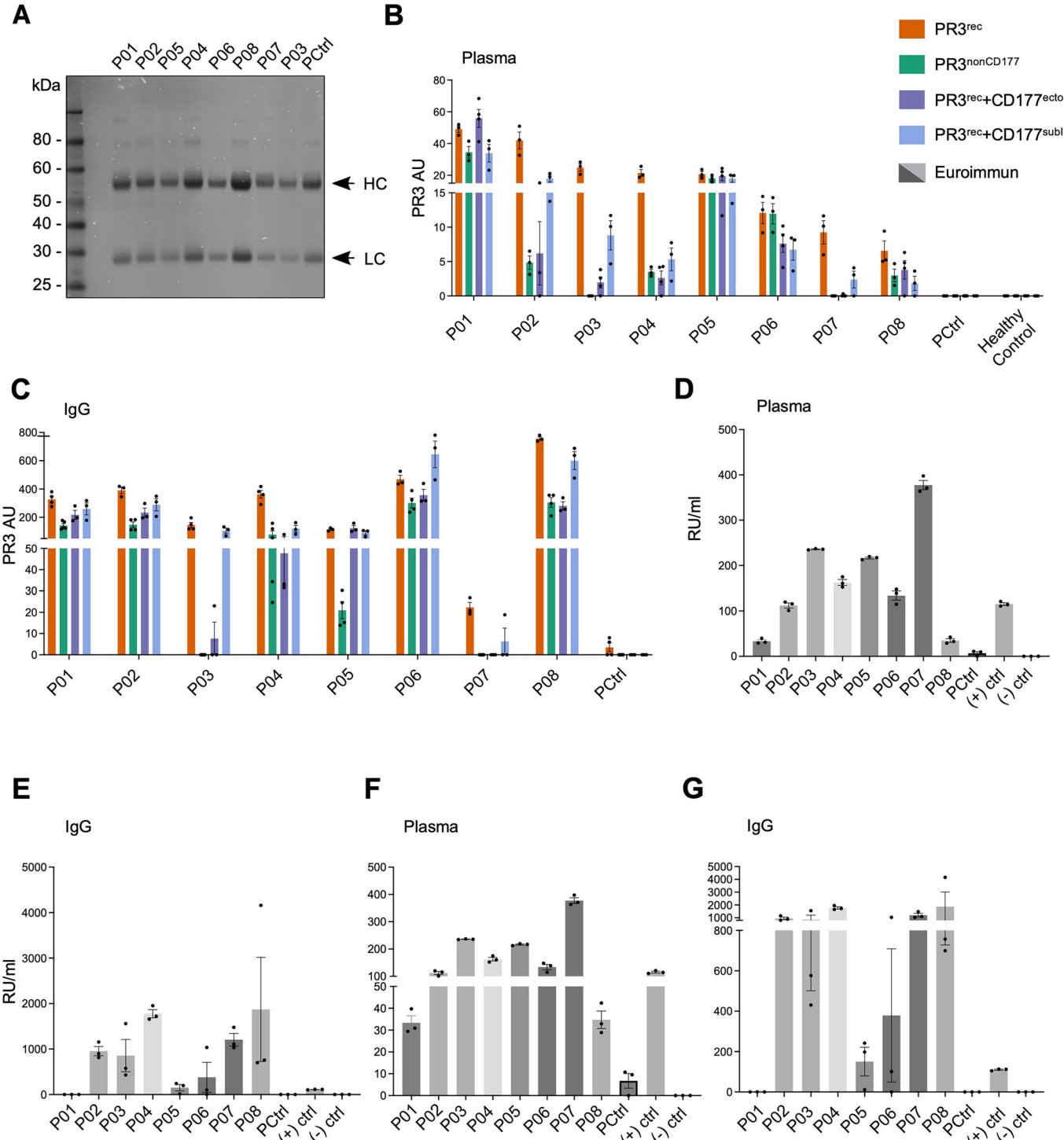

**Figure EV5. IgG preparations from patient plasma.**

(A) Reducing Coomassie-stained SDS-PAGE gel of patient purified IgG samples. Heavy chain (HC) and light chain (LC) bands are shown. Samples were loaded at 5 µg per well. (B, C) Zoomed in view of standardised ELISA data using plasma and IgG samples from PR3-ANCA positive GPA patients (P01-P08) and a MPO-ANCA positive patient (PCtrl), and plasma from a healthy control plasma sample. ANCA binding is reported in arbitrary units (PR3 AU). IgG samples were tested at 10 or 5 µg/mL. Responses were adjusted by the experiment dilution factor and then normalised by their purification dilution factor. $n = 3/4$. (D, E) Standardised ELISA data from a commercially available ELISA kit (Euroimmun) using the same patient samples. Binding is reported as response units per ml (RU/mL). (+) and (−) ctrl refer to the positive and negative controls provided with the kit. IgG samples were tested at 5 µg/mL and their purification dilution factor was used to normalise responses. $n = 3$. (F, G) Zoomed in views of (D, E). A minimum of three technical repeats were done for each ligand and sample. Source data are available online for this figure. $n = 3$. *$P \leq 0.05$, **$P \leq 0.01$, ***$P \leq 0.001$, ****$P \leq 0.0001$. Data are presented as mean ± S.E.M. For exact $P$ values, please refer to Dataset EV1. Source data are available online for this figure.

