## [Peer Review File · EMBO Reports]

Structures of proteinase 3 and the CD177 receptor complex reveal a major autoantibody epitope

Céline Zheng-Gérard, Jana Joha, Maria Carrasquero, Kamel El Omari, Edward Lowe, Shirish Dubey, Simon Draper, Yu-Chi Chang, Hsi-Hsien Lin, Alan Salama, Kirsty McHugh, and Elena Seiradake

Corresponding author(s): Elena Seiradake (elena.seiradake@bioch.ox.ac.uk) , Hsi-Hsien Lin (hhlin@mail.cgu.edu.tw), Kirsty McHugh (kirsty.mchugh@paediatrics.ox.ac.uk), Alan Salama (a.salama@ucl.ac.uk)

Review Timeline:

Submission Date:	1st Apr 25
Editorial Decision:	26th May 25
Revision Received:	30th Sep 25
Editorial Decision:	16th Dec 25
Revision Received:	13th Jan 26
Accepted:	30th Jan 26

Editor: Achim Breiling

Transaction Report:

Dear Dr. Seiradake,

I have already forwarded to you the reports I received from the three referees that I asked to evaluate your study. Please find them again below. After going through your preliminary point-by-points response (revision plan), and after getting feedback from the referees during cross-commenting, I have decided to proceed with the manuscript.

Given the constructive referee comments, I would thus like to invite you to revise your manuscript with the understanding that the concerns of the referees must be addressed in the revised manuscript and/or in a detailed point-by-point response, as indicated in your revision plan. Acceptance of your manuscript will depend on a positive outcome of a second round of review. It is EMBO reports policy to allow a single round of revision only and acceptance of the manuscript will therefore depend on the completeness of your responses included in the next, final version of the manuscript.

1) a .docx formatted version of the final manuscript text (including legends for main figures, EV figures and tables), but without the figures included. Figure legends should be compiled at the end of the manuscript text.

2) individual production quality figure files as .eps, .tif, .jpg (one file per figure), of main figures and EV figures. Please upload these as separate, individual files upon re-submission.

4) a complete author checklist, which you can download from our author guidelines

(<https://www.embopress.org/page/journal/14693178/authorguide>). Please insert page numbers in the checklist to indicate where the requested information can be found in the manuscript. The completed author checklist will also be part of the RPF.

5) that primary datasets produced in this study (e.g. RNA-seq, ChIP-seq, structural and array data) are deposited in an appropriate public database. If no primary datasets have been deposited, please also state this in a dedicated section (e.g. 'No primary datasets have been generated and deposited'), see below.

The accession numbers and database should be listed in a formal "Data Availability" section that follows the model below. This is now mandatory (like the COI statement). Please note that the Data Availability Section is restricted to new primary data that are part of this study. This section is mandatory. As indicated above, if no primary datasets have been deposited, please state this in this section

Data availability

6) We now request the publication of original source data with the aim of making primary data more accessible and transparent to the reader. You will receive a separate email with instructions for providing source data with your revised manuscript, including information how to upload and organize the files.

8) Regarding data quantification and statistics, please make sure that the number "n" for how many independent experiments were performed, their nature (biological versus technical replicates), the bars and error bars (e.g. SEM, SD) and the test used to calculate p-values is indicated in the respective figure legends (also for EV and Appendix figures). Please also check that all the p-values are explained in the legend, and that these fit to those shown in the figure. Please provide statistical testing where applicable. Please avoid the phrase 'independent experiment', but clearly state if these were biological or technical replicates. Please also indicate (e.g. with n.s.) if testing was performed, but the differences are not significant. In case n=2, please show the data as separate datapoints without error bars and statistics. See also:
<http://www.embopress.org/page/journal/14693178/authorguide#statisticalanalysis>

9) Please add scale bars of similar style and thickness to all microscopic images, using clearly visible black or white bars (depending on the background). Please place these in the lower right corner of the images themselves. Please do not write on or near the bars in the image but define the size in the respective figure legend.

10) Please note our reference format:

12) We now use CRediT to specify the contributions of each author in the journal submission system. CRediT replaces the author contribution section. Please use the free text box to provide more detailed descriptions and do NOT provide your final manuscript text file with an author contributions section. See also our guide to authors:
<https://www.embopress.org/page/journal/14693178/authorguide#authorshippinguidelines>

13) All Materials and Methods need to be described in the main text using our 'Structured Methods' format, which is required for all research articles. According to this format, the Methods section should include a Reagents and Tools Table (listing key reagents, experimental models, software, and relevant equipment and including their sources and relevant identifiers), uploaded as separate file, and a Methods section in which we encourage the authors to describe their methods using a step-by-step

protocol format with bullet points, to facilitate the adoption of the methodologies across labs. More information on how to adhere to this format as well as downloadable templates (.doc) for the Reagents and Tools Table can be found in our author guidelines (section 'Structured Methods'):

14) Please add up to five keywords to the manuscript text file and order the sections like this, using these names: Title page - Abstract - Keywords - Introduction - Results - Discussion - Methods - Data availability section - Acknowledgements - Disclosure and Competing Interests Statement - References - Figure legends - Expanded View Figure legends

15) Please make sure that all the funding information is also entered into the online submission system and that it is complete and similar to the one in the acknowledgement section of the manuscript text file.

Please note that corresponding authors are required to supply an ORCID ID upon submission of a revised manuscript and an institutional e-mail address (the latter is missing only for author Hsi-Hsien Lin. Please find instructions on how to link the ORCID ID to the account in our manuscript tracking system in our Author guidelines:

<http://www.embopress.org/page/journal/14693178/authorguide#authorshipguidelines>

Please note that corresponding authors are required to supply an ORCID ID upon submission of a revised manuscript and an institutional e-mail address (the latter is missing only for author Hsi-Hsien Lin. Please find instructions on how to link the ORCID ID to the account in our manuscript tracking system in our Author guidelines:

<http://www.embopress.org/page/journal/14693178/authorguide#authorshipguidelines>

I look forward to seeing a revised form of your manuscript when it is ready.

Yours sincerely,

Referee #1:

This manuscript by Zheng-Gérard et al. reports an integrated study spearheaded by structural studies of the complex of CD177 with Proteinase Pr3 to provide insights into the binding epitope targeted by autoantibodies in GPA. This study provides novel insights based on high quality data and new molecular tools that will help an important field of biomedical research move forward. I herewith provide input for the authors' consideration towards improving their manuscript.

The title of the manuscript does not quite capture fully the message of the work. For instance, one might expect to see reference to the reported complex of Pr3 and the CD177 receptor.

The authors need to discuss prior work on anti-Pr3 antibodies, for instance as summarized by Peen Ralph Arthritis Res. 2000 Jun 12;2(4):255-259. doi: 10.1186/ar97, as well as other contributions.

Pr3 is known to exist in a membrane-bound form, albeit in a small fraction, which also has an altered activity profile due to conformational differences compared to soluble Pr3. For instance, work by Guarino et al. J Biol Chem 2023 Feb 26;299(4):103072. doi: 10.1016/j.jbc.2023.103072

Despite presenting several high quality SPR data, with well-fitted sensorgrams (presumably to 1:1 Langmuir models), the authors do not provide the kinetics of binding (k_a and k_d). This will need to be reported and discussed, not only to complete the data presentation, but mainly because on/off rates are often very relevant in comparatively understanding antibody binding

profiles.

How do the authors think the introduced N-linked glycosylation on Pr3 might affect binding?

It would be very instructive and useful to readers and future adopters of the approach to include in Suppl. Fig. 1 an SDS-PAGE analysis of PNGase-treated protein samples to illustrate the presence of introduced N-glycans.

What are the authors' thoughts on why the apparent Pr3 hydrophobicity is there in the first place? Any implications for the binding of other proteins, antibodies etc. Please discuss accordingly.

+++++

Technical input:

+++++

Please make sure that the correct nomenclature for binding constants is used throughout the manuscript, i.e. capital K in italics and subscript D in non-italics for the KD, and likewise for the eventual reports of ka and ks (see above), k in italics and a/d as non-italic subscripts.

Table S1 and Table S2: The R-merge reported is an obsolete (in fact erroneous to a large extent) crystallographic metric and this reviewer would strongly discourage its use given that Rmeas/Rpim and CC1/2 have been appropriately included.

Table S1 and Table S2: In addition to the average B factor reported for protein atoms, please report average B factors (or Atomic Displacement Parameters) for protein, solvent, heteroatoms etc, as well.

Table S1. Please report the anomalous differences for the S-SAD data set of CD177ecto.

Referee #2:

17.04.2025

Review for EMBOR-2025-61658V1 „Structures of proteinase 3 and the CD177 receptor reveal major autoantibody epitope"

The manuscript (MS) by Zheng-Gérard, et al. reports the structure of the neutrophil PR3:CD177 complex as well as that of CD177 alone, thus providing two major pieces of data missing from the thorough characterization of the complex. This is a most welcome advance for the field and it is wonderful to finally see these structures that, together with the extensive biochemical data already in the literature, will be of great value in improving our understanding of neutrophil biology and for the development of treatments for PR3-ANCA diseases. It is therefore most unfortunate that a careful analysis of the structural data in light of the ample biochemical information already published is *absent from the text*. The MS is furthermore marred by examples of sweeping generalization that are neither necessary nor warranted, as well as statements that are quite misleadingly formulated or altogether inaccurate. Most peculiar indeed is the reporting of a mutagenesis scheme for preparing a monomeric PR3 - an effort occupying substantial space in the MS and presented as novel - *although the "monomerization" of PR3 based on mutating the same positions had already been reported elsewhere*; that study is actually referenced by the authors of the present MS but no discussion of that PR3 data is presented in the text. The inaccuracies, contradictions and omissions lead one to wonder whether the authors actually read many of the studies that they cite in their reference list; reading the text does not convey the impression that the authors are aware of data published by many groups over the last several decades. Despite the welcome advance provided by the structural data, this MS is **not fit for publication anywhere** in its present form.

A discussion of the major flaws in the MS follows here and all of these issues must be adequately addressed before the MS can be submitted to any journal with a reasonable expectation of acceptance.

1) We are long past the days when the generation of a structure alone, however competently done (as the current example appears to be), suffices for a publication (unless combined with some new technical advance or other novelty that was necessary for achieving the structure). It is here emphasized that it is very exciting to now have these structures and their thoughtful analysis will be valuable for understanding the biology of ANCA-related diseases and also for developing treatments. But just such a thoughtful analysis - with confirmatory experimental data - is, unfortunately, missing from the MS. A number of experiments with the complex are reported but there isn't any meaningful analysis in light of previously published predictions - for example, Kantari et al. (reference 31 in the MS) made the "hydrophobic patch" responsible for the attachment of PR3 to the outer neutrophil membrane and Korkmaz, et al. (reference 81 in the MS) claimed that the "hydrophobic patch" was responsible for the interaction with CD177 - does the structural data (which is what is new) provide any insight into the correctness of these suppositions?

A rather detailed model for the complex based on biochemical data was proposed in 2017 (Jerke, et al. - reference number 19 in the MS) - how does the presently reported complex structure compare with those predictions from the biochemical characterization? These examples - and more - should make up a substantial portion of the discussion section of the current manuscript but are entirely absent (although the primary sources are in the reference list). The structures alone are the only new information.

2) The authors give substantial space to their mutagenesis effort to achieve a monodisperse PR3 construct and report finding

mutations that render the enzyme monomeric. Quoting from their text: **"The study was possible using a site-directed mutagenesis screen to identify stable Pr3 variants that are readily expressed and purified from HEK293 cultures."**

Perhaps it is an oversight (but then a very serious one) that a successful monomerization effort *was already published in 2017* (reference 19 in the MS): the mutated Ile - Trp (i - i+1) pair was shown to render PR3 monomeric and the affinity of that monomer for CD177 by SPR was also reported. The data in the current MS are therefore not new as the authors claim (using another residue numbering scheme does not alter the sequence itself) and this previous work is not acknowledged in their text (although the paper in question does appear in the reference list). This reviewer will stop short of labeling this an ethical problem, but the "oversight" must be corrected no matter where the next version of the MS is to be published.

3) This reviewer feels that a comment on style here is also necessary: apparently also gone are the days when formulations in scientific reports minimized use of the first person; while this can sometimes not be gracefully avoided, the practice of introducing basic (if not trivial) results in this MS with overly self-promoting phrases such as "We pioneered..." (page 11), as well as numerous other "We" statements scattered throughout, should be and, considering that much of what is described in the MS was already reported by others and/or not sufficiently acknowledged as such, maximized the discomfort felt when reading the text. These deficiencies, as well as the grammar mistakes and confusing formulations (some, but not all, of which are addressed explicitly below) must be corrected before the next submission, to whichever journal that may be.

4) There are a number of statements throughout the text that are either partly or wholly inaccurate; some examples follow (this list is NOT exhaustive), with comments from this reviewer in **bold**:

From the Abstract, page 1:

"Granulomatosis with polyangiitis is a life-threatening systemic vasculitis, characterised by antineutrophil cytoplasmic autoantibodies (ANCA) against proteinase 3 (Pr3) expressed on the surface of neutrophils. Most cell surface Pr3 is bound to the receptor CD177..."

These first two sentences of the paper are quite garbled - the first reads as if the ANCA themselves are "expressed" on neutrophils...which of course is not the case; furthermore, PR3 is found on the extracellular surface of all neutrophils, even those not displaying CD177...and even on neutrophils from persons who are *genetically negative for CD177*, i.e. whose neutrophils completely lack CD177.

From the Abstract, page 1:

"We describe an extensive, mainly hydrophobic binding interface between Pr3 and CD177..."

This is an unnecessarily and incorrectly qualified statement - the authors report the buried surface area in the complex to be " ~655 Å² " - a value rather at the low end of the buried surface area spectrum that extends, on average, between 1500 and 3000 Å² (see for example 10.1073/pnas.93.1.13); How have the authors decided that the PR3:CD177 interaction is "extensive"?

From the Abstract, page 2:

"Using a panel of Pr3-ANCA positive patient samples, we show that CD177 inhibits Pr3-ANCA binding to Pr3".

It is obvious that this statement on its face cannot be accurate. Certainly the binding of some ANCAs is blocked by the presence of CD177 (as the authors show with their experiments using patient samples) but, as reported later in the MS and has been shown previously in multiple studies (see for example J Immunol. 1994 May 1;152(9):4722-37, J Autoimmun 2010 35, 299-308 (reference 78 in the MS) and also reference 20 in the MS, among others) ANCA epitopes are also found outside of the CD177 binding site and some ANCAs can bind to PR3 that is complexed with CD177.

The authors also write on page 5: "We discover that most Pr3-ANCA positive samples compete with the interaction between Pr3 and CD177..." and furthermore on page 9: "Most GPA patient samples harbour autoantibodies that target the CD177-binding site of Pr3" because, in their ELISA, 7 of 8 samples appear to bind an epitope coinciding with the CD177 binding site (their conclusion). Now, even accepting the authors' own - not referenced!! - "estimate" of the prevalence of GPA (Introduction, page 2) of "2 - 150 per million individuals", it is clear that these statements cannot be true. Taking the lower limit of this - again not referenced - estimate (and since PR3-ANCA are involved in other diseases and the number of *undiagnosed* cases of PR3-ANCA diseases is thought to be substantial, the authors' estimate is clearly extremely low), the number of patients must number in the hundreds of thousands/millions, depending on where the authors' "estimate" applies (in England? In Taiwan?? Worldwide?). To have tested samples from <0,001% of all patients (based on the absolute - and incorrect - minimum estimate) and then drawing a firm conclusion about them *all* cannot be taken seriously.

From the Introduction, page 3:

"...although some PR3 appears to bind directly to cell membrane components".

This is a perplexingly misleading statement- the word "appears" here is simply inaccurate; there is ample published evidence, over many years, demonstrating that PR3 is present on the neutrophil cell surface - even CD177 negative individuals have mPR3 on their neutrophils. This surface interaction has also been repeatedly shown to occur on non-myeloid cells when incubated with purified PR3; again, several of these studies are to be found in the reference list of this MS! It is perhaps the case - and then also perfectly acceptable - that the authors do not agree with all of this published data. But then this should be openly discussed and substantiated and not simply called into question via such confusing statements.

From the Introduction, page 4:

"It is thought that this response is at least in part due to Pr3-ANCA-mediated recruitment and activation of Fcγ receptors (37,44,45)".

This statement refers to the involvement of Fc receptor recruitment for the PR3-ANCA stimulated respiratory burst response of neutrophils...this was substantiated also in reference 20 in the MS but has not been included in the citation.

From the Introduction, page 5:

"The study was possible using a site-directed mutagenesis screen to identify stable Pr3 variants that are readily expressed and purified from HEK293 cultures".

As mentioned above, the mutation of the residues producing the monomeric and "stable" PR3 variant - which had been initiated by the observation that wtPR3 forms an array of higher order aggregates in solution at physiological salt concentrations - was previously fully described (reference 19 in the MS) along with measurement of its binding affinity to CD177 and an evaluation of the relevance of the mutations with regard to the binding interface. The current MS claims a modified approach (including adding a glycosylation) at the identical positions to achieve the same effect that the authors of reference 19 managed with alanine substitutions. It is of note here that, while a PR3 monomer peak is visible in the size exclusion data shown in Fig. S1 f, the mutagenesis apparently did not result in an exclusively monomeric species as misleadingly claimed in the text ("...is expressed at high levels yielding stable and monodisperse protein samples") as it did for the authors of reference 19 . (If the larger species in the chromatogram of Fig. S1 f are NOT higher order PR3 aggregates, the authors should convincingly show this, for example via Western blot). It is curious indeed that all of these previously and quite explicitly reported results are presented in the current MS as original with *no reference* to the older study that is actually cited several times for other reasons in the text. *This is a very serious error and raises a number of questions as to how it could have occurred.*

From the Introduction, page 3:

" CD177 (also named HNA-2a or NB1) is a ~50 kDa glycosylphosphatidylinositol (GPI)-anchored surface glycoprotein".

First of all, although one assumes that the CD177 homolog in question is the human one, this reviewer could find no indication of this - nor the association of the word "human" directly linked to CD177 - anywhere in the main body of the text. Fortunately the authors did include the Uniprot accession number in the Methods section; but where does their claim of "~50 kDa" come from? The full length protein sequence - that includes an N-terminal signal sequence not present in the final active form of the molecule - is calculated to be 46,4 kDa; the molecule that the authors crystallized and used for their other experiments is calculated to be 41,3 kDa...how are either of these values "~50 kDa"?? While these may seem like minor points, they are rather indicative of the lack of rigor that characterizes much of the MS - when unambiguous, established facts cannot be faithfully repeated, what is one to think of the "new" data presented and the interpretations thereof? A further issue is the repeated use of the tilde (~) to report measured values - where are the decimal places and standard error values?

From the Results, page 6:

"Surface plasmon resonance (SPR) and SEC-MALS experiments confirmed that Pr3rec binds CD177ecto with high affinity (Kd ~ 10 nM)".

Why is the affinity reported in such an imprecise way? Where are the decimal places and error calculations?

From the Results, page 7:

"The data show that CD177 binding does not occlude the Pr3 catalytic site, likely leaving it available for substrates".

While this statement - strictly speaking - may not be false, it is devoid of meaning. The issue is not whether the catalytic site is "available for substrates", it is whether enzymatic activity is affected upon PR3:CD177 complex formation (which could happen without a direct occlusion of the active site). Studies cited in the reference list of this MS have reported inhibition of PR3 activity in the presence of CD177 - *in vitro* and *in vivo*. Conformational changes occurring in multiple serine proteases have also been demonstrated. Incomprehensibly, none of this is included in discussion of the structures or mentioned in the text at all.

The Discussion section is comprised mostly of meandering filler instead of a thoughtful evaluation of the established biochemical data from many other groups in light of the only novelty in the MS, the structures of CD177 and the PR3:CD177 complex. The authors' own statement on page 11: "...the molecular architecture of Pr3 surface presentation also impacts Pr3-ANCA interactions and therefore warrants investigation" is not at all effectively followed up on in the remainder of the text. Based on the observation of wtPR3 aggregation alone (again, reported in the MS as a novelty although already published years ago), the "model" of the PR3:CD177 complex/ANCA interaction presented in Fig. 5 is almost certainly - completely - wrong. It is also unclear to which "hypothesis" is referred on page 12 - the issue of where a handful of patient ANCAs may bind PR3 has already been mentioned above. The 'theorizing' comprising the bulk of pages 12 and 13 could be deleted in its entirety with no deleterious effect at all on the MS. The discussion of the ELISA data with wt and mutated PR3 - which makes no mention of the "...faster detection..." observed with the mutant that is slipped into the legend of Figure 3D - provides no new insights and somehow also does not include the (not) novel observation of wtPR3 aggregation. Even the suggestions of better diagnostics and potential therapies based on knowledge of the complex and PR3 mutations is old news; there is nothing novel in any of it, all of these points have been suggested/discussed in work by others - as a reading of the cited literature and even a cursory perusal of the patent databases make clear.

Referee #3:

Referees are asked to supply answers to the following questions, with brief accompanying comments where appropriate:

1. Does this manuscript report a single key finding? YES/NO

If YES, please describe it in one sentence.

YES, The authors have resolved the crystal structure of PR3rec (catalytically inactive, lacking N-terminal pro dipeptide and mutated at three residues within the hydrophobic surface patch -to N-glycosylation motif) In complex with CD177 and demonstrated that CD177-binding site on PR3 has an epitope specific for PR3-ANCA of the patients used in this study.

2. Is the reported work of significance (YES), or does it describe a confirmatory finding or one that has already been documented using other methods or in other organisms etc (NO)? YES/NO

YES

3. Is it of general interest to the molecular biology community? YES/NO

If YES, please say why, in a single sentence. If NO, please state which more specialized community you feel it is aimed at (or none), in a single word or phrase.

NO, Immunology/Clinical immunology

4. Is the single major finding robustly documented using independent lines of experimental evidence (YES), or is it really just a preliminary report requiring significant further data to become convincing, and thus more suited to a longer format article (NO)?

YES/NO

YES

1. What are the major claims and how significant are they?

The manuscript focuses on the structural characterization of PR3rec-CD177sub1 (subdomain I of CD177) complex. The authors have resolved the crystal structure of recombinant CD177 ectodomain (CD177ecto) , which shows different domains previously described in the literature. The authors have performed mutations to introduce N-linked glycosylation in the CD177 binding site of PR3 and have shown that these mutations inhibit CD177 binding. Further, they show that PR3rec-CD177ecto binds weakly to PR3-ANCA obtained from PR3-patients compared to PR3rec. Moreover, PR3 mutant (Pr3nonCD177), which does not bind to CD177 binds weakly to PR3-ANCA similar to PR3rec-CD177ecto (PR3-ANCA for not all patients behave in terms of binding to PR3rec-CD177ecto and Pr3nonCD177). Hence, the PR3 residues which bind to CD177 form epitope for PR3-ANCA. binding.

2. Are the claims novel and convincing?

The claims are novel and convincing. However, the authors should also mention that glycosylation linked mutations introduced on PR3 can influence the binding of PR3-ANCA which can result in the high affinity observed. Indeed epitope specificity of PR3-ANCA can be altered because of these mutations. Moreover, the additional mutations performed in PR3rec to produce Pr3nonCD177 can be the cause of reduced PR3-ANCA binding.

3. Are the claims appropriately discussed in the context of earlier literature?

Yes, but it will be important to mention that authors haven't compared their PR3-ANCA binding ELISA (figure4B) results with the most used form of recombinant PR3 protein.

4. Who will be interested and why?

PR3-AAV is a rare autoimmune disease where autoantibodies target autoantigen PR3 and has no cure. Current treatments involve broad spectrum drugs, which targets the immune cells of the patients. These drugs have side effects and are associated with relapses. Therefore, a better understanding of the molecular and cellular mechanisms involved in the pathophysiology of the disease are needed. Autoantibodies target the PR3 protein expressed on the neutrophil membrane which is often found in complex with CD177. This study shows that the CD177 binding site of PR3 can also bind PR3-ANCA. Therefore, it is an important addition in understanding the molecular pathways involved in the progression of PR3-AAV.

5. Does the paper stand out in some way from the others in its field?

Yes

6. Are the experimental data of sufficient quality to justify the conclusions?

Following should be done to improve the data before accepting the manuscript:

Figure 3C - Graph should be replotted with statistics by including the data of two other experiments .

Figure 3D -

Mix of plasma from three different patients (P02, P05, P08) was used to perform ELISA. The results of Figure 3D are not matching with the results of figure 4B. In figure 3D PR3-ANCA mix from P02, P05, P08 show almost similar binding pattern with PR3rec, Pr3nonCD177 and PR3rec-CD177ecto but in figure 4B plasma from patients P02 binds significantly less to PR3rec-CD177ecto and Pr3nonCD177 compared to PR3rec.

Therefore, it is important to repeat the experiment with only plasma from P05 and P08, and a mix of P05 and P08 and add statistics to the experiment.

Figure 3E- Experiment with only CD177ecto should also be performed to justify the enhanced anti-HIS signal obtained in PR3rec-CD177ecto

Figure 4B -

It would be important to perform ELISA with i plasma from healthy donors

It would be important to include PR3rec-CD177subl as a ligand because study has resolved the crystal structure of PR3rec-CD177subl .

In the graph label the data which is non-significant

Supplementary Figure 1(F): Label the PR3 protein band

Supplementary Figure 2 : Western blotting of the fraction eluted from SEC against CD177 and PR3 proteins should be included

In the supplementary data section, it would be important to include the SPR data.

I suggest that Pr3 should be written as PR3 according to the nomenclature.

I think the experiments suggested for revision are achievable in the timeframe of 12 months.

Authors: we thank the reviewers for their careful reviews and comments which have helped us to improve the manuscript. We have responded with additional data and analysis, added more clarification where needed, and we revised text and figures accordingly.

Please note that we renamed the supplementary figures to “extended view” figures (Fig. EV1-5), as required by *EMBO R*.

Referee #1

This manuscript by Zheng-Gérard et al. reports an integrated study spearheaded by structural studies of the complex of CD177 with Proteinase Pr3 to provide insights into the binding epitope targeted by autoantibodies in GPA. This study provides novel insights based on high quality data and new molecular tools that will help an important field of biomedical research move forward. I herewith provide input for the authors' consideration towards improving their manuscript.

Authors: we are grateful for the positive assessment and constructive comments.

The title of the manuscript does not quite capture fully the message of the work. For instance, one might expect to see reference to the reported complex of Pr3 and the CD177 receptor.

Authors: We are happy to modify the title to reflect this aspect and have adjusted it as follows:

“Structure of proteinase 3 and the CD177 receptor (complex) reveals major autoantibody epitope”

The authors need to discuss prior work on anti-Pr3 antibodies, for instance as summarized by Peen Ralph Arthritis Res. 2000 Jun 12;2(4):255-259. doi: 10.1186/ar97, as well as other contributions.

Pr3 is known to exist in a membrane-bound form, albeit in a small fraction, which also has an altered activity profile due to conformational differences compared to soluble Pr3. For instance, work by Guarino et al. J Biol Chem 2023 Feb 26;299(4):103072. doi: 10.1016/j.jbc.2023.103072

Authors: We thank the reviewer for their comment and have now added a brief discussion of prior work on anti-PR3 antibodies in the introduction and discussion section: “*Pioneering work in mapping PR3 epitopes used synthetic overlapping peptides* (Williams et al., 1994; Griffith et al., 2001), *recombinant chimeric recombinant PR3* (Van Der Geld, Limburg and Kallenberg, 1999; Kuhl et al., 2010; Silva et al., 2010), *and competition experiments with mouse monoclonal antibodies (mAbs) or a patient-derived PR3-ANCA* (Van Der Geld, Limburg and Kallenberg, 1999; Kuhl et al., 2010; Silva et al., 2010). *The latter, together with grafting experiments, revealed four major conformational epitopes on the PR3 surface (43–45) targeted by mouse mAbs and PR3-ANCA. An anti-CD177 Fab, which blocks PR3-CD177 binding, was shown to remove CD177-bound PR3 from neutrophil surfaces, presumably by competing for binding* (Marino et al., 2022). *The presence of this anti-CD177 Fab also*

reduced PR3-ANCA-induced activation of CD177^{pos}/PR3^{high} neutrophils, down to the lower level measured for CD177^{neg}/PR3^{low} neutrophils (Marino et al., 2022).” (page 4).

“...previous studies identified IgM anti-PR3 autoantibodies (Sibilia et al., 1997; Davis et al., 1998; Peen and Williams, 2000). Analysis of one autoantibody (WGH1) indicated that the CDR3 region was accessible for interaction with positively charged residues on PR3 (Davis et al., 1998). However, a more recent study identified 19 PR3-specific B cells expressing IgG, with them being predominantly IgG1 and exhibiting an enrichment for IgG4 (Kelly et al., 2024). Another study used a patient-derived monoclonal ANCA (moANCA518) and demonstrated that it bound selectively to Epitope 3 of PR3 (Casal Moura et al., 2023)”(pages 12-13)

Thank you for highlighting the important work by Guarino et al. 2023, which we now reference: *“Most extracellular PR3 is presented on the neutrophil cell surface in complex with its receptor CD177 (also named HNA-2a or NB1) (Bauer et al., 2007a; Von Vietinghoff et al., 2007a; Abdgawad et al., 2010a), while some PR3 can bind directly to cell membrane components (Goldmann, Niles and Arnaout, 1999a; Hajjar et al., 2007; Hu et al., 2009a; Broemstrup and Reuter, 2010a; Kantari et al., 2011; Carla Guarino et al., 2023)” (page 3).*

Despite presenting several high quality SPR data, with well-fitted sensorgrams (presumably to 1:1 Langmuir models), the authors do not provide the kinetics of binding (k_a and k_d). This will need to be reported and discussed, not only to complete the data presentation, but mainly because on/off rates are often very relevant in comparatively understanding antibody binding profiles.

Authors:

We thank the reviewer for this helpful comment and agree that reporting kinetic parameters will strengthen the interpretation of our SPR data.

We agree that the kinetics of receptor-antigen interactions provide critical context for interpreting antibody binding profiles. Although the K_D reflects overall affinity, it does not capture the underlying dynamics, and when antibodies compete with a receptor for antigen binding, their relative on and off rates can be as important as the net affinity in determining functional potency. A kinetic analysis of the binding data has now been done in addition to the original steady-state analysis. The resulting kinetic parameters can be found in Figure EV 2F.

How do the authors think the introduced N-linked glycosylation on Pr3 might affect binding? It would be very instructive and useful to readers and future adopters of the approach to include in Suppl. Fig. 1 an SDS-PAGE analysis of PNGase-treated protein samples to illustrate the presence of introduced N-glycans.

Authors: We have added the requested SDS-PAGE analysis illustrating the presence of N-glycans on our engineered PR3 constructs in Figure EV1panel H. The analysis of our SPR experiment suggests a K_D of 10.8 nM (steady state) or 7.3 nM (kinetics analysis) for PR3^{rec} with CD177^{ecto}, which is broadly within the same range as the K_D of 4 nM previously published by Jerke et al. (ref 18 in paper). We now discuss how changes in the wild type sequence of PR3 could potentially exert local or allosteric conformational changes that impact ligand binding and included a brief discussion of previous studies that have detected

mutation-induced changes in PR3 conformation : “It is important to note that deviations from the wild type PR3 sequence (be it through swaps, mutations and/or introductions of glycans) could result in local or allosteric conformational changes of PR3, which in turn could influence ANCA binding...” (page 13).

What are the authors' thoughts on why the apparent Pr3 hydrophobicity is there in the first place? Any implications for the binding of other proteins, antibodies etc. Please discuss accordingly.

Authors: This is an interesting question and we now added the following in the introduction. “...

“Pioneering experiments suggested that a “hydrophobic patch” on human PR3, which protects from apoptotic cell clearance (Kantari *et al.*, 2011; Gabillet *et al.*, 2012), is centred around the residues F180, F181, I221, W222, L228, and F229. This patch is not conserved in other primates and thought to bind to hydrophobic surfaces on CD177 (Korkmaz *et al.*, 2008). These findings are consistent with the fact that gibbon PR3 does not interact with human CD177-expressing cells (Korkmaz *et al.*, 2008).”

+++++

Technical input:

+++++

Please make sure that the correct nomenclature for binding constants is used throughout the manuscript, i.e. capital K in italics and subscript D in non-italics for the KD, and likewise for the eventual reports of ka and ks (see above), k in italics and a/d as non-italic subscripts.

Authors: Done

Table S1 and Table S2: The R-merge reported is an obsolete (in fact erroneous to a large extent) crystallographic metric and this reviewer would strongly discourage its use given that Rmeas/Rpim and CC1/2 have been appropriately included.

Authors: We agree and have amended the supplementary table accordingly.

Table S1 and Table S2: In addition to the average B factor reported for protein atoms, please report average B factors (or Atomic Displacement Parameters) for protein, solvent, heteroatoms etc, as well.

Table S1. Please report the anomalous differences for the S-SAD data set of CD177ecto.

Authors: We thank the reviewer for this suggestion and have amended the supplementary tables accordingly. We also show SigAno and $\langle d \rangle / \text{sig} \rangle$ as a function of resolution, in Supplementary Figure 1 (now called Figure EV1, according to the new EMBO R format).

Referee #2

The manuscript (MS) by Zheng-Gérard, et al. reports the structure of the neutrophil PR3:CD177 complex as well as that of CD177 alone, thus providing two major pieces of data missing from the thorough characterization of the complex. This is a most welcome advance for the field and it is wonderful to finally see these structures that, together with the extensive biochemical data already in the literature, will be of great value in improving our understanding of neutrophil biology and for the development of treatments for PR3-ANCA diseases.

Authors: We thank the reviewer for this positive assessment.

It is therefore most unfortunate that a careful analysis of the structural data in light of the ample biochemical information already published is *absent from the text*. The MS is furthermore marred by examples of sweeping generalization that are neither necessary nor warranted, as well as statements that are quite misleadingly formulated or altogether inaccurate. Most peculiar indeed is the reporting of a mutagenesis scheme for preparing a monomeric PR3 - an effort occupying substantial space in the MS and presented as novel - *although the "monomerization" of PR3 based on mutating the **same positions** had already been reported elsewhere*; that study is actually referenced by the authors of the present MS but no discussion of that PR3 data is presented in the text. The inaccuracies, contradictions and omissions lead one to wonder whether the authors actually read many of the studies that they cite in their reference list; reading the text does not convey the impression that the authors are aware of data published by many groups over the last several decades. Despite the welcome advance provided by the structural data, this MS is **not fit for publication anywhere** in its present form.

Authors: We appreciate the feedback and have improved the main text accordingly (see points below), for example by discussing the work by Jerke et al (Jerke *et al.*, 2017) at appropriate points in the text. Please note that the recombinant mutant PR3 we present was developed independently of other studies and that we started the project before the publication of the aforementioned paper. In contrast to Jerke et al. who performed alanine substitutions, we introduced mutations that result in an N-linked glycosylation site, rather than alanines. This significantly improves the solubility and homogeneity of the sample and was instrumental to being able to co-crystallise the protein with CD177, in our hands.

We have rewritten the relevant section in our results to relate our construct to the previously published mutant used by Jerke et al.:

“Using an established protein engineering approach that introduces an N-linked glycosylation site (del Toro et al., 2020; Akkermans et al., 2022), we mutated three residues within the PR3 hydrophobic surface patch: I221N-W222G-G223T (Figure EV 1F). A previously published mutant in which two of these residues were mutated to alanines (I221A-W222A) had also been reported to improve protein solubility (Jerke et al., 2017).”

In the discussion we write:

“The hydrophobic nature of wild type PR3, which is thought to be due to the “hydrophobic patch” on the protein surface (Korkmaz et al., 2008), has presented a challenge for the production and purification of this protein for structural studies. Over the last 30 years, protocols were established for the expression of recombinant PR3 in yeast (Harmsen et al., 1997), Sf9 insect cells (Fujinaga et al., 1996), human mast cell line-1 HMC-1 (Specks et al., 1996; Sun et al., 1998; Van Der Geld et al., 2000) and HEK293 cells (Sun et al., 1998; Van

Der Geld *et al.*, 2000; Korkmaz *et al.*, 2008; Jerke *et al.*, 2017). *Several of these purification trials have successfully used detergents such as 0.1% Triton X-100 (Robert F. Halenbeck et al., 1995) 1% β -OG (Stummann and Wiik, 1997), 0.1% Tween-80 (Van Der Geld et al., 2002) or 0.02% DDM (Jerke et al., 2017) to mitigate protein aggregation, although it is recognised that these detergents can hamper biological experiments and crystallisation trials. Mutagenesis of the “hydrophobic patch” residues I221 and W222 to alanines resulted in a promising construct with improved solubility (Jerke et al., 2017). Our work was catalysed by introducing an N-linked glycosylation site in position 221 (PR3^{rec}), which made production and purification from HEK293 cells and successful co-crystallisation with CD177 receptor without the use of detergent possible. “*

A discussion of the major flaws in the MS follows here and all of these issues must be adequately addressed before the MS can be submitted to any journal with a reasonable expectation of acceptance.

1) We are long past the days when the generation of a structure alone, however competently done (as the current example appears to be), suffices for a publication (unless combined with some new technical advance or other novelty that was necessary for achieving the structure). It is here emphasized that it is very exciting to now have these structures and their thoughtful analysis will be valuable for understanding the biology of ANCA-related diseases and also for developing treatments. But just such a thoughtful analysis - with confirmatory experimental data - is, unfortunately, missing from the MS. A number of experiments with the complex are reported but there isn't any meaningful analysis in light of previously published predictions - for example, Kantari et al. (reference 31 in the MS) made the "hydrophobic patch" responsible for the attachment of PR3 to the outer neutrophil membrane and Korkmaz, et al. (reference 81 in the MS) claimed that the "hydrophobic patch" was responsible for the interaction with CD177 - does the structural data (which is what is new) provide any insight into the correctness of these suppositions?

A rather detailed model for the complex based on biochemical data was proposed in 2017 (Jerke, et al. - reference number 19 in the MS) - how does the presently reported complex structure compare with those predictions from the biochemical characterization? These examples - and more - should make up a substantial portion of the discussion section of the current manuscript but are entirely absent (although the primary sources are in the reference list). The structures alone are the only new information.

Authors: We thank the reviewer for this feedback and appreciation of the structural results. We have improved and expanded our results section, supplemental figures and discussion section to the published definition of the “hydrophobic patch”:

Results: “The centre of the binding surface contains hydrophobic residues (F180, F181, F190 and L228 from PR3, and L117, P118 and W120 from CD177) (Figure 2D-F). These residues are located within the previously described “hydrophobic patch” of PR3 (Korkmaz et al., 2008) (Figure EV3 panel C).”

Figures: see Figure EV3 panel C.

Discussion: “*The structure shows that the PR3-CD177 binding interface overlaps, at least in part, with the ‘hydrophobic patch’* (Korkmaz *et al.*, 2008; Jerke *et al.*, 2017) (Figure EV3 C), validating previous studies suggesting that mutations in this region reduce CD177 binding (Jerke *et al.*, 2017).”

Please note that, given the focus of this manuscript on how PR3 interacts with its receptor CD177 and the importance of the binding interface as an epitope targeted by PR3-ANCA, it is beyond the scope of this report to analyse how the “hydrophobic patch” as defined by others may function to mediate PR3 attachment to cell membranes etc.

2) The authors give substantial space to their mutagenesis effort to achieve a monodisperse PR3 construct and report finding mutations that render the enzyme monomeric. Quoting from their text: “**The study was possible using a site-directed mutagenesis screen to identify stable Pr3 variants that are readily expressed and purified from HEK293 cultures.**” Perhaps it is an oversight (but then a very serious one) that a successful monomerization effort *was already published in 2017* (reference 19 in the MS): the mutated Ile - Trp (i - i+1) pair was shown to render PR3 monomeric and the affinity of that monomer for CD177 by SPR was also reported. The data in the current MS are therefore not new as the authors claim (using another residue numbering scheme does not alter the sequence itself) and this previous work is not acknowledged in their text (although the paper in question does appear in the reference list). This reviewer will stop short of labeling this an ethical problem, but the “oversight” must be corrected no matter where the next version of the MS is to be published.

Authors: We are grateful for the reviewers suggestion to discuss the Jerke *et al.* paper in more depth. As noted above, our mutant (PR3^{rec}) is not the same as the one published in Jerke *et al.*, and we have now added a clear description of how the two constructs compare in the relevant sections (see our response to previous comments further up). Regarding the numbering scheme, we have followed the scheme that is also used by the UniProt database, where the first Met residue in a sequence is numbered 1.

3) This reviewer feels that a comment on style here is also necessary: apparently also gone are the days when formulations in scientific reports minimized use of the first person; while this can sometimes not be gracefully avoided, the practice of introducing basic (if not trivial) results in this MS with overly self-promoting phrases such as “We pioneered...” (page 11), as well as numerous other “We” statements scattered throughout, should be and, considering that much of what is described in the MS was already reported by others and/or not sufficiently acknowledged as such, maximized the discomfort felt when reading the text. These deficiencies, as well as the grammar mistakes and confusing formulations (some, but not all, of which are addressed explicitly below) must be corrected before the next submission, to whichever journal that may be.

Authors: this is an interesting point and we refer to the editor for guidance. Many journals ask for an active style (“we did this...”, “we did that...”) rather than passive voice (“the sample was subjected...”, “it was thought that...”). We will adapt to whichever style is preferred by the editors. We will also do our very best to weed out any typos etc.

4) There are a number of statements throughout the text that are either partly or wholly inaccurate; some examples follow (this list is NOT exhaustive), with comments from this reviewer in **bold**:

From the Abstract, page 1:

"Granulomatosis with polyangiitis is a life-threatening systemic vasculitis, characterised by antineutrophil cytoplasmic autoantibodies (ANCA) against proteinase 3 (Pr3) expressed on the surface of neutrophils. Most cell surface Pr3 is bound to the receptor CD177...".

These first two sentences of the paper are quite garbled - the first reads as if the ANCA themselves are "expressed" on neutrophils...which of course is not the case; furthermore, PR3 is found on the extracellular surface of all neutrophils, even those not displaying CD177...and even on neutrophils from persons who are *genetically negative for CD177*, i.e. whose neutrophils completely lack CD177.

Authors: this has been adjusted to read 'Granulomatosis with polyangiitis is a life-threatening systemic vasculitis, characterised by anti-neutrophil cytoplasmic autoantibodies (ANCA) against proteinase 3 (PR3), a protease expressed on the surface of neutrophils.'

From the Abstract, page 1:

"We describe an extensive, mainly hydrophobic binding interface between Pr3 and CD177...".

This is an unnecessarily and incorrectly qualified statement - the authors report the buried surface area in the complex to be " ~655 Å² " - a value rather at the low end of the buried surface area spectrum that extends, on average, between 1500 and 3000 Å² (see for example 10.1073/pnas.93.1.13); How have the authors decided that the PR3:CD177 interaction is "extensive"?

Authors:

We have removed the word 'extensive' from that sentence.

From the Abstract, page 2:

"Using a panel of Pr3-ANCA positive patient samples, we show that CD177 inhibits Pr3-ANCA binding to Pr3".

It is obvious that this statement on its face cannot be accurate. Certainly the binding of some ANCAs is blocked by the presence of CD177 (as the authors show with their experiments using patient samples) but, as reported later in the MS and has been shown previously in multiple studies (see for example J Immunol. 1994 May 1;152(9):4722-37, J Autoimmun 2010 35, 299-308 (reference 78 in the MS) and also reference 20 in the MS, among others) ANCA epitopes are also found outside of the CD177 binding site and some ANCAs can bind to PR3 that is complexed with CD177.

Authors: We have revised this sentence to say ‘Using a panel of PR3-ANCA positive patient samples, we show that a significant proportion of ANCAs target the CD177-binding site of PR3.’

The authors also write on page 5: "We discover that most Pr3-ANCA positive samples compete with the interaction between Pr3 and CD177..." **and furthermore on page 9:** "Most GPA patient samples harbour autoantibodies that target the CD177-binding site of Pr3" **because, in their ELISA, 7 of 8 samples appear to bind an epitope coinciding with the CD177 binding site (their conclusion). Now, even accepting the authors' own - not referenced!! - "estimate" of the prevalence of GPA (Introduction, page 2) of "2 - 150 per million individuals", it is clear that these statements cannot be true. Taking the lower limit of this - again not referenced - estimate (and since PR3-ANCA are involved in other diseases and the number of *undiagnosed* cases of PR3-ANCA diseases is thought to be substantial, the authors' estimate is clearly extremely low), the number of patients must number in the hundreds of thousands/millions, depending on where the authors' "estimate" applies (in England? In Taiwan?? Worldwide?). To have tested samples from <0,001% of all patients (based on the absolute - and incorrect - minimum estimate) and then drawing a firm conclusion about them *all* cannot be taken seriously.**

Authors:

We are of course only focusing on the panel of GPA patient samples presented in this study therefore we have adjusted this sentence to make it clearer: “*We discover that most of the PR3-ANCA positive samples tested here target the CD177-binding site of PR3 and display reduced binding to PR3 protein where the CD177-binding site is masked.*” (page 5).

From the Introduction, page 3:

"...although some PR3 appears to bind directly to cell membrane components".

This is a perplexingly misleading statement- the word "appears" here is simply inaccurate; there is ample published evidence, over many years, demonstrating that PR3 is present on the neutrophil cell surface - even CD177 negative individuals have mPR3 on their neutrophils. This surface interaction has also been repeatedly shown to occur on non-myeloid cells when incubated with purified PR3; again, several of these studies are to be found in the reference list of this MS! It is perhaps the case - and then also perfectly acceptable - that the authors do not agree with all of this published data. But then this should be openly discussed and substantiated and not simply called into question via such confusing statements.

Authors: We have revised the text and it now reads “*Most extracellular PR3 is presented on the neutrophil cell surface in complex with its receptor CD177 (also named HNA-2a or NBI) (Bauer et al., 2007b; Von Vietinghoff et al., 2007b; Abdgawad et al., 2010b), although PR3 can also bind directly to cell membrane components (Goldmann, Niles and Arnaout, 1999b; Hajjar et al., 2008; Hu et al., 2009b; Broemstrup and Reuter, 2010b; Kantari et al., 2011).*”(page 3).

As mentioned above, this aspect of PR3 biology is not a major focus of the paper and we do not wish to prove or disprove previous work.

From the Introduction, page 4:

"It is thought that this response is at least in part due to Pr3-ANCA-mediated recruitment and activation of Fcγ receptors (37,44,45)".

This statement refers to the involvement of Fc receptor recruitment for the PR3-ANCA stimulated respiratory burst response of neutrophils...this was substantiated also in reference 20 in the MS but has not been included in the citation.

Authors: we would have been happy to cite this paper, but in the process of streamlining the paper to make it more focused and reader-friendly (in response to other comments), we have removed this section from the introduction. As the data presented in our paper is not focused on the Fc-gamma receptor, nor involves experiments with neutrophils, we think that this paragraph is not strictly necessary.

From the Introduction, page 5:

"The study was possible using a site-directed mutagenesis screen to identify stable Pr3 variants that are readily expressed and purified from HEK293 cultures".

As mentioned above, the mutation of the residues producing the monomeric and "stable" PR3 variant - which had been initiated by the observation that wtPR3 forms an array of higher order aggregates in solution at physiological salt concentrations - was previously fully described (reference 19 in the MS) along with measurement of its binding affinity to CD177 and an evaluation of the relevance of the mutations with regard to the binding interface. The current MS claims a modified approach (including adding a glycosylation) at the identical positions to achieve the same effect that the authors of reference 19 managed with alanine substitutions. It is of note here that, while a PR3 monomer peak is visible in the size exclusion data shown in Fig. S1 f, the mutagenesis apparently did not result in an exclusively monomeric species as misleadingly claimed in the text ("...is expressed at high levels yielding stable and monodisperse protein samples") as it did for the authors of reference 19 . (If the larger species in the chromatogram of Fig. S1 f are NOT higher order PR3 aggregates, the authors should convincingly show this, for example via Western blot). It is curious indeed that all of these previously and quite explicitly reported results are presented in the current MS as original with *no reference* to the older study that is actually cited several times for other reasons in the text. *This is a very serious error and raises a number of questions as to how it could have occurred.*

Authors: Thank you for this helpful comment. We now updated the relevant supplemental figure legend (Fig. EV1 panel G) to include a distinct description of the SDS-PAGE gel (Coomassie) and anti-His western blot (a-his). The originally submitted figure showed one of the first purifications attempted in the lab where the higher band at 60 kDa in the western blot -but not on the SDS-PAGE gel- accounts for a histidine-rich impurity that is sometimes co-purified in small amounts, depending on the quality of the immobilised metal ion affinity chromatography (IMAC) experiment. We see this sometimes, and for different proteins,

produced in our system. This impurity, if present in tiny amounts, typically does not impact our crystallisation trials, but we do make sure to remove it by thorough wash for any functional studies, such as ELISA. To reflect this, we now show the SEC chromatogram, SDS-PAGE and western blot analysis from a later, cleaner purification of PR3^{rec}, see Figure EV1 panel G.

From the Introduction, page 3:

" CD177 (also named HNA-2a or NB1) is a ~50 kDa glycosylphosphatidylinositol (GPI)-anchored surface glycoprotein".

First of all, although one assumes that the CD177 homolog in question is the human one, this reviewer could find no indication of this - nor the association of the word "human" directly linked to CD177 - anywhere in the main body of the text. Fortunately the authors did include the Uniprot accession number in the Methods section; but where does their claim of "~50 kDa" come from? The full length protein sequence - that includes an N-terminal signal sequence not present in the final active form of the molecule - is calculated to be 46,4 kDa; the molecule that the authors crystallized and used for their other experiments is calculated to be 41,3 kDa...how are either of these values "~50 kDa"?? While these may seem like minor points, they are rather indicative of the lack of rigor that characterizes much of the MS - when unambiguous, established facts cannot be faithfully repeated, what is one to think of the "new" data presented and the interpretations thereof? A further issue is the repeated use of the tilde (~) to report measured values - where are the decimal places and standard error values?

Authors: Indeed the reviewer is right that the predicted protein mass of human CD177 is less than 50kDa, however, the protein includes 3 predicted N-linked glycosylation sites, each accounting for an unknown size glycan. A previous mass spectrometry study (Kissel *et al.*, 2001) found native glycosylated CD177 to be 50.556 kDa. Our use of "~50 kDa ... glycoprotein" reflects the findings in this previous study. We have now added the reference for clarity. (page 3)

From the Results, page 6:

"Surface plasmon resonance (SPR) and SEC-MALS experiments confirmed that Pr3^{rec} binds CD177ecto with high affinity (Kd ~ 10 nM)".

Why is the affinity reported in such an imprecise way? Where are the decimal places and error calculations?

Authors: We have now included the standard deviation values in the main text as well as in Figure EV 2F.

From the Results, page 7:

"The data show that CD177 binding does not occlude the Pr3 catalytic site, likely leaving it available for substrates".

While this statement - strictly speaking - may not be false, it is devoid of meaning. The issue is not whether the catalytic site is "available for substrates", it is whether enzymatic activity is affected upon PR3:CD177 complex formation (which could happen without a direct occlusion of the active site). Studies cited in the reference list of this MS have reported inhibition of PR3 activity in the presence of CD177 - *in vitro* and *in vivo*. Conformational changes occurring in multiple serine proteases have also been demonstrated. Incomprehensibly, none of this is included in discussion of the structures or mentioned in the text at all.

Authors: We now provide a discussion of this point: “

Previous work suggested a negative effect of CD177 on PR3 activity: the addition of purified CD177 was shown to inhibit nPR3 activity in vitro, in neutrophil degranulation experiments, and at the surface of neutrophils or CD177-expressing HEK cells (Jerke et al., 2017). Here we show that CD177-binding does not directly occlude the PR3 active site, nor does it perturb the PR3 structure compared to unliganded PR3 (Figure EV3 B). Therefore the mechanism of inhibition is likely more subtle, for example CD177 could sterically affect substrate binding away from the active site.” (page 11)

The Discussion section is comprised mostly of meandering filler instead of a thoughtful evaluation of the established biochemical data from many other groups in light of the only novelty in the MS, the structures of CD177 and the PR3:CD177 complex. The authors’ own statement on page 11: "...the molecular architecture of Pr3 surface presentation also impacts Pr3-ANCA interactions and therefore warrants investigation" is not at all effectively followed up on in the remainder of the text. Based on the observation of wtPR3 aggregation alone (again, reported in the MS as a novelty although already published years ago), the "model" of the PR3:CD177 complex/ANCA interaction presented in Fig. 5 is almost certainly - completely - wrong. It is also unclear to which "hypothesis" is referred on page 12 - the issue of where a handful of patient ANCAs may bind PR3 has already been mentioned above. The ‘theorizing’ comprising the bulk of pages 12 and 13 could be deleted in its entirety with no deleterious effect at all on the MS. The discussion of the ELISA data with wt and mutated PR3 - which makes no mention of the "...faster detection..." observed with the mutant that is slipped into the legend of Figure 3D - provides no new insights and somehow also does not include the (not) novel observation of wtPR3 aggregation. Even the suggestions of better diagnostics and potential therapies based on knowledge of the complex and PR3 mutations is old news; there is nothing novel in any of it, all of these points have been suggested/discussed in work by others - as a reading of the cited literature and even a cursory perusal of the patent databases make clear.

Authors: We have reworked our discussion, for example, including previous biochemical analysis done to reduce PR3 aggregation. We also replaced the “structure-based” views of CD177 and PR3 in Figure 5 with more abstract representations to avoid any confusion with regards to the detailed molecular mechanisms of ANCA-PR3 interaction. In response to the comments, we now provide a better context for the section on the future development of diagnostics and therapies. We maintain that our work is novel not just with regards to the structural and biophysical work (as recognised by all three reviewers), but also with regards to the ELISA study. In contrast to previous studies, we are able to compare the binding of patient samples to different PR3 variants (or complex), using an anti-His. We believe that the

newly revised discussion now provides a balanced view of previous and novel results, within the limitations of a concise manuscript.

Referee #3

Referees are asked to supply answers to the following questions, with brief accompanying comments where appropriate:

1. Does this manuscript report a single key finding? YES/NO

If YES, please describe it in one sentence.

YES, The authors have resolved the crystal structure of PR3rec (catalytically inactive, lacking N-terminal pro dipeptide and mutated at three residues within the hydrophobic surface patch -to N-glycosylation motif) In complex with CD177 and demonstrated that CD177-binding site on PR3 has an epitope specific for PR3-ANCA of the patients used in this study.

2. Is the reported work of significance (YES), or does it describe a confirmatory finding or one that has already been documented using other methods or in other organisms etc (NO)?

YES/NO

YES

3. Is it of general interest to the molecular biology community? YES/NO

If YES, please say why, in a single sentence. If NO, please state which more specialized community you feel it is aimed at (or none), in a single word or phrase.

NO, Immunology/Clinical immunology

4. Is the single major finding robustly documented using independent lines of experimental evidence (YES), or is it really just a preliminary report requiring significant further data to become convincing, and thus more suited to a longer-format article (NO)? YES/NO

YES

Authors: thank you for the positive evaluation.

1. What are the major claims and how significant are they?

The manuscript focuses on the structural characterization of PR3rec-CD177subI (subdomain I of CD177) complex. The authors have resolved the crystal structure of recombinant CD177 ectodomain (CD177ecto) , which shows different domains previously described in the literature. The authors have performed mutations to introduce N-linked glycosylation in the CD177 binding site of PR3 and have shown that these mutations inhibit CD177 binding. Further, they show that PR3rec-CD177ecto binds weakly to PR3-ANCA obtained from PR3-patients compared to PR3rec. Moreover, PR3 mutant (Pr3nonCD177), which does not bind to CD177 binds weakly to PR3-ANCA similar to PR3rec-CD177ecto (PR3-ANCA for not all patients behave in terms of binding to PR3rec-CD177ecto and Pr3nonCD177). Hence, the PR3 residues which bind to CD177 form epitope for PR3-ANCA. binding.

Authors: thank you for the good summary

2. Are the claims novel and convincing?

The claims are novel and convincing. However, the authors should also mention that glycosylation linked mutations introduced on PR3 can influence the binding of PR3-ANCA which can result in the high affinity observed. Indeed epitope specificity of PR3-ANCA can be altered because of these mutations. Moreover, the additional mutations performed in PR3^{rec} to produce Pr3nonCD177 can be the cause of reduced PR3-ANCA binding.

Authors: We thank the reviewer for their positive assessment and helpful comments. Regarding the effects of the glycosylation, we agree and addressed this point in the new version of the discussion: *“It is important to note that deviations from the wild type PR3 sequence (be it through swaps, mutations and/or introductions of glycans) could result in local or allosteric conformational changes of PR3, which in turn could influence ANCA binding. This was highlighted by mutation of the previously reported epitope 3, which led to an unexpected increase in PR3-ANCA binding, possibly due to allosteric conformational changes and the exposure of a latent epitope (Casal Moura et al., 2023). Indeed, molecular dynamic simulations indicated that a distal region away from the mutation became more accessible for ANCA binding (Pang et al., 2019). Allosteric effects on PR3 activity have also been shown, for example for mouse anti-PR3 MCPR3-7 (Hinkofer et al., 2013) and upon CD177 binding (Jerke et al., 2017). Our mutants were designed such as to minimise the predicted disturbance of the PR3 fold. Additional reassurance is given by the fact that the structure of PR3^{rec} as found in complex with CD177 is near-identical to the published structure of unliganded PR3 containing the native sequence (Fujinaga et al., 1996) (Figure EV3 B) and AlphaFold models suggest that PR3^{rec} and PR3^{nonCD177} adopt essentially the same fold as native PR3 (Figure EV3 E). Whilst the PR3 fold is likely preserved in PR3^{rec}, the mutations we introduced could potentially inhibit the binding of PR3-ANCA subpopulations targeting this surface. “* (page 13)

3. Are the claims appropriately discussed in the context of earlier literature?

Yes, but it will be important to mention that authors haven't compared their PR3-ANCA binding ELISA (figure4B) results with the most used form of recombinant PR3 protein.

Authors:

It is not clear to us from the literature, what the most used form of recombinant PR3 is. This information is not shared by many suppliers of PR3-ANCA kits as it is proprietary. Therefore, we opted to using a commercially available ELISA kit provided by Euroimmun (Figure EV5) for comparison.

4. Who will be interested and why?

PR3-AAV is a rare autoimmune disease where autoantibodies target autoantigen PR3 and has no cure. Current treatments involve broad spectrum drugs, which targets the immune cells of the patients. These drugs have side effects and are associated with relapses. Therefore, a better understanding of the molecular and cellular mechanisms involved in the pathophysiology of the disease are needed. Autoantibodies target the PR3 protein expressed on the neutrophil membrane which is often found in complex with CD177. This study shows

that the CD177 binding site of PR3 can also bind PR3-ANCA. Therefore, it is an important addition in understanding the molecular pathways involved in the progression of PR3-AAV.

Authors: we agree and thank the reviewer

5. Does the paper stand out in some way from the others in its field?

Yes

Authors: thank you for the positive evaluation

6. Are the experimental data of sufficient quality to justify the conclusions?

Following should be done to improve the data before accepting the manuscript:

Figure 3C - Graph should be replotted with statistics by including the data of two other experiments .

Authors: we have now completed those experiments and adjusted the figure to include statistical analysis (Figure 3C)

Figure 3D -

Mix of plasma from three different patients (P02, P05, P08) was used to perform ELISA. The results of Figure 3D are not matching with the results of figure 4B. In figure 3D PR3-ANCA mix from P02, P05, P08 show almost similar binding pattern with PR3^{rec}, Pr3^{nonCD177} and PR3^{rec}-CD177^{ecto} but in figure 4B plasma from patients P02 binds significantly less to PR3^{rec}-CD177^{ecto} and Pr3^{nonCD177} compared to PR3^{rec}.

Therefore, it is important to repeat the experiment with only plasma from P05 and P08, and a mix of P05 and P08 and add statistics to the experiment.

Authors: we agree that this requires clarification. Indeed, in the original Fig 3D the curves for PR3^{nonCD177} and PR3^{rec}+CD177^{ecto} are shallower than for PR3^{rec}, indicating lower binding efficacy, in agreement with P02 which seems to respond the most. It is a good idea to repeat this with the individual patient samples. We opted to repeat the whole experiment using nPR3, PR3^{rec}, PR3^{nonCD177}, PR3^{rec}+CD177^{ecto} and CD177^{ecto} where CD177^{ecto} is TwinStrep-tagged (as in the experiments shown in Figure 4). The results are broadly consistent with the data in Figure 4, where P02 has the highest binding for PR3^{rec} and P08 shows the lowest across all variants. We have also added statistics as requested doing simple regression analysis to compare the binding curves of PR3^{nonCD177} and PR3^{rec}+CD177^{ecto} to PR3^{rec}. We observed significant differences for P02, consistent with Figure 4, but we also saw significant differences for P05, P08, and P05+P08 in contrast to Figure 4. However, as our standardised ELISA calculates binding at only one timepoint, this might explain why these significant differences were not detected.

Figure 3E- Experiment with only CD177^{ecto} should also be performed to justify the enhanced anti-HIS signal obtained in PR3^{rec}-CD177^{ecto}

Authors: As mentioned above, we repeated the experiments for Figure 3D and E, using Twin-Strep CD177 ligands (same as used in Figure 4) instead of the previous His-tagged ligands. This removes the complication of detecting CD177 alongside PR3 in these experiments. The results are consistent with broadly consistent amounts of PR3 being

immobilised. Note that, whilst reassuring that we are able to immobilise relatively consistent amounts, the way we standardise our experiments in Figure 4 (using anti-His) means that we do not rely on equal ligand coating to directly compare results from different plates using different PR3 variants.

Figure 4B -

It would be important to perform ELISA with i plasma from healthy donors

It would be important to include PR3rec-CD177subI as a ligand because study has resolved the crystal structure of PR3rec-CD177subI .

In the graph label the data which is non-significant

Authors: Agreed, we now performed the ELISA test using plasma from a healthy donor (source: Cambridge Bioscience) and performed standardised ELISAs for all samples using the CD177^{subI} construct in complex with PR3^{rec}, too. Data is shown in Figure 4B (patient samples) and EV5 panel B (includes the negative controls, such as the healthy donor, which are all negative). A new CD177^{SubI} with a TwinStrep instead of a His tag was cloned and produced for these experiments. We now have also included non-significant (ns) comparisons on the graphs.

Supplementary Figure 1(F): Label the PR3 protein band

Authors: We thank the reviewer for spotting this oversight and have amended the figure with the label.

Supplementary Figure 2 : Western blotting of the fraction eluted from SEC against CD177 and PR3 proteins should be included

Authors:

Unfortunately, we do not have western blots of those exact fractions eluted from SEC or the original samples from this exact experiment. However, we opted to repeat western blotting against all His-tagged and TwinStrep-tagged PR3 and CD177 constructs used in this study. These data are shown in Figure EV2 panels D and E.

In the supplementary data section, it would be important to include the SPR data.

Authors: We thank the reviewer for requesting this and will submit SPR data as a supplementary data file.

I suggest that Pr3 should be written as PR3 according to the nomenclature.

Authors: we have change all mention of Pr3 to PR3 in the manuscript.

I think the experiments suggested for revision are achievable in the timeframe of 12 months.

Authors: agreed.

References:

Abdgawad, M. *et al.* (2010a) "Elevated neutrophil membrane expression of proteinase 3 is dependent upon CD177 expression," *Clinical and Experimental Immunology*, 161(1), pp. 89–97. Available at: <https://doi.org/10.1111/j.1365-2249.2010.04154.x>.

Abdgawad, M. *et al.* (2010b) "Elevated neutrophil membrane expression of proteinase 3 is dependent upon CD177 expression," *Clinical and Experimental Immunology*, 161(1). Available at: <https://doi.org/10.1111/j.1365-2249.2010.04154.x>.

Akkermans, O. *et al.* (2022) "GPC3-Unc5 receptor complex structure and role in cell migration," *Cell*, 185(21). Available at: <https://doi.org/10.1016/j.cell.2022.09.025>.

Bauer, S. *et al.* (2007a) "Proteinase 3 and CD177 are expressed on the plasma membrane of the same subset of neutrophils," *Journal of leukocyte biology*, 81(2), pp. 458–464. Available at: <https://doi.org/10.1189/jlb.0806514>.

Bauer, S. *et al.* (2007b) "Proteinase 3 and CD177 are expressed on the plasma membrane of the same subset of neutrophils," *Journal of Leukocyte Biology*, 81(2). Available at: <https://doi.org/10.1189/jlb.0806514>.

Broemstrup, T. and Reuter, N. (2010a) "How does Proteinase 3 interact with lipid bilayers?," *Physical Chemistry Chemical Physics*, 12(27), pp. 7487–7496. Available at: <https://doi.org/10.1039/b924117e>.

Broemstrup, T. and Reuter, N. (2010b) "How does Proteinase 3 interact with lipid bilayers?," *Physical Chemistry Chemical Physics*, 12(27). Available at: <https://doi.org/10.1039/b924117e>.

Carla Guarino *et al.* (2023) "Constitutive and induced forms of membrane-bound proteinase 3 interact with antineutrophil cytoplasmic antibodies and promote immune activation of neutrophils," *Journal of Biological Chemistry*, 299(4). Available at: <https://doi.org/10.1016/j.jbc.2023.103072>.

Casal Moura, M. *et al.* (2023) "Activation of a Latent Epitope Causing Differential Binding of Antineutrophil Cytoplasmic Antibodies to Proteinase 3," *Arthritis and Rheumatology*, 75(5). Available at: <https://doi.org/10.1002/art.42418>.

Davis, J.A. *et al.* (1998) "Determination of primary amino acid sequence and unique three-dimensional structure of WGH1, a monoclonal human IgM antibody with anti-PR3 specificity," *Clinical Immunology and Immunopathology*, 89(1). Available at: <https://doi.org/10.1006/clin.1998.4582>.

Fujinaga, M. *et al.* (1996) "The crystal structure of PR3, a neutrophil serine proteinase antigen of Wegener's granulomatosis antibodies," *Journal of Molecular Biology*, 261(2). Available at: <https://doi.org/10.1006/jmbi.1996.0458>.

- Gabillet, J. *et al.* (2012) "Proteinase 3, the Autoantigen in Granulomatosis with Polyangiitis, Associates with Calreticulin on Apoptotic Neutrophils, Impairs Macrophage Phagocytosis, and Promotes Inflammation," *The Journal of Immunology*, 189(5). Available at: <https://doi.org/10.4049/jimmunol.1200600>.
- Van Der Geld, Y.M. *et al.* (2000) "Recombinant proteinase 3 produced in different expression systems: Recognition by anti-PR3 antibodies," *Journal of Immunological Methods*, 244(1–2). Available at: [https://doi.org/10.1016/S0022-1759\(00\)00261-1](https://doi.org/10.1016/S0022-1759(00)00261-1).
- Van Der Geld, Y.M. *et al.* (2002) "Expression of recombinant proteinase 3, the autoantigen in Wegener's granulomatosis, in insect cells," *Journal of Immunological Methods*, 264(1–2). Available at: [https://doi.org/10.1016/S0022-1759\(02\)00101-1](https://doi.org/10.1016/S0022-1759(02)00101-1).
- Van Der Geld, Y.M., Limburg, P.C. and Kallenberg, C.G.M. (1999) "Characterization of monoclonal antibodies to proteinase 3 (PR3) as candidate tools for epitope mapping of human anti-PR3 autoantibodies," *Clinical and Experimental Immunology*, 118(3). Available at: <https://doi.org/10.1046/j.1365-2249.1999.01079.x>.
- Goldmann, W.H., Niles, J.L. and Arnaout, M.A. (1999a) "Interaction of purified human proteinase 3 (PR3) with reconstituted lipid bilayers," *European Journal of Biochemistry*, 261(1), pp. 155–162. Available at: <https://doi.org/10.1046/j.1432-1327.1999.00259.x>.
- Goldmann, W.H., Niles, J.L. and Arnaout, M.A. (1999b) "Interaction of purified human proteinase 3 (PR3) with reconstituted lipid bilayers," *European Journal of Biochemistry*, 261(1). Available at: <https://doi.org/10.1046/j.1432-1327.1999.00259.x>.
- Griffith, M.E. *et al.* (2001) "Anti-neutrophil cytoplasmic antibodies (ANCA) from patients with systemic vasculitis recognize restricted epitopes of proteinase 3 involving the catalytic site," *Clinical and Experimental Immunology*, 123(1). Available at: <https://doi.org/10.1046/j.1365-2249.2001.01420.x>.
- Hajjar, E. *et al.* (2007) "Computational prediction of the binding site of proteinase 3 to the plasma membrane," *Proteins: Structure, Function, and Bioinformatics*, 71(4), pp. 1655–1669. Available at: <https://doi.org/10.1002/prot.21853>.
- Hajjar, E. *et al.* (2008) "Computational prediction of the binding site of proteinase 3 to the plasma membrane," *Proteins: Structure, Function and Genetics*, 71(4). Available at: <https://doi.org/10.1002/prot.21853>.
- Harmsen, M.C. *et al.* (1997) "Recombinant proteinase 3 (Wegener's antigen) expressed in *Pichia pastoris* is functionally active and is recognized by patient sera," *Clinical and Experimental Immunology*, 110(2). Available at: <https://doi.org/10.1111/j.1365-2249.1997.tb08325.x>.

Hinkofer, L.C. *et al.* (2013) "A monoclonal antibody (MCPR3-7) interfering with the activity of proteinase 3 by an allosteric mechanism," *Journal of Biological Chemistry*, 288(37). Available at: <https://doi.org/10.1074/jbc.M113.495770>.

Hu, N. *et al.* (2009a) "Coexpression of CD177 and membrane proteinase 3 on neutrophils in antineutrophil cytoplasmic autoantibody-associated systemic vasculitis: Anti-proteinase 3-mediated neutrophil activation is independent of the role of CD177-expressing neutrophils," *Arthritis and Rheumatism*, 60(5), pp. 1548–1556. Available at: <https://doi.org/10.1002/art.24442>.

Hu, N. *et al.* (2009b) "Coexpression of CD177 and membrane proteinase 3 on neutrophils in antineutrophil cytoplasmic autoantibody-associated systemic vasculitis: Anti-proteinase 3-mediated neutrophil activation is independent of the role of CD177-expressing neutrophils," *Arthritis and Rheumatism*, 60(5). Available at: <https://doi.org/10.1002/art.24442>.

Jerke, U. *et al.* (2017) "Characterization of the CD177 interaction with the ANCA antigen proteinase 3," *Scientific Reports*, 7. Available at: <https://doi.org/10.1038/srep43328>.

Kantari, C. *et al.* (2011) "Molecular analysis of the membrane insertion domain of proteinase 3, the Wegener's autoantigen, in RBL cells: implication for its pathogenic activity," *Journal of Leukocyte Biology*, 90(5). Available at: <https://doi.org/10.1189/jlb.1210695>.

Kelly, S. *et al.* (2024) "Isolation and characterisation of PR3-specific B cells and their immunoglobulin sequences," *Journal of Autoimmunity*, 142. Available at: <https://doi.org/10.1016/j.jaut.2023.103129>.

Kissel, K. *et al.* (2001) "Molecular basis of the neutrophil glycoprotein NB1 (CD177) involved in the pathogenesis of immune neutropenias and transfusion reactions," *European Journal of Immunology*, 31(5). Available at: [https://doi.org/10.1002/1521-4141\(200105\)31:5<1301::AID-IMMU1301>3.0.CO;2-J](https://doi.org/10.1002/1521-4141(200105)31:5<1301::AID-IMMU1301>3.0.CO;2-J).

Korkmaz, B. *et al.* (2008) "A hydrophobic patch on proteinase 3, the target of autoantibodies in Wegener granulomatosis, mediates membrane binding via NB1 receptors," *Journal of Biological Chemistry*, 283(51). Available at: <https://doi.org/10.1074/jbc.M806754200>.

Kuhl, A. *et al.* (2010) "Mapping of Conformational Epitopes on Human Proteinase 3, the Autoantigen of Wegener's Granulomatosis," *The Journal of Immunology*, 185(1). Available at: <https://doi.org/10.4049/jimmunol.0903887>.

Marino, S.F. *et al.* (2022) "Competitively disrupting the neutrophil-specific receptor-autoantigen CD177:proteinase 3 membrane complex reduces anti-PR3 antibody-induced neutrophil activation," *Journal of Biological Chemistry*, 298(3). Available at: <https://doi.org/10.1016/j.jbc.2022.101598>.

Pang, Y.P. *et al.* (2019) "Remote activation of a latent epitope in an autoantigen decoded with simulated b-factors," *Frontiers in Immunology*, 10(OCT). Available at: <https://doi.org/10.3389/fimmu.2019.02467>.

Peen, E. and Williams, R.C. (2000) "What you should know about PR3-ANCA. Structural aspects of antibodies to proteinase 3 (PR3).," *Arthritis research*, 2(4), pp. 255–9. Available at: <https://doi.org/10.1186/ar97>.

Robert F. Halenbeck *et al.* (1995) "Recombinant PR-3 and compositions thereof." United States: United States Patent and Trademark Office.

Sibilia, J. *et al.* (1997) "Structural analysis of human antibodies to proteinase 3 from patients with Wegener granulomatosis.," *The Journal of Immunology*, 159(2). Available at: <https://doi.org/10.4049/jimmunol.159.2.712>.

Silva, F. *et al.* (2010) "Discrimination and variable impact of ANCA binding to different surface epitopes on proteinase 3, the Wegener's autoantigen," *Journal of Autoimmunity*, 35(4). Available at: <https://doi.org/10.1016/j.jaut.2010.06.021>.

Specks, U. *et al.* (1996) "Recombinant human proteinase 3, the Wegener's autoantigen, expressed in HMC-1 cells is enzymatically active and recognized by c-ANCA," *FEBS Letters*, 390(3). Available at: [https://doi.org/10.1016/0014-5793\(96\)00669-2](https://doi.org/10.1016/0014-5793(96)00669-2).

Stummann, L. and Wiik, A. (1997) "A simple high yield procedure for purification of human proteinase 3, the main molecular target of cANCA," *Journal of Immunological Methods*, 206(1–2). Available at: [https://doi.org/10.1016/S0022-1759\(97\)00082-3](https://doi.org/10.1016/S0022-1759(97)00082-3).

Sun, J. *et al.* (1998) "A proportion of proteinase 3 (PR3)-specific anti-neutrophil cytoplasmic antibodies (ANCA) only react with PR3 after cleavage of its N-terminal activation dipeptide," *Clinical and Experimental Immunology*, 114(2). Available at: <https://doi.org/10.1046/j.1365-2249.1998.00730.x>.

del Toro, D. *et al.* (2020) "Structural Basis of Teneurin-Latrophilin Interaction in Repulsive Guidance of Migrating Neurons," *Cell*, 180(2). Available at: <https://doi.org/10.1016/j.cell.2019.12.014>.

Von Vietinghoff, S. *et al.* (2007a) "NB1 mediates surface expression of the ANCA antigen proteinase 3 on human neutrophils," *Blood*, 109(10), pp. 4487–4493. Available at: <https://doi.org/10.1182/blood-2006-10-055327>.

Von Vietinghoff, S. *et al.* (2007b) "NB1 mediates surface expression of the ANCA antigen proteinase 3 on human neutrophils," *Blood*, 109(10). Available at: <https://doi.org/10.1182/blood-2006-10-055327>.

Williams, R.C. *et al.* (1994) "Epitopes on proteinase-3 recognized by antibodies from patients with Wegener's granulomatosis.," *The Journal of Immunology*, 152(9). Available at: <https://doi.org/10.4049/jimmunol.152.9.4722>.

Dear Dr. Seiradake,

Thank you for the submission of your revised manuscript to our editorial offices. I have already forwarded to you the reports I received from the three referees that I asked to re-evaluate your study. Please find them again below.

After going through your preliminary point-by-points response (further revision plan), I have decided to proceed. Please address the remaining concerns of the referees in a final revised manuscript and/or in a final detailed point-by-point response, as indicated in your revision plan. Moreover, I have several editorial requests. Please also provide a p-b-p-response regarding these editorial requests with your final submission.

Editorial requests:

- Please add up to five keywords to the manuscript and order the manuscript sections like this, using only these names: Title page - Abstract (max. 175 words) - Keywords (up to five) - Introduction - Results - Discussion - Methods - Data availability section - Acknowledgements (please include here all the funding information) - Disclosure and Competing Interests Statement - References - Figure legends - Expanded View Figure legends

- We now use CRediT to specify the contributions of each author in the journal submission system. CRediT replaces the author contribution section. Please use the free text box to provide more detailed descriptions and do NOT provide your final manuscript text file with an author contributions section. See also our guide to authors (section 'Author contributions'): <https://link.springer.com/journal/44319/submission-guidelines#cms-Revised-submissions>

- Please add the email addresses of the 4 co-corresponding authors to the title page.

- The email for Kamel El Omari (kamel.omari@some.ox.ac.uk) bounced. The author should thus either be removed from the author list in the system and then added back using the new/current email address, or please send us the new/current email address and we will update the author's account.

- Please upload individual production quality figure files as .eps, .tif, .jpg (one file per figure), of main figures and EV figures. Please upload these as separate, individual files upon re-submission.

<https://link.springer.com/journal/44319/submission-guidelines#cms-Revised-submissions>

<https://media.springernature.com/original/springer-cms/rest/v1/content/27825798/data/v1>

- Please use our reference format:

<https://link.springer.com/journal/44319/submission-guidelines#cms-Reference-guidelines>

Please use 'et al' needs to be used after the 10th author name; DOIs should only be used for preprints and datasets that have not been published yet.

- Per journal policy, we do not allow 'data not shown', which is stated in the manuscript (page 9). All data referred to in the paper should be displayed in the main or Expanded View figures, or an Appendix. Thus, please add these data (or change the text accordingly if these data are not central to the study). See:

<https://link.springer.com/partners/embo-press/editorial-policies#Data%20deposition>

- Please check again that the number "n" for how many independent experiments were performed, their nature (biological versus technical replicates), the bars and error bars (e.g. SEM, SD) and the test used to calculate p-values is indicated in the respective figure legends (main, EV and Appendix figures). Please also check that all the p-values are explained in the legend, and that these fit to those shown in the figure. Please provide statistical testing where applicable. Please avoid the phrase 'independent experiment' but clearly state if these were biological or technical replicates. Please also indicate (e.g. with n.s.) if testing was performed, but the differences are not significant. In case n=2, please show the data as separate datapoints without error bars and statistics. See also:

<https://link.springer.com/journal/44319/submission-guidelines#cms-Figure-and-data-presentation>

If n<5, please show single datapoints for diagrams. Moreover:

- Please note that the exact p values are not provided in the legends of figures 3C, D; 4B, C; EV4 A.

- Please note that information related to n is missing in the legends of figures EV5 B-G

- Please note that the error bars are not defined in the legends of figures EV5 B-G

- Please note that the asterisk is not defined in the legend of figures EV1 I, EV2 B. This needs to be rectified.
- Please upload the datasets as Dataset EV1, Dataset EV2 and Dataset EV3 and make sure they are all called out like this. Please add a legend for each on the first TAB of the respective excel file.
- Please label the tables in the manuscript as Table 1 and Table 2. Or should these be EV tables? If yes, please remove them from the manuscript text file and upload the files separately (as Table EV1 and Table EV2). Please add legends for both tables on the first TAB of the respective excel sheets and also update any callouts.
- Please make sure that all the funding information is also entered into the online submission system and that it is complete and similar to the one in the acknowledgement section of the manuscript text file. Presently, the grants Wellcome Trust (202827/Z/16/Z and 226647/Z/22/Z) and the EMBO Young Investigator Programme; Chang Gung Memorial Hospital-Linkou, Taiwan (CMRPD1M0033) and the National Science and Technology Council (NSTC), Taiwan (NSTC-113-2918-I-182-001 and NSTC-113-2320- B-182-009) are missing in the submission system. Please check.
- Thanks for providing the source data. Please upload the source data as one folder per main figure, grouping together all the files (separate files for each panel) for this figure (and ZIPed together), and one folder for all the EV Figures, grouping together all the files for each EV Figure in separate folders (and ZIPed together). Please also make sure that all links to externally deposited source data are updated and present in the final Data Availability Section.
- Please also add a direct link for the deposited structural data to the Data Availability section.

In addition, I would need from you uploaded separately:

I look forward to seeing a new revised version of your manuscript as soon as possible.

Best,
 Achim Breiling
 Senior Editor
 EMBO Reports

 Referee #1:

The authors have addressed the input they have received very well.

 Referee #2:

1. Does this manuscript report a single key finding? YES

The authors report the X ray structures of the neutrophil protein CD177 alone and in complex with a modified neutrophil serine protease PR3; the wildtype PR3 is a known autoantigen.

2. Is the reported work of significance (YES), or does it describe a confirmatory finding or one that has already been documented using other methods or in other organisms etc (NO)?

The structures represent an important advance in the field of neutrophil biology and, particularly, in the study of PR3 focused autoimmune diseases. While the structures themselves are new, some results presented in the manuscript - though also presented as novel - are not, and proper attribution is still lacking.

3. Is it of general interest to the molecular biology community? YES

While the topic is rather specialized, new three dimensional structures are always of general interest to the wider MolBio community.

4. Is the single major finding robustly documented using independent lines of experimental evidence or is it really just a preliminary report requiring significant further data to become convincing, and thus more suited to a longer –format article.

NO. While the structures themselves are convincing (for what they are), the analyses that should accompany them are still lacking - there is ample previously published data that should be used in placing the structural information into the broader context. Unfortunately, the text still reads like "hype" rather than consisting of careful, reasoned argumentation.

Detailed comments:

My initial review of the manuscript by Zheng-Gérard et al. identified numerous faults and made clear that a number of claims made in the text were poorly argued, unfounded or simply incorrect. A thorough analysis of their new data - the X ray structures - in the context of previously published data was not present. Additionally, they presented the preparation of the monomeric PR3 construct used for their crystallization as a novel achievement although a monomeric PR3 had already been reported in more than one publication starting in 2017. A clear acknowledgement of this previously published data was not included in their manuscript. The manuscript also claimed that ELISA experiments using epitope tags for quantification of their antigens was "pioneered" (the authors' word) by them and, after testing the autoantibodies present in eight (8) ANCA patient samples, extrapolated their results to all of the (potentially millions) of other (GPA) ANCA sufferers, claiming discovery of "a major ANCA epitope" illuminated by their structure. These, and other, faults were flagged by this reviewer in a detailed treatment of the manuscript.

A revised manuscript has now been submitted. The authors, to their credit, have reworked substantial portions of the text and have indeed made it more readable. They have also included more citations in their text (some of which are unnecessary and/or incorrect), ostensibly as a reaction to the criticism that much previously published data was not taken into proper account in the first version (a point also made by the other reviewers). Unfortunately, the major criticisms levelled by this reviewer have still not been adequately addressed. I have taken the time to respond to each of the "point by point" responses of the authors to my original criticisms and include that text with my latest comments here. I believe this to be the clearest way to convey my point of view on each issue. The new comments are set off by arrows (>>>) in each case:

Referee #2

The manuscript (MS) by Zheng-Gérard, et al. reports the structure of the neutrophil PR3:CD177 complex as well as that of CD177 alone, thus providing two major pieces of data missing from the thorough characterization of the complex. This is a most welcome advance for the field and it is wonderful to finally see these structures that, together with the extensive biochemical data already in the literature, will be of great value in improving our understanding of neutrophil biology and for the development of treatments for PR3-ANCA diseases.

Authors: We thank the reviewer for this positive assessment.

It is therefore most unfortunate that a careful analysis of the structural data in light of the ample biochemical information already published is absent from the text. The MS is furthermore marred by examples of sweeping generalization that are neither necessary nor warranted, as well as statements that are quite misleadingly formulated or altogether inaccurate. Most peculiar indeed is the reporting of a mutagenesis scheme for preparing a monomeric PR3 - an effort occupying substantial space in the MS and presented as novel - although the "monomerization" of PR3 based on mutating the same positions had already been reported elsewhere; that study is actually referenced by the authors of the present MS but no discussion of that PR3 data is presented in the text. The inaccuracies, contradictions and omissions lead one to wonder whether the authors actually read many of the studies that they cite in their reference list; reading the text does not convey the impression that the authors are aware of data published by many groups over the last several decades. Despite the welcome advance provided by the structural data, this MS is not fit for publication anywhere in its present form.

Authors: We appreciate the feedback and have improved the main text accordingly (see points below), for example by discussing the work by Jerke et al (Jerke et al., 2017) at appropriate points in the text. Please note that the recombinant mutant PR3 we present was developed independently of other studies and that we started the project before the publication of the aforementioned paper. In contrast to Jerke et al. who performed alanine substitutions, we introduced mutations that result in an N-linked glycosylation site, rather than alanines. This significantly improves the solubility and homogeneity of the sample and was instrumental to being able to co-crystallise the protein with CD177, in our hands.

We have rewritten the relevant section in our results to relate our construct to the previously published mutant used by Jerke et al.:

"Using an established protein engineering approach that introduces an N-linked glycosylation site (del Toro et al., 2020; Akkermans et al., 2022), we mutated three residues within the PR3 hydrophobic surface patch: I221N-W222G-G223T (Figure EV 1F). A previously published mutant in which two of these residues were mutated to alanines (I221A-W222A) had also been reported to improve protein solubility (Jerke et al., 2017)."

In the discussion we write:

"The hydrophobic nature of wild type PR3, which is thought to be due to the "hydrophobic patch" on the protein surface (Korkmaz et al., 2008), has presented a challenge for the production and purification of this protein for structural studies. Over the last 30 years, protocols were established for the expression of recombinant PR3 in yeast (Harmsen et al., 1997), Sf9 insect cells (Fujinaga et al., 1996), human mast cell line-1 HMC-1 (Specks et al., 1996; Sun et al., 1998; Van Der Geld et al., 2000) and HEK293 cells (Sun et al., 1998; Van Der Geld et al., 2000; Korkmaz et al., 2008; Jerke et al., 2017). Several of these purification trials have successfully used detergents such as 0.1% Triton X-100 (Robert F. Halenbeck et al., 1995) 1% β -OG (Stummann and Wiik, 1997), 0.1% Tween-80 (Van Der Geld et al., 2002) or 0.02% DDM (Jerke et al., 2017) to mitigate protein aggregation, although it is recognised that these detergents can hamper biological experiments and crystallisation trials. Mutagenesis of the "hydrophobic patch" residues I221 and W222 to alanines resulted in a promising construct with improved solubility (Jerke et al., 2017). Our work was catalysed by introducing an N-linked glycosylation site in position 221 (PR3rec), which made production and purification from HEK293 cells and successful co-crystallisation with CD177 receptor without the use of detergent possible. "

>>>

>>>

It may be that the authors' efforts began before the publication of Jerke, et al., but this is irrelevant: independently achieving the same known result (even by different means) does not absolve one from properly acknowledging previously published examples. There is certainly no justification for presenting such a result as unique. Is the claim that the authors' approach "significantly improves the solubility and homogeneity of the sample" compared to that of Jerke, et al. (which is how the text reads) substantiated by real results - i.e. did the authors also prepare the construct of Jerke et al. and compare the two? If so (which would allow one to make the statement quoted here) why is this data not presented?

In the revised text, the authors write that the mutations made by Jerke et al. were "reported to improve protein solubility". This is only partly true - the mutations were reported to result in PR3 monomers. The authors have (again) intentionally left this clarification out of their revision, despite the criticism levelled in the first review. In the discussion, they refer to the mutant from Jerke et al. as "a promising construct". What do they mean by this statement? The data published by Jerke, et al. demonstrated that the resulting mutant was monomeric. The successful purification of that mutant from HEK_293 cells and experimental data with that mutant were reported in the same paper. The authors have not demonstrated that their construct has properties (besides extra shrubbery) that haven't already been demonstrated by that in Jerke, et al., but make a substantial effort to imply this in their choice of phrasing (which this reviewer finds unacceptable). One is not entitled to make claims - particularly in scientific publications - that have not been substantiated by experimental observations/results.

The authors' unqualified comment that detergents like β -OG and DDM "can hamper...crystallization trials" is simply not correct (as a perusal of the PDB makes clear) - they are, indeed, sometimes essential for crystallization (and where are the citations that substantiate the authors' statement?).

>>>

>>>

A discussion of the major flaws in the MS follows here and all of these issues must be adequately addressed before the MS can be submitted to any journal with a reasonable expectation of acceptance.

1) We are long past the days when the generation of a structure alone, however competently done (as the current example appears to be), suffices for a publication (unless combined with some new technical advance or other novelty that was necessary for achieving the structure). It is here emphasized that it is very exciting to now have these structures and their thoughtful analysis will be valuable for understanding the biology of ANCA-related diseases and also for developing treatments. But just such a thoughtful analysis - with confirmatory experimental data - is, unfortunately, missing from the MS. A number of experiments with the complex are reported but there isn't any meaningful analysis in light of previously published predictions - for example, Kantari et al. (reference 31 in the MS) made the "hydrophobic patch" responsible for the attachment of PR3 to the outer neutrophil membrane and Korkmaz, et al. (reference 81 in the MS) claimed that the "hydrophobic patch" was responsible for the interaction with CD177 - does the structural data (which is what is new) provide any insight into the correctness of these suppositions?

A rather detailed model for the complex based on biochemical data was proposed in 2017 (Jerke, et al. - reference number 19 in the MS) - how does the presently reported complex structure compare with those predictions from the biochemical characterization? These examples - and more - should make up a substantial portion of the discussion section of the current manuscript but are entirely absent (although the primary sources are in the reference list). The structures alone are the only new information.

Authors: We thank the reviewer for this feedback and appreciation of the structural results. We have improved and expanded our results section, supplemental figures and discussion section to the published definition of the "hydrophobic patch":

Results: "The centre of the binding surface contains hydrophobic residues (F180, F181, F190 and L228 from PR3, and L117, P118 and W120 from CD177) (Figure 2D-F). These residues are located within the previously described "hydrophobic patch" of PR3 (Korkmaz et al., 2008) (Figure EV3 panel C)."

Figures: see Figure EV3 panel C.

Discussion: "The structure shows that the PR3-CD177 binding interface overlaps, at least in part, with the 'hydrophobic patch' (Korkmaz et al., 2008; Jerke et al., 2017) (Figure EV3 C), validating previous studies suggesting that mutations in this region reduce CD177 binding (Jerke et al., 2017)."

>>>

>>>

The citation of Jerke et al. here is misleading and potentially incorrect. What Jerke et al. suggested was that the "reduce(d) CD177 binding" of their mutant could be due to its monomeric form in comparison to wt which they clearly showed was multimeric - i.e. the increased affinity for CD177 by wtPR3 could be due to avidity effects. The quote from the paper is: "The higher affinity measured for wtPR3 binding is likely due to multivalent interaction of wtPR3 multimers with immobilized CD177." This reviewer additionally notes that the positions mutated by Jerke et al. are NOT in the PR3:CD177 binding interface reported by the authors. The initial statement from this reviewer still stands: in this manuscript, the structures themselves are the only new information. No new meaningful analysis has been incorporated into the revision.

>>>

>>>

Please note that, given the focus of this manuscript on how PR3 interacts with its receptor CD177 and the importance of the binding interface as an epitope targeted by PR3-ANCA, it is beyond the scope of this report to analyse how the "hydrophobic patch" as defined by others may function to mediate PR3 attachment to cell membranes etc.

2) The authors give substantial space to their mutagenesis effort to achieve a monodisperse PR3 construct and report finding mutations that render the enzyme monomeric. Quoting from their text: "The study was possible using a site-directed mutagenesis screen to identify stable Pr3 variants that are readily expressed and purified from HEK293 cultures." Perhaps it is an oversight (but then a very serious one) that a successful monomerization effort was already published in 2017 (reference 19 in the MS): the mutated Ile - Trp (i - i+1) pair was shown to render PR3 monomeric and the affinity of that monomer for CD177 by SPR was also reported. The data in the current MS are therefore not new as the authors claim (using another residue numbering scheme does not alter the sequence itself) and this previous work is not acknowledged in their text (although the paper in question does appear in the reference list). This reviewer will stop short of labeling this an ethical problem, but the "oversight" must be corrected no matter where the next version of the MS is to be published.

Authors: We are grateful for the reviewers suggestion to discuss the Jerke et al. paper in more depth. As noted above, our mutant (PR3rec) is not the same as the one published in Jerke et al., and we have now added a clear description of how the two constructs compare in the relevant sections (see our response to previous comments further up). Regarding the numbering scheme, we have followed the scheme that is also used by the UniProt database, where the first Met residue in a sequence is numbered 1.

>>>

>>>

This reviewer did not claim that the PR3 mutant reported by the authors is the same as that reported in Jerke, et al. - it was stated that the authors' rather more involved mutagenesis efforts produced the same superficial result as had been published by others 8 years prior. It was also noted that in the present manuscript the outcome of the mutagenesis was presented as a unique/novel development - i.e. that something new had been achieved. This treatment was either so poorly written that it gave this - demonstrably false - impression or it was an active attempt to mislead the reader. On the first pass, the former possibility was suggested; but since the "clear description" now claimed to have been added is nothing of the sort, the second possibility must be seriously considered.

On this basis alone, in the opinion of this reviewer, the manuscript should be rejected by any journal to which it is submitted.

Arriving at the same result by another method is surely of interest and worthy of reporting - but that can in no way justify an effort to ignore or play down the initially published result - and no data are presented in the current revised manuscript showing that the 'new' scheme for producing soluble and monomeric PR3 is superior to what was published 8 years ago. The manner of this presentation - and the recalcitrance in correcting the initial error - combined with the refusal to clearly correct other misleading statements in the text (see the point below concerning patient ANCA epitopes) falls far short of the standards of honest discourse.

>>>

>>>

3) This reviewer feels that a comment on style here is also necessary: apparently also gone are the days when formulations in scientific reports minimized use of the first person; while this can sometimes not be gracefully avoided, the practice of introducing basic (if not trivial) results in this MS with overly self-promoting phrases such as "We pioneered..." (page 11), as well as numerous other "We" statements scattered throughout, should be and, considering that much of what is described in the MS was already reported by others and/or not sufficiently acknowledged as such, maximized the discomfort felt when reading the text. These deficiencies, as well as the grammar mistakes and confusing formulations (some, but not all, of which are addressed explicitly below) must be corrected before the next submission, to whichever journal that may be.

Authors: this is an interesting point and we refer to the editor for guidance. Many journals ask for an active style ("we did this...", "we did that...") rather than passive voice ("the sample was subjected...", "it was thought that..."). We will adapt to whichever style is preferred by the editors. We will also do our very best to weed out any typos etc.

>>>

>>>

The authors' writing style is, of course, up to them and the Editor(s). But since "We pioneered..." remains a demonstrably false statement, it should, in my view, be removed from the text.

>>>

>>>

4) There are a number of statements throughout the text that are either partly or wholly inaccurate; some examples follow (this list is NOT exhaustive), with comments from this reviewer in bold:

From the Abstract, page 1:

"Granulomatosis with polyangiitis is a life-threatening systemic vasculitis, characterised by antineutrophil cytoplasmic autoantibodies (ANCA) against proteinase 3 (Pr3) expressed on the surface of neutrophils. Most cell surface Pr3 is bound to the receptor CD177..."

These first two sentences of the paper are quite garbled - the first reads as if the ANCA themselves are "expressed" on neutrophils...which of course is not the case; furthermore, PR3 is found on the extracellular surface of all neutrophils, even those not displaying CD177...and even on neutrophils from persons who are genetically negative for CD177, i.e. whose neutrophils completely lack CD177.

Authors: this has been adjusted to read 'Granulomatosis with polyangiitis is a life-threatening systemic vasculitis, characterised by anti-neutrophil cytoplasmic autoantibodies (ANCA) against proteinase 3 (PR3), a protease expressed on the surface of neutrophils.'

>>>

>>>

Fine.

>>>

>>>

From the Abstract, page 1:

"We describe an extensive, mainly hydrophobic binding interface between Pr3 and CD177..."

This is an unnecessarily and incorrectly qualified statement - the authors report the buried surface area in the complex to be " $\sim 655 \text{ \AA}^2$ " - a value rather at the low end of the buried surface area spectrum that extends, on average, between 1500 and 3000 \AA^2 (see for example 10.1073/pnas.93.1.13); How have the authors decided that the PR3:CD177 interaction is "extensive"?

Authors:

We have removed the word 'extensive' from that sentence.

>>>

>>>

Fine.

>>>

>>>

From the Abstract, page 2:

"Using a panel of Pr3-ANCA positive patient samples, we show that CD177 inhibits Pr3-ANCA binding to Pr3".

It is obvious that this statement on its face cannot be accurate. Certainly the binding of some ANCAs is blocked by the presence of CD177 (as the authors show with their experiments using patient samples) but, as reported later in the MS and has been shown previously in multiple studies (see for example J Immunol. 1994 May 1;152(9):4722-37, J Autoimmun 2010 35, 299-308 (reference 78 in the MS) and also reference 20 in the MS, among others) ANCA epitopes are also found outside of the CD177 binding site and some ANCAs can bind to PR3 that is complexed with CD177.

Authors: We have revised this sentence to say 'Using a panel of PR3-ANCA positive patient samples, we show that a significant proportion of ANCAs target the CD177-binding site of PR3.'

>>>

>>>

The statement is still misleading and, when the actual data are considered: false (for all the reasons cited in the initial review). The fact that a handful of tested patient samples interfere with CD177 binding cannot itself be trumpeted as a significant new finding. Extrapolating this handful of results so concretely to include the entirety of ANCA patient samples is even less warranted. And, considering that the subheading of the respective Results section still reads: "Most GPA patient samples harbour autoantibodies that target the CD177-binding site of PR3", it is clear that the authors still do not acknowledge the criticism. The results shown also do not indicate that these ANCAs "target the CD177 binding site" - steric effects (which the authors invoke to avoid explaining why PR3 activity is reduced in the presence of CD177 although the active site is not occluded in the complex structure, see below) could easily explain the interference with CD177 binding; the authors write that some ANCAs "substantially reduce" CD177 binding - none completely prevented binding. In this regard, the title of the manuscript is also invalid - no ANCA epitope has been determined in this work. It has only been shown with the ELISAs that some ANCAs can interfere with CD177 binding to PR3 - this is not the same thing as saying that the binding epitopes of these ANCAs have been "revealed" by the structure(s). And no justification is provided (or can be gleaned from the data) for referring to these theoretical epitopes as "major".

>>>

>>>

The authors also write on page 5: "We discover that most Pr3-ANCA positive samples compete with the interaction between Pr3 and CD177..." and furthermore on page 9: "Most GPA patient samples harbour autoantibodies that target the CD177-binding site of Pr3" because, in their ELISA, 7 of 8 samples appear to bind an epitope coinciding with the CD177 binding site (their conclusion). Now, even accepting the authors' own - not referenced!! - "estimate" of the prevalence of GPA (Introduction, page 2) of "2 - 150 per million individuals", it is clear that these statements cannot be true. Taking the lower limit of this - again not referenced - estimate (and since PR3-ANCA are involved in other diseases and the number of undiagnosed cases of PR3-ANCA diseases is thought to be substantial, the authors' estimate is clearly extremely low), the number of patients must number in the hundreds of thousands/millions, depending on where the authors' "estimate" applies (in England? In Taiwan?? Worldwide?). To have tested samples from <0,001% of all patients (based on the absolute - and incorrect - minimum estimate) and then drawing a firm conclusion about them all cannot be taken seriously.

Authors:

We are of course only focusing on the panel of GPA patient samples presented in this study therefore we have adjusted this sentence to make it clearer: "We discover that most of the PR3-ANCA positive samples tested here target the CD177-binding site of PR3 and display reduced binding to PR3 protein where the CD177-binding site is masked." (page 5).

>>>

>>>

Modifying one sentence as an attempt to satisfy a criticism while leaving the remaining context (which was the target of the criticism) unchanged comes across here as disingenuous. The authors may have adequately qualified this particular statement (only after being called out on the point) but the related statements in the text are not so carefully worded. See for example the comment above concerning the Results subheading. The criticism by this reviewer stands, as does, apparently, its rejection by the authors.

>>>

>>>

From the Introduction, page 3:

"...although some PR3 appears to bind directly to cell membrane components".

This is a perplexingly misleading statement- the word "appears" here is simply inaccurate; there is ample published evidence, over many years, demonstrating that PR3 is present on the neutrophil cell surface - even CD177 negative individuals have mPR3 on their neutrophils. This surface interaction has also been repeatedly shown to occur on non-myeloid cells when incubated with purified PR3; again, several of these studies are to be found in the reference list of this MS! It is perhaps the case - and then also perfectly acceptable - that the authors do not agree with all of this published data. But then this should be openly discussed and substantiated and not simply called into question via such confusing statements.

Authors: We have revised the text and it now reads "Most extracellular PR3 is presented on the neutrophil cell surface in complex with its receptor CD177 (also named HNA-2a or NB1) (Bauer et al., 2007b; Von Vietinghoff et al., 2007b; Abdgawad et al., 2010b), although PR3 can also bind directly to cell membrane components (Goldmann, Niles and Arnaout, 1999b; Hajjar et al., 2008; Hu et al., 2009b; Broemstrup and Reuter, 2010b; Kantari et al., 2011). "(page 3).

As mentioned above, this aspect of PR3 biology is not a major focus of the paper and we do not wish to prove or disprove previous work.

>>>

>>>

This reviewer did not claim the topic to be a "major focus of the paper" - but when a statement is made, it should be clear and properly referenced. No excuses.

And the modified sentence is still incorrect - how can "most" mPR3 be in complex with CD177 in individuals who are genetically CD177 negative?

>>>

>>>

From the Introduction, page 4:

"It is thought that this response is at least in part due to Pr3-ANCA-mediated recruitment and activation of Fcg receptors (37,44,45)".

This statement refers to the involvement of Fc receptor recruitment for the PR3-ANCA stimulated respiratory burst response of neutrophils...this was substantiated also in reference 20 in the MS but has not been included in the citation.

Authors : we would have been happy to cite this paper, but in the process of streamlining the paper to make it more focused and reader-friendly (in response to other comments), we have removed this section from the introduction. As the data presented in our paper is not focused on the Fc-gamma receptor, nor involves experiments with neutrophils, we think that this paragraph is not strictly necessary.

>>>

>>>

Fine.

>>>

>>>

From the Introduction, page 5:

"The study was possible using a site-directed mutagenesis screen to identify stable Pr3 variants that are readily expressed and purified from HEK293 cultures".

As mentioned above, the mutation of the residues producing the monomeric and "stable" PR3 variant - which had been initiated by the observation that wtPR3 forms an array of higher order aggregates in solution at physiological salt concentrations - was previously fully described (reference 19 in the MS) along with measurement of its binding affinity to CD177 and an evaluation of the relevance of the mutations with regard to the binding interface. The current MS claims a modified approach (including adding a glycosylation) at the identical positions to achieve the same effect that the authors of reference 19 managed with alanine substitutions. It is of note here that, while a PR3 monomer peak is visible in the size exclusion data shown in Fig. S1 f, the mutagenesis apparently did not result in an exclusively monomeric species as misleadingly claimed in the text ("...is expressed at high levels yielding stable and monodisperse protein samples") as it did for the authors of reference 19 . (If the larger species in the chromatogram of Fig. S1 f are NOT higher order PR3 aggregates, the authors should convincingly show this, for example

via Western blot). It is curious indeed that all of these previously and quite explicitly reported results are presented in the current MS as original with no reference to the older study that is actually cited several times for other reasons in the text. This is a very serious error and raises a number of questions as to how it could have occurred.

Authors: Thank you for this helpful comment. We now updated the relevant supplemental figure legend (Fig. EV1 panel G) to include a distinct description of the SDS-PAGE gel (Coomassie) and anti-His western blot (a-his). The originally submitted figure showed one of the first purifications attempted in the lab where the higher band at 60 kDa in the western blot -but not on the SDS-PAGE gel- accounts for a histidine-rich impurity that is sometimes co-purified in small amounts, depending on the quality of the immobilised metal ion affinity chromatography (IMAC) experiment. We see this sometimes, and for different proteins, produced in our system. This impurity, if present in tiny amounts, typically does not impact our crystallisation trials, but we do make sure to remove it by thorough wash for any functional studies, such as ELISA. To reflect this, we now show the SEC chromatogram, SDS-PAGE and western blot analysis from a later, cleaner purification of PR3rec, see Figure EV1 panel G.

>>>

>>>

It is noted that the authors pointedly refuse to comment on the lack of attribution mentioned in the initial criticism and have not corrected the problem.

>>>

>>>

From the Introduction, page 3:

" CD177 (also named HNA-2a or NB1) is a ~50 kDa glycosylphosphatidylinositol (GPI)-anchored surface glycoprotein".

First of all, although one assumes that the CD177 homolog in question is the human one, this reviewer could find no indication of this - nor the association of the word "human" directly linked to CD177 - anywhere in the main body of the text. Fortunately the authors did include the Uniprot accession number in the Methods section; but where does their claim of "~50 kDa" come from? The full length protein sequence - that includes an N-terminal signal sequence not present in the final active form of the molecule - is calculated to be 46,4 kDa; the molecule that the authors crystallized and used for their other experiments is calculated to be 41,3 kDa...how are either of these values "~50 kDa"?? While these may seem like minor points, they are rather indicative of the lack of rigor that characterizes much of the MS - when unambiguous, established facts cannot be faithfully repeated, what is one to think of the "new" data presented and the interpretations thereof? A further issue is the repeated use of the tilde (~) to report measured values - where are the decimal places and standard error values?

Authors: Indeed the reviewer is right that the predicted protein mass of human CD177 is less than 50kDa, however, the protein includes 3 predicted N-linked glycosylation sites, each accounting for an unknown size glycan . A previous mass spectrometry study (Kissel et al., 2001) found native glycosylated CD177 to be 50.556 kDa. Our use of "~50 kDa ... glycoprotein" reflects the findings in this previous study. We have now added the reference for clarity. (page 3)

>>>

>>>

The refusal of the authors to answer direct criticisms is perplexing; what was the mass of the actual molecule that was purified and crystallized?

>>>

>>>

From the Results, page 6:

"Surface plasmon resonance (SPR) and SEC-MALS experiments confirmed that Pr3rec binds CD177ecto with high affinity (Kd ~ 10 nM)".

Why is the affinity reported in such an imprecise way? Where are the decimal places and error calculations?

Authors: We have now included the standard deviation values in the main text as well as in Figure EV 2F.

>>>

>>>

Fine.

>>>
>>>

From the Results, page 7:

"The data show that CD177 binding does not occlude the Pr3 catalytic site, likely leaving it available for substrates".

While this statement - strictly speaking - may not be false, it is devoid of meaning. The issue is not whether the catalytic site is "available for substrates", it is whether enzymatic activity is affected upon PR3:CD177 complex formation (which could happen without a direct occlusion of the active site). Studies cited in the reference list of this MS have reported inhibition of PR3 activity in the presence of CD177 - in vitro and in vivo. Conformational changes occurring in multiple serine proteases have also been demonstrated. Incomprehensibly, none of this is included in discussion of the structures or mentioned in the text at all.

Authors: We now provide a discussion of this point: "

Previous work suggested a negative effect of CD177 on PR3 activity: the addition of purified CD177 was shown to inhibit nPR3 activity in vitro, in neutrophil degranulation experiments, and at the surface of neutrophils or CD177-expressing HEK cells (Jerke et al., 2017). Here we show that CD177-binding does not directly occlude the PR3 active site, nor does it perturb the PR3 structure compared to unliganded PR3 (Figure EV3 B). Therefore the mechanism of inhibition is likely more subtle, for example CD177 could sterically affect substrate binding away from the active site ." (page 11)

>>>
>>>

It is not clear to this reviewer what the statement "...CD177 could sterically affect substrate binding away from the active site ." intends to convey - do PR3 substrates bind "away from the active site"? What does this statement mean? (One could also ask: why is some manner of steric effect invoked here but when the exact same argument could be used for the inhibition of CD177 binding by ANCAs, the authors rather call that 'targeting of the CD177 epitope'?). Jerke, et al. are not the only investigators who have shown differences in PR3 activity depending on its disposition (even on neutrophils)...others have invoked different conformations of PR3 to account for differences in activity between assayed pools of PR3 on neutrophils. Further underlining the lack of a thoughtful analysis of the data in this manuscript, the authors - although now clearly stating that their changes to PR3 could conceivably affect conformational changes necessary for the enzyme's function - claim simply that "Our mutants were designed such as to minimise the predicted disturbance of the PR3 fold." - but this says nothing about possible disturbance of potential conformational changes. These are mentioned in the context of ANCA binding but not with reference to enzymatic activity. In their mentions of cited reports of decreased wtPR3 activity in the presence of CD177, the authors fail to emphasize (or realize?) that their construct is decidedly not wildtype - in fact, the complex structure omits half of CD177 and the authors fail to discuss how the multimerization of wtPR3 plays into complex formation in vivo. This deficiency is carried over to their fanciful model (shown in Figure 5 and still almost certainly incorrect) that fuses the full length CD177 structure with that of their artificial PR3 construct complex structure. What effect does PR3 multimerization have in vivo? Where is this depicted in this "model"? And why are the two structures fused together in the "model" - where is the evidence that the two lobes of CD177 remain identically oriented with or without wtPR3? The authors seem also to have failed to consider that the influence of their PR3 modifications on conformational flexibility may underlie the incongruities they report when using different commercial screens for PR3 detection. All of these points are low hanging fruit but none of them - either in the first or in this second incarnation of the manuscript - are elaborated.

>>>
>>>

The Discussion section is comprised mostly of meandering filler instead of a thoughtful evaluation of the established biochemical data from many other groups in light of the only novelty in the MS, the structures of CD177 and the PR3:CD177 complex. The authors' own statement on page 11: "...the molecular architecture of Pr3 surface presentation also impacts Pr3-ANCA interactions and therefore warrants investigation" is not at all effectively followed up on in the remainder of the text. Based on the observation of wtPR3 aggregation alone (again, reported in the MS as a novelty although already published years ago), the "model" of the PR3:CD177 complex/ANCA interaction presented in Fig. 5 is almost certainly - completely - wrong. It is also unclear to which "hypothesis" is referred on page 12 - the issue of where a handful of patient ANCAs may bind PR3 has already been mentioned above. The 'theorizing' comprising the bulk of pages 12 and 13 could be deleted in its entirety with no deleterious effect at all on the MS. The discussion of the ELISA data with wt and mutated PR3 - which makes no mention of the "...faster detection..." observed with the mutant that is slipped into the legend of Figure 3D - provides no new insights and somehow also does not include the (not) novel observation of wtPR3 aggregation. Even the suggestions of better diagnostics and potential therapies based on knowledge of the complex and PR3 mutations is old news; there is nothing novel in any of it, all of these points have been suggested/discussed in work by others - as a reading of the cited literature and even a cursory perusal of the patent databases make clear.

Authors: We have reworked our discussion, for example, including previous biochemical analysis done to reduce PR3

aggregation. We also replaced the "structure-based" views of CD177 and PR3 in Figure 5 with more abstract representations to avoid any confusion with regards to the detailed molecular mechanisms of ANCA-PR3 interaction. In response to the comments, we now provide a better context for the section on the future development of diagnostics and therapies. We maintain that our work is novel not just with regards to the structural and biophysical work (as recognised by all three reviewers), but also with regards to the ELISA study. In contrast to previous studies, we are able to compare the binding of patient samples to different PR3 variants (or complex), using an anti-His. We believe that the newly revised discussion now provides a balanced view of previous and novel results, within the limitations of a concise manuscript.

>>>

>>>

The authors write here: "We also replaced the "structure-based" views of CD177 and PR3 in Figure 5 with more abstract representations to avoid any confusion with regards to the detailed molecular mechanisms of ANCA-PR3 interaction."

To which "detailed molecular mechanisms" do they refer? Where - at all - in this manuscript is there any treatment of "detailed molecular mechanisms"? There is no meaningful consideration of the properties of wtPR3, an unsubstantiated assumption that the CD177 alone and the truncated CD177:PR3rec complex structures are interchangeable with each other and with their in vivo counterparts and no results that clearly map onto the in vivo relevance of the native complex (except some differential antibody binding with no insights into mechanisms).

The authors "maintain that our work is novel" but there is no demonstration that they have developed or invented anything unique. The identification of epitopes based on the structure is claimed but there is no description of these epitopes. ELISA studies are not novel and the ones reported in the manuscript are nothing out of the ordinary. Solid biochemistry is just that and doesn't need to be advertised as being "pioneering" or "novel". There is entirely too much hype in the text for so little thoughtful analysis.

Additional comments:

- On page 4 of the manuscript, the authors write: "The precise mechanism on PR3-CD177 interaction remains to be shown." - grammar problems aside, the statement is accurate; one must therefore ask: what then was the topic of this manuscript supposed to be?

- Further down on page 4, the authors state that neutrophils with more mPR3 are more strongly activated by ANCAs than neutrophils with low mPR3 levels - but their citation is incorrect; the correct reference for that data is:

Adrian Schreiber, Friedrich C Luft, Ralph Kettritz. *Kidney Int.* 2004 Jun;65(6):2172-83. doi: 10.1111/j.1523-1755.2004.00640.x.

- The authors mark the binding site of antibody CLB12.8 in figure EV1E and state in the figure legend that this antibody is "mentioned in this report"...but this reviewer is unable to find any mention of this antibody anywhere else in the text (perhaps the authors have spent too much time looking at Fig. 3g in Jerke, et al.).

>>>

>>>

Referee #3:

The authors were able to address most of the questions posed during revision, which is greatly appreciated. However, it would be important to include, in Figure 4B, plasma from healthy individuals as a control.

Additionally, the authors should provide a justification for Figure 5. It is unclear how ANCA can remove PR3 from the neutrophil surface. Presumably, ANCA would bind to PR3 present on the neutrophil surface. Normally, PR3 is secreted upon neutrophil activation, and in the current study, the authors do not present or cite any data demonstrating neutrophil activation followed by PR3 release.

A correction is also needed for Figure 3D: the figure legend and figure do not correspond to the text. Specifically, the text states: "We show that PR3^{rec} is generally more effective in detecting PR3-ANCA in representative PR3-ANCA-positive plasma samples compared to 14 nPR3 (Figure 3D)," but the figure does not include nPR3.

Point-by-point response

Referee #1:

The authors have addressed the input they have received very well.

Authors: we are very grateful for the reviewer's assessment.

Referee #3

The authors were able to address most of the questions posed during revision, which is greatly appreciated. However, it would be important to include, in Figure 4B, plasma from healthy individuals as a control.

Authors: we have now added the graphs for the MPO-ANCA patient control (PCtrl: plasma and IgG) and a healthy control plasma (HCtrl) to main Figure 4 panel C for comparison. We had previously put this data into the corresponding supplemental figure but we agree that is important for readers to be able to compare these in a main figure. Note that the results for most conditions are very low or zero (as indicated). To make space for these graphs, we have moved the previous Figure 4C graphs to main Figure 5A.

Additionally, the authors should provide a justification for Figure 5. It is unclear how ANCA can remove PR3 from the neutrophil surface. Presumably, ANCA would bind to PR3 present on the neutrophil surface. Normally, PR3 is secreted upon neutrophil activation, and in the current study, the authors do not present or cite any data demonstrating neutrophil activation followed by PR3 release.

Authors: we thank the reviewer for their helpful feedback on this figure, we have now decided to show a summary diagram of our ELISA data instead of speculating whether ANCA are able to remove PR3 from CD177 on the neutrophil surface.

A correction is also needed for Figure 3D: the figure legend and figure do not correspond to the text. Specifically, the text states: "We show that PR3^{rec} is generally more effective in detecting PR3-ANCA in representative PR3-ANCA-positive plasma samples compared to 14 nPR3 (Figure 3D)," but the figure does not include nPR3.

Authors: thank you for spotting this error, we have now amended the text to reference the correct figure: 'We show that PR3^{rec} is more effective in detecting PR3-ANCA in representative PR3-ANCA-positive plasma samples compared to nPR3 (Figure EV4B)'.

Referee #2:

Authors (general remarks): in our feedback, we have striven to respond to those aspects of the review that deal with the content of the manuscript. We are grateful for these comments. However, we feel that the sarcasm and speculation regards our intentions is out of place. We are sorry that the reviewer feels this way about our paper and rather than pointing this out throughout our responses, may we please assure him/her that our intention has always been to produce a high-quality manuscript and to reference previous work adequately.

1. Does this manuscript report a single key finding? YES

The authors report the X ray structures of the neutrophil protein CD177 alone and in complex with a modified neutrophil serine protease PR3; the wildtype PR3 is a known autoantigen.

2. Is the reported work of significance (YES), or does it describe a confirmatory finding or one that has already been documented using other methods or in other organisms etc (NO)?

The structures represent an important advance in the field of neutrophil biology and, particularly, in the study of PR3 focused autoimmune diseases. While the structures themselves are new, some results presented in the manuscript - though also presented as novel - are not, and proper attribution is still lacking.

3. Is it of general interest to the molecular biology community? YES

While the topic is rather specialized, new three dimensional structures are always of general interest to the wider MolBio community.

4. Is the single major finding robustly documented using independent lines of experimental evidence or is it really just a preliminary report requiring significant further data to become convincing, and thus more suited to a longer format article.

NO. While the structures themselves are convincing (for what they are), the analyses that should accompany them are still lacking - there is ample previously published data that should be used in placing the structural information into the broader context.

Unfortunately, the text still reads like "hype" rather than consisting of careful, reasoned argumentation.

Authors: please see below for detailed responses to these comments.

Detailed comments:

My initial review of the manuscript by Zheng-Gérard et al. identified numerous faults and

made clear that a number of claims made in the text were poorly argued, unfounded or simply incorrect. A thorough analysis of their new data - the X ray structures - in the context of previously published data was not present. Additionally, they presented the preparation of the monomeric PR3 construct used for their crystallization as a novel achievement although a monomeric PR3 had already been reported in more than one publication starting in 2017. A clear acknowledgement of this previously published data was not included in their manuscript. The manuscript also claimed that ELISA experiments using epitope tags for quantification of their antigens was "pioneered" (the authors' word) by them and, after testing the autoantibodies present in eight (8) ANCA patient samples, extrapolated their results to all of the (potentially millions) of other (GPA) ANCA sufferers, claiming discovery of "a major ANCA epitope" illuminated by their structure. These, and other, faults were flagged by this reviewer in a detailed treatment of the manuscript.

A revised manuscript has now been submitted. The authors, to their credit, have reworked substantial portions of the text and have indeed made it more readable. They have also included more citations in their text (some of which are unnecessary and/or incorrect), ostensibly as a reaction to the criticism that much previously published data was not taken into proper account in the first version (a point also made by the other reviewers). Unfortunately, the major criticisms levelled by this reviewer have still not been adequately addressed. I have taken the time to respond to each of the "point by point" responses of the authors to my original criticisms and include that text with my latest comments here. I believe this to be the clearest way to convey my point of view on each issue. The new comments are set off by arrows (>>>) in each case:

The manuscript (MS) by Zheng-Gérard, et al. reports the structure of the neutrophil PR3:CD177 complex as well as that of CD177 alone, thus providing two major pieces of data missing from the thorough characterization of the complex. This is a most welcome advance for the field and it is wonderful to finally see these structures that, together with the extensive biochemical data already in the literature, will be of great value in improving our understanding of neutrophil biology and for the development of treatments for PR3-ANCA diseases.

Authors (revision round1): We thank the reviewer for this positive assessment.

It is therefore most unfortunate that a careful analysis of the structural data in light of the ample biochemical information already published is absent from the text. The MS is furthermore marred by examples of sweeping generalization that are neither necessary nor warranted, as well as statements that are quite misleadingly formulated or altogether inaccurate. Most peculiar indeed is the reporting of a mutagenesis scheme for preparing a monomeric PR3 - an effort occupying substantial space in the MS and presented as novel - although the "monomerization" of PR3 based on mutating the same positions had already been reported elsewhere; that study is actually referenced by the authors of the present MS but no discussion of that PR3 data is presented in the text. The inaccuracies, contradictions and omissions lead one to wonder whether the authors

actually read many of the studies that they cite in their reference list; reading the text does not convey the impression that the authors are aware of data published by many groups over the last several decades. Despite the welcome advance provided by the structural data, this MS is not fit for publication anywhere in its present form.

Authors (revision round 1): We appreciate the feedback and have improved the main text accordingly (see points below), for example by discussing the work by Jerke et al (Jerke et al., 2017) at appropriate points in the text. Please note that the recombinant mutant PR3 we present was developed independently of other studies and that we started the project before the publication of the aforementioned paper. In contrast to Jerke et al. who performed alanine substitutions, we introduced mutations that result in an N-linked glycosylation site, rather than alanines. This significantly improves the solubility and homogeneity of the sample and was instrumental to being able to co-crystallise the protein with CD177, in our hands.

We have rewritten the relevant section in our results to relate our construct to the previously published mutant used by Jerke et al.:

"Using an established protein engineering approach that introduces an N-linked glycosylation site (del Toro et al., 2020; Akkermans et al., 2022), we mutated three residues within the PR3 hydrophobic surface patch: I221N-W222G-G223T (Figure EV 1F). A previously published mutant in which two of these residues were mutated to alanines (I221A-W222A) had also been reported to improve protein solubility (Jerke et al., 2017)."

In the discussion we write:

"The hydrophobic nature of wild type PR3, which is thought to be due to the "hydrophobic patch" on the protein surface (Korkmaz et al., 2008), has presented a challenge for the production and purification of this protein for structural studies. Over the last 30 years, protocols were established for the expression of recombinant PR3 in yeast (Harmsen et al., 1997), Sf9 insect cells (Fujinaga et al., 1996), human mast cell line-1 HMC-1 (Specks et al., 1996; Sun et al., 1998; Van Der Geld et al., 2000) and HEK293 cells (Sun et al., 1998; Van Der Geld et al., 2000; Korkmaz et al., 2008; Jerke et al., 2017). Several of these purification trials have successfully used detergents such as 0.1% Triton X-100 (Robert F. Halenbeck et al., 1995) 1% β -OG (Stummann and Wiik, 1997), 0.1% Tween-80 (Van Der Geld et al., 2002) or 0.02% DDM (Jerke et al., 2017) to mitigate protein aggregation, although it is recognised that these detergents can hamper biological experiments and crystallisation trials. Mutagenesis of the "hydrophobic patch" residues I221 and W222 to alanines resulted in a promising construct with improved solubility (Jerke et al., 2017). Our work was catalysed by introducing an N-linked glycosylation site in position 221 (PR3rec), which made production and purification from HEK293 cells and successful co-crystallisation with CD177 receptor without the use of detergent possible. "

>>>

>>>

It may be that the authors' efforts began before the publication of Jerke, et al., but this is irrelevant: independently achieving the same known result (even by different means) does not absolve one from properly acknowledging previously published examples. There is certainly no justification for presenting such a result as unique. Is the claim that the authors' approach "significantly improves the solubility and homogeneity of the sample" compared to that of Jerke, et al. (which is how the text reads) substantiated by real results - i.e. did the authors also prepare the construct of Jerke et al. and compare the two? If so (which would allow one to make the statement quoted here) why is this data not presented?

In the revised text, the authors write that the mutations made by Jerke et al. were "reported to improve protein solubility". This is only partly true - the mutations were reported to result in PR3 monomers. The authors have (again) intentionally left this clarification out of their revision, despite the criticism levelled in the first review. In the discussion, they refer to the mutant from Jerke et al. as "a promising construct". What do they mean by this statement? The data published by Jerke, et al. demonstrated that the resulting mutant was monomeric. The successful purification of that mutant from HEK_293 cells and experimental data with that mutant were reported in the same paper. The authors have not demonstrated that their construct has properties (besides extra shrubbery) that haven't already been demonstrated by that in Jerke, et al., but make a substantial effort to imply this in their choice of phrasing (which this reviewer finds unacceptable). One is not entitled to make claims - particularly in scientific publications - that have not been substantiated by experimental observations/results.

The authors' unqualified comment that detergents like b-OG and DDM "can hamper...crystallization trials" is simply not correct (as a perusal of the PDB makes clear) - they are, indeed, sometimes essential for crystallization (and where are the citations that substantiate the authors' statement?).

>>>

>>>

Authors:

We are grateful for the reviewer's help with showcasing the work of Jerke et al. in our paper, which we certainly wish to do within the limitations of space and alongside other work in the field. We have amended the sentence in the discussion to attribute that the mutations made by Jerke et al., resulted in a monomeric PR3 and removed the word 'promising': 'Mutagenesis of the "hydrophobic patch" residues I221 and W222 to alanine residues resulted in a monomeric construct with improved solubility (Jerke *et al.*, 2017).'

To address the reviewer's comment about comparing our PR3 mutant to the one made by Jerke et al., we have amended the text to read 'Our work introduces mutations at the same site but we opted for an N-linked glycosylation site in position 221 (PR3^{rec}), which made production and purification from HEK293 cells and successful co-crystallisation with CD177 receptor without the use of detergent possible.' May we please also point out again that our work was not influenced by the results in Jerke et al., which was

published after we had optimised our PR3 construct and purification strategy. We therefore did not try out their construct.

In response to the comments regarding detergent, we have rephrased the relevant sentence, which now reads: 'Several of these purification trials have successfully used detergents such as 0.1% Triton X-100 (Robert F. Halenbeck *et al.*, 1995), 1% β -OG (Stumann and Wiik, 1997), 0.1% Tween-80 (Van Der Geld *et al.*, 2002) or 0.02% DDM (Jerke *et al.*, 2017) to mitigate protein aggregation.'

A discussion of the major flaws in the MS follows here and all of these issues must be adequately addressed before the MS can be submitted to any journal with a reasonable expectation of acceptance.

1) We are long past the days when the generation of a structure alone, however competently done (as the current example appears to be), suffices for a publication (unless combined with some new technical advance or other novelty that was necessary for achieving the structure). It is here emphasized that it is very exciting to now have these structures and their thoughtful analysis will be valuable for understanding the biology of ANCA-related diseases and also for developing treatments. But just such a thoughtful analysis - with confirmatory experimental data - is, unfortunately, missing from the MS. A number of experiments with the complex are reported but there isn't any meaningful analysis in light of previously published predictions - for example, Kantari *et al.* (reference 31 in the MS) made the "hydrophobic patch" responsible for the attachment of PR3 to the outer neutrophil membrane and Korkmaz, *et al.* (reference 81 in the MS) claimed that the "hydrophobic patch" was responsible for the interaction with CD177 - does the structural data (which is what is new) provide any insight into the correctness of these suppositions?

A rather detailed model for the complex based on biochemical data was proposed in 2017 (Jerke, *et al.* - reference number 19 in the MS) - how does the presently reported complex structure compare with those predictions from the biochemical characterization? These examples - and more - should make up a substantial portion of the discussion section of the current manuscript but are entirely absent (although the primary sources are in the reference list). The structures alone are the only new information.

Authors (revision round 1): We thank the reviewer for this feedback and appreciation of the structural results. We have improved and expanded our results section, supplemental figures and discussion section to the published definition of the "hydrophobic patch":

Results: "The centre of the binding surface contains hydrophobic residues (F180, F181, F190 and L228 from PR3, and L117, P118 and W120 from CD177) (Figure 2D-F). These residues are located within the previously described "hydrophobic patch" of PR3 (Korkmaz *et al.*, 2008) (Figure EV3 panel C)."

Figures: see Figure EV3 panel C.

Discussion: "The structure shows that the PR3-CD177 binding interface overlaps, at least in part, with the 'hydrophobic patch' (Korkmaz et al., 2008; Jerke et al., 2017) (Figure EV3 C), validating previous studies suggesting that mutations in this region reduce CD177 binding (Jerke et al., 2017)."

>>>

>>>

The citation of Jerke et al. here is misleading and potentially incorrect. What Jerke et al. suggested was that the "reduce(d) CD177 binding" of their mutant could be due to its monomeric form in comparison to wt which they clearly showed was multimeric - i.e. the increased affinity for CD177 by wtPR3 could be due to avidity effects. The quote from the paper is: "The higher affinity measured for wtPR3 binding is likely due to multivalent interaction of wtPR3 multimers with immobilized CD177." This reviewer additionally notes that the positions mutated by Jerke et al. are NOT in the PR3:CD177 binding interface reported by the authors. The initial statement from this reviewer still stands: in this manuscript, the structures themselves are the only new information. No new meaningful analysis has been incorporated into the revision.

>>>

>>>

Authors: We thank the reviewer for this clarification and have shortened the sentence accordingly.

Please note that, given the focus of this manuscript on how PR3 interacts with its receptor CD177 and the importance of the binding interface as an epitope targeted by PR3-ANCA, it is beyond the scope of this report to analyse how the "hydrophobic patch" as defined by others may function to mediate PR3 attachment to cell membranes etc.

2) The authors give substantial space to their mutagenesis effort to achieve a monodisperse PR3 construct and report finding mutations that render the enzyme monomeric. Quoting from their text: "The study was possible using a site-directed mutagenesis screen to identify stable Pr3 variants that are readily expressed and purified from HEK293 cultures." Perhaps it is an oversight (but then a very serious one) that a successful monomerization effort was already published in 2017 (reference 19 in the MS): the mutated Ile - Trp (i - i+1) pair was shown to render PR3 monomeric and the affinity of that monomer for CD177 by SPR was also reported. The data in the current MS are therefore not new as the authors claim (using another residue numbering scheme does not alter the sequence itself) and this previous work is not acknowledged in their text (although the paper in question does appear in the reference list). This reviewer will stop short of labeling this an ethical problem, but the "oversight" must be corrected no matter where the next version of the MS is to be published.

Authors (revision round 1): We are grateful for the reviewers suggestion to discuss the Jerke et al. paper in more depth. As noted above, our mutant (PR3rec) is not the same as the one published in Jerke et al., and we have now added a clear description of how the two constructs compare in the relevant sections (see our response to previous comments further up). Regarding the numbering scheme, we have followed the scheme that is also used by the UniProt database, where the first Met residue in a sequence is numbered 1.

>>>

>>>

This reviewer did not claim that the PR3 mutant reported by the authors is the same as that reported in Jerke, et al. - it was stated that the authors' rather more involved mutagenesis efforts produced the same superficial result as had been published by others 8 years prior. It was also noted that in the present manuscript the outcome of the mutagenesis was presented as a unique/novel development - i.e. that something new had been achieved. This treatment was either so poorly written that it gave this - demonstrably false - impression or it was an active attempt to mislead the reader. On the first pass, the former possibility was suggested; but since the "clear description" now claimed to have been added is nothing of the sort, the second possibility must be seriously considered.

On this basis alone, in the opinion of this reviewer, the manuscript should be rejected by any journal to which it is submitted.

Arriving at the same result by another method is surely of interest and worthy of reporting – but that can in no way justify an effort to ignore or play down the initially published result - and no data are presented in the current revised manuscript showing that the 'new' scheme for producing soluble and monomeric PR3 is superior to what was published 8 years ago. The manner of this presentation - and the recalcitrance in correcting the initial error - combined with the refusal to clearly correct other misleading statements in the text (see the point below concerning patient ANCA epitopes) falls far short of the standards of honest discourse.

>>>

>>>

Authors: We are sorry to see that the reviewer still feels this way regarding our manuscript. We do not deny that others have made attempts and reported success with purifying a stable, monomeric form of PR3 before us. As described, we designed an independent strategy for producing PR3 and ended up with a different construct compared to others, including an extra glycan, which we used to solve our PR3 complex structure.

3) This reviewer feels that a comment on style here is also necessary: apparently also

gone are the days when formulations in scientific reports minimized use of the first person; while this can sometimes not be gracefully avoided, the practice of introducing basic (if not trivial) results in this MS with overly self-promoting phrases such as "We pioneered..." (page 11), as well as numerous other "We" statements scattered throughout, should be and, considering that much of what is described in the MS was already reported by others and/or not sufficiently acknowledged as such, maximized the discomfort felt when reading the text. These deficiencies, as well as the grammar mistakes and confusing formulations (some, but not all, of which are addressed explicitly below) must be corrected before the next submission, to whichever journal that may be.

Authors (revision round 1): this is an interesting point and we refer to the editor for guidance. Many journals ask for an active style ("we did this...", "we did that...") rather than passive voice ("the sample was subjected...", "it was thought that..."). We will adapt to whichever style is preferred by the editors. We will also do our very best to weed out any typos etc.

>>>

>>>

The authors' writing style is, of course, up to them and the Editor(s). But since "We pioneered..." remains a demonstrably false statement, it should, in my view, be removed from the text.

>>>

>>>

Authors: we have replaced the word 'pioneered' with 'developed' .

4) There are a number of statements throughout the text that are either partly or wholly inaccurate; some examples follow (this list is NOT exhaustive), with comments from this reviewer in bold:

From the Abstract, page 1:

"Granulomatosis with polyangiitis is a life-threatening systemic vasculitis, characterised by antineutrophil cytoplasmic autoantibodies (ANCA) against proteinase 3 (Pr3) expressed on the surface of neutrophils. Most cell surface Pr3 is bound to the receptor CD177..."

These first two sentences of the paper are quite garbled - the first reads as if the ANCA themselves are "expressed" on neutrophils...which of course is not the case; furthermore, PR3 is found on the extracellular surface of all neutrophils, even those not displaying CD177...and even on neutrophils from persons who are genetically negative for CD177, i.e. whose neutrophils completely lack CD177.

Authors (revision round 1): this has been adjusted to read 'Granulomatosis with

polyangiitis is a life-threatening systemic vasculitis, characterised by anti-neutrophil cytoplasmic autoantibodies (ANCA) against proteinase 3 (PR3), a protease expressed on the surface of neutrophils.'

>>>
>>>
Fine.
>>>
>>>

From the Abstract, page 1:

"We describe an extensive, mainly hydrophobic binding interface between Pr3 and CD177...".

This is an unnecessarily and incorrectly qualified statement - the authors report the buried surface area in the complex to be " ~655 Å² " - a value rather at the low end of the buried surface area spectrum that extends, on average, between 1500 and 3000 Å² (see for example 10.1073/pnas.93.1.13); How have the authors decided that the PR3:CD177 interaction is "extensive"?

Authors (revision round 1):

We have removed the word 'extensive' from that sentence.

>>>
>>>
Fine.
>>>
>>>

From the Abstract, page 2:

"Using a panel of Pr3-ANCA positive patient samples, we show that CD177 inhibits Pr3-ANCA binding to Pr3".

It is obvious that this statement on its face cannot be accurate. Certainly the binding of some ANCAs is blocked by the presence of CD177 (as the authors show with their experiments using patient samples) but, as reported later in the MS and has been shown previously in multiple studies (see for example J Immunol. 1994 May 1;152(9):4722-37, J Autoimmun 2010 35, 299-308 (reference 78 in the MS) and also reference 20 in the MS, among others) ANCA epitopes are also found outside of the CD177 binding site and

some ANCA can bind to PR3 that is complexed with CD177.

Authors (revision round 1): We have revised this sentence to say 'Using a panel of PR3-ANCA positive patient samples, we show that a significant proportion of ANCA target the CD177-binding site of PR3.'

>>>

>>>

The statement is still misleading and, when the actual data are considered: false (for all the reasons cited in the initial review). The fact that a handful of tested patient samples interfere with CD177 binding cannot itself be trumpeted as a significant new finding. Extrapolating this handful of results so concretely to include the entirety of ANCA patient samples is even less warranted. And, considering that the subheading of the respective Results section still reads: "Most GPA patient samples harbour autoantibodies that target the CD177-binding site of PR3", it is clear that the authors still do not acknowledge the criticism. The results shown also do not indicate that these ANCA "target the CD177 binding site" - steric effects (which the authors invoke to avoid explaining why PR3 activity is reduced in the presence of CD177 although the active site is not occluded in the complex structure, see below) could easily explain the interference with CD177 binding; the authors write that some ANCA "substantially reduce" CD177 binding - none completely prevented binding. In this regard, the title of the manuscript is also invalid - no ANCA epitope has been determined in this work. It has only been shown with the ELISAs that some ANCA can interfere with CD177 binding to PR3 - this is not the same thing as saying that the binding epitopes of these ANCA have been "revealed" by the structure(s). And no justification is provided (or can be gleaned from the data) for referring to these theoretical epitopes as "major".

>>>

>>>

Authors: Indeed, for research into rare diseases there is no globally accepted minimum sample size. Our sample cohort comprised of 4 female and 5 male donors, ranging from ages 15 to 77, with the median age being 52. We used all the samples available to us and the title reflects our results. We are clear in presenting our results, as we highlight the number of patients in multiple places across the paper. To be as clear as possible, we have now amended the subheading cited above to read: 'Most GPA patient samples in this study harbour autoantibodies that target the CD177-binding site of PR3.' In the results section, we amended the relevant sentence to read 'Using a panel of PR3-ANCA positive patient samples, we show that a significant proportion of ANCA target the CD177-binding site of PR3 in these samples.'

We disagree that our ELISA results do not show that PR3-ANCA target the CD177 binding site. We have shown evidence for this using a site-directed mutagenesis approach that introduces a glycan at the PR3-CD177 binding site, and backed this up with the addition of purified CD177 protein, in our ELISA experiments. Given this, and the

fact that the other two reviewers were satisfied with our title, we are unable to address this comment in a meaningful way.

We would also like to point out that it is unclear to us why the reviewer discusses reduced PR3 activity in the presence of CD177 when we were using an inactive form of PR3 and did not investigate its activity in this study. This appears to be a comment that is directed at the authors of Jerke et al., which is a paper that the reviewer is very focused on, but which is quite distinct from our paper.

The authors also write on page 5: "We discover that most Pr3-ANCA positive samples compete with the interaction between Pr3 and CD177..." and furthermore on page 9: "Most GPA patient samples harbour autoantibodies that target the CD177-binding site of Pr3" because, in their ELISA, 7 of 8 samples appear to bind an epitope coinciding with the CD177 binding site (their conclusion). Now, even accepting the authors' own - not referenced!! - "estimate" of the prevalence of GPA (Introduction, page 2) of "2 - 150 per million individuals", it is clear that these statements cannot be true. Taking the lower limit of this - again not referenced - estimate (and since PR3-ANCA are involved in other diseases and the number of undiagnosed cases of PR3-ANCA diseases is thought to be substantial, the authors' estimate is clearly extremely low), the number of patients must number in the hundreds of thousands/millions, depending on where the authors' "estimate" applies (in England? In Taiwan?? Worldwide?). To have tested samples from <0,001% of all patients (based on the absolute - and incorrect - minimum estimate) and then drawing a firm conclusion about them all cannot be taken seriously.

Authors (revision round 1):

We are of course only focusing on the panel of GPA patient samples presented in this study therefore we have adjusted this sentence to make it clearer: "We discover that most of the PR3-ANCA positive samples tested here target the CD177-binding site of PR3 and display reduced binding to PR3 protein where the CD177-binding site is masked." (page 5).

>>>

>>>

Modifying one sentence as an attempt to satisfy a criticism while leaving the remaining context (which was the target of the criticism) unchanged comes across here as disingenuous. The authors may have adequately qualified this particular statement (only after being called out on the point) but the related statements in the text are not so carefully worded. See for example the comment above concerning the Results subheading. The criticism by this reviewer stands, as does, apparently, its rejection by the authors.

>>>

>>>

Authors: please see our previous comment above.

From the Introduction, page 3:

"...although some PR3 appears to bind directly to cell membrane components".

This is a perplexingly misleading statement- the word "appears" here is simply inaccurate; there is ample published evidence, over many years, demonstrating that PR3 is present on the neutrophil cell surface - even CD177 negative individuals have mPR3 on their neutrophils. This surface interaction has also been repeatedly shown to occur on non-myeloid cells when incubated with purified PR3; again, several of these studies are to be found in the reference list of this MS! It is perhaps the case - and then also perfectly acceptable - that the authors do not agree with all of this published data. But then this should be openly discussed and substantiated and not simply called into question via such confusing statements.

Authors (revision round 1): We have revised the text and it now reads "Most extracellular PR3 is presented on the neutrophil cell surface in complex with its receptor CD177 (also named HNA-2a or NB1) (Bauer et al., 2007b; Von Vietinghoff et al., 2007b; Abdgawad et al., 2010b), although PR3 can also bind directly to cell membrane components (Goldmann, Niles and Arnaout, 1999b; Hajjar et al., 2008; Hu et al., 2009b; Broemstrup and Reuter, 2010b; Kantari et al., 2011). "(page 3).

As mentioned above, this aspect of PR3 biology is not a major focus of the paper and we do not wish to prove or disprove previous work.

>>>

>>>

This reviewer did not claim the topic to be a "major focus of the paper" - but when a statement is made, it should be clear and properly referenced. No excuses.

And the modified sentence is still incorrect - how can "most" mPR3 be in complex with CD177 in individuals who are genetically CD177 negative?

>>>

>>>

Authors: it is thought that 90-99% of people are CD177 positive. As the reviewer will be aware, a number of previous papers have shown that neutrophil surface PR3 correlates with the expression of CD177 and that neutrophils which express little or no CD177 also have strongly reduced surface PR3. To be clear we have now amended the text to read : 'In CD177-positive individuals, the majority of extracellular PR3 is presented on the neutrophil cell surface in complex with its receptor CD177 (also named HNA-2a or NB1) (Bauer *et al.*, 2007; Von Vietinghoff *et al.*, 2007; Abdgawad *et al.*, 2010), although PR3 can

also bind directly to cell membrane components (Goldmann, Niles and Arnaout, 1999; Hajjar *et al.*, 2008; Hu *et al.*, 2009; Broemstrup and Reuter, 2010; Kantari *et al.*, 2011).'

From the Introduction, page 4:

"It is thought that this response is at least in part due to Pr3-ANCA-mediated recruitment and activation of Fcg receptors (37,44,45)".

This statement refers to the involvement of Fc receptor recruitment for the PR3-ANCA stimulated respiratory burst response of neutrophils...this was substantiated also in reference 20 in the MS but has not been included in the citation.

Authors (revision round 1): we would have been happy to cite this paper, but in the process of streamlining the paper to make it more focused and reader-friendly (in response to other comments), we have removed this section from the introduction. As the data presented in our paper is not focused on the Fc-gamma receptor, nor involves experiments with neutrophils, we think that this paragraph is not strictly necessary.

>>>

>>>

Fine.

>>>

>>>

From the Introduction, page 5:

"The study was possible using a site-directed mutagenesis screen to identify stable Pr3 variants that are readily expressed and purified from HEK293 cultures".

As mentioned above, the mutation of the residues producing the monomeric and "stable" PR3 variant - which had been initiated by the observation that wtPR3 forms an array of higher order aggregates in solution at physiological salt concentrations - was previously fully described (reference 19 in the MS) along with measurement of its binding affinity to CD177 and an evaluation of the relevance of the mutations with regard to the binding interface. The current MS claims a modified approach (including adding a glycosylation) at the identical positions to achieve the same effect that the authors of reference 19 managed with alanine substitutions. It is of note here that, while a PR3 monomer peak is visible in the size exclusion data shown in Fig. S1 f, the mutagenesis apparently did not result in an exclusively monomeric species as misleadingly claimed in the text ("...is expressed at high levels yielding stable and monodisperse protein samples") as it did for the authors of reference 19. (If the larger species in the chromatogram of Fig. S1 f are NOT higher order PR3 aggregates, the authors should convincingly show this, for example via Western blot). It is curious indeed

that all of these previously and quite explicitly reported results are presented in the current MS as original with no reference to the older study that is actually cited several times for other reasons in the text. This is a very serious error and raises a number of questions as to how it could have occurred.

Authors (revision round 1): Thank you for this helpful comment. We now updated the relevant supplemental figure legend (Fig. EV1 panel G) to include a distinct description of the SDS-PAGE gel (Coomassie) and anti-His western blot (a-his). The originally submitted figure showed one of the first purifications attempted in the lab where the higher band at 60 kDa in the western blot -but not on the SDS-PAGE gel- accounts for a histidine-rich impurity that is sometimes co-purified in small amounts, depending on the quality of the immobilised metal ion affinity chromatography (IMAC) experiment. We see this sometimes, and for different proteins, produced in our system. This impurity, if present in tiny amounts, typically does not impact our crystallisation trials, but we do make sure to remove it by thorough wash for any functional studies, such as ELISA. To reflect this, we now show the SEC chromatogram, SDS-PAGE and western blot analysis from a later, cleaner purification of PR3rec, see Figure EV1 panel G.

>>>

>>>

It is noted that the authors pointedly refuse to comment on the lack of attribution mentioned in the initial criticism and have not corrected the problem.

>>>

>>>

Authors: We have actually attributed the work by Jerke et al., in the first round of revisions. Please see our comments above.

From the Introduction, page 3:

" CD177 (also named HNA-2a or NB1) is a ~50 kDa glycosylphosphatidylinositol (GPI)-anchored surface glycoprotein".

First of all, although one assumes that the CD177 homolog in question is the human one, this reviewer could find no indication of this - nor the association of the word "human" directly linked to CD177 - anywhere in the main body of the text. Fortunately the authors did include the Uniprot accession number in the Methods section; but where does their claim of "~50 kDa" come from? The full length protein sequence - that includes an N-terminal signal sequence not present in the final active form of the molecule - is calculated to be 46,4 kDa; the molecule that the authors crystallized and used for their other experiments is calculated to be 41,3 kDa...how are either of these values "~50 kDa"? While these may seem like minor points, they are rather indicative of the lack of rigor that characterizes much of the MS - when unambiguous, established facts cannot be faithfully repeated, what is one to think of the "new" data presented and

the interpretations thereof? A further issue is the repeated use of the tilde (~) to report measured values - where are the decimal places and standard error values?

Authors (revision round 1): Indeed the reviewer is right that the predicted protein mass of human CD177 is less than 50kDa, however, the protein includes 3 predicted N-linked glycosylation sites, each accounting for an unknown size glycan . A previous mass spectrometry study (Kissel et al., 2001) found native glycosylated CD177 to be 50.556 kDa. Our use of "~50 kDa ... glycoprotein" reflects the findings in this previous study. We have now added the reference for clarity. (page 3)

>>>

>>>

The refusal of the authors to answer direct criticisms is perplexing; what was the mass of the actual molecule that was purified and crystallized?

>>>

>>>

Authors: the reviewer does not explain why our reply is not satisfactory. We cite previous analysis in the field, which matches with what we see in our SDS-page analysis (Figure EV2B). If the reviewer meant to request a different type of mass analysis of our purified CD177 construct, then they do not explain why this is relevant. This is not usually requested without a reason.

From the Results, page 6:

"Surface plasmon resonance (SPR) and SEC-MALS experiments confirmed that Pr3rec binds CD177ecto with high affinity ($K_d \sim 10$ nM)".

Why is the affinity reported in such an imprecise way? Where are the decimal places and error calculations?

Authors (revision round 1): We have now included the standard deviation values in the main text as well as in Figure EV 2F.

>>>

>>>

Fine.

>>>

>>>

From the Results, page 7:

"The data show that CD177 binding does not occlude the Pr3 catalytic site, likely leaving it available for substrates".

While this statement - strictly speaking - may not be false, it is devoid of meaning. The issue is not whether the catalytic site is "available for substrates", it is whether enzymatic activity is affected upon PR3:CD177 complex formation (which could happen without a direct occlusion of the active site). Studies cited in the reference list of this MS have reported inhibition of PR3 activity in the presence of CD177 - in vitro and in vivo. Conformational changes occurring in multiple serine proteases have also been demonstrated. Incomprehensibly, none of this is included in discussion of the structures or mentioned in the text at all.

Authors (revision round 1): We now provide a discussion of this point: "

Previous work suggested a negative effect of CD177 on PR3 activity: the addition of purified CD177 was shown to inhibit nPR3 activity in vitro, in neutrophil degranulation experiments, and at the surface of neutrophils or CD177-expressing HEK cells (Jerke et al., 2017). Here we show that CD177-binding does not directly occlude the PR3 active site, nor does it perturb the PR3 structure compared to unliganded PR3 (Figure EV3 B). Therefore the mechanism of inhibition is likely more subtle, for example CD177 could sterically affect substrate binding away from the active site ." (page 11)

>>>

>>>

It is not clear to this reviewer what the statement "...CD177 could sterically affect substrate binding away from the active site ." intends to convey - do PR3 substrates bind "away from the active site"? What does this statement mean? (One could also ask: why in some manner of steric effect invoked here but when the exact same argument could be used for the inhibition of CD177 binding by ANCAs, the authors rather call that 'targeting of the CD177 epitope'?). Jerke, et al. are not the only investigators who have shown differences in PR3 activity depending on its disposition (even on neutrophils)...others have invoked different conformations of PR3 to account for differences in activity between assayed pools of PR3 on neutrophils. Further underlining the lack of a thoughtful analysis of the data in this manuscript, the authors - although now clearly stating that their changes to PR3 could conceivably affect conformational changes necessary for the enzyme's function - claim simply that "Our mutants were designed such as to minimise the predicted disturbance of the PR3 fold." - but this says nothing about possible disturbance of potential conformational changes. These are mentioned in the context of ANCA binding but not with reference to enzymatic activity. In their mentions of cited reports of decreased wtPR3 activity in the presence of CD177, the authors fail to emphasize (or realize?) that their construct is decidedly not wildtype - in fact, the complex structure omits half of CD177 and the authors fail to discuss how the multimerization of wtPR3 plays into complex formation in vivo. This deficiency is carried over to their fanciful model (shown in Figure 5 and still almost certainly incorrect) that fuses the full length CD177 structure with that of their artificial PR3 construct complex

structure. What effect does PR3 multimerization have *in vivo*? Where is this depicted in this "model"? And why are the two structures fused together in the "model" - where is the evidence that the two lobes of CD177 remain identically oriented with or without wtPR3? The authors seem also to have failed to consider that the influence of their PR3 modifications on conformational flexibility may underlie the incongruities they report when using different commercial screens for PR3 detection. All of these points are low hanging fruit but none of them - either in the first or in this second incarnation of the manuscript - are elaborated.

>>>

>>>

Authors:

Indeed the paper opens many questions for future work, for example, regarding the mechanism of CD177 affecting PR3 enzymatic activity, which is not explained by the structural results. We have shortened the relevant sentence to be less speculative: 'Therefore, the mechanism of inhibition is not clear.'

As mentioned above, this study only used inactive PR3 mutants, therefore we do not comment on enzymatic activity. We also are aware that we are not using a wildtype PR3 and therefore do not comment on the multimerization of wtPR3 *in vivo*. We have shown in previous work that PR3 exists as part of a larger multi-receptor complex (Chu et al. 2022), and there is no clear data for the exact higher order architecture(s) PR3 adopts *in vivo*. Given the lack of clarity which (if any) PR3 multimers are physiologically relevant, and the fact the paper's focus is elsewhere, we believe that an extensive discussion of this point would be distracting.

That said, we are grateful and have taken the feedback regarding Figure 5. We have now changed the figure to show a summary of our ELISA data.

Note that we do consider the effect of our PR3 modifications on ANCA binding as stated here in the discussion: 'It is important to note that deviations from the wild type PR3 sequence (be it through swaps, mutations and/or introductions of glycans) could result in local or allosteric conformational changes of PR3, which in turn could influence ANCA binding... Whilst the PR3 fold is likely preserved in PR3^{rec}, the mutations we introduced could potentially inhibit the binding of PR3-ANCA subpopulations targeting this surface.'

The Discussion section is comprised mostly of meandering filler instead of a thoughtful evaluation of the established biochemical data from many other groups in light of the only novelty in the MS, the structures of CD177 and the PR3:CD177 complex. The authors' own statement on page 11: "...the molecular architecture of Pr3 surface presentation also impacts Pr3-ANCA interactions and therefore warrants investigation" is not at all effectively followed up on in the remainder of the text. Based on the

observation of wtPR3 aggregation alone (again, reported in the MS as a novelty although already published years ago), the "model" of the PR3:CD177 complex/ANCA interaction presented in Fig. 5 is almost certainly - completely - wrong. It is also unclear to which "hypothesis" is referred on page 12 - the issue of where a handful of patient ANCAs may bind PR3 has already been mentioned above. The 'theorizing' comprising the bulk of pages 12 and 13 could be deleted in its entirety with no deleterious effect at all on the MS. The discussion of the ELISA data with wt and mutated PR3 - which makes no mention of the "...faster detection..." observed with the mutant that is slipped into the legend of Figure 3D - provides no new insights and somehow also does not include the (not) novel observation of wtPR3 aggregation. Even the suggestions of better diagnostics and potential therapies based on knowledge of the complex and PR3 mutations is old news; there is nothing novel in any of it, all of these points have been suggested/discussed in work by others - as a reading of the cited literature and even a cursory perusal of the patent databases make clear.

Authors (revision round 1): We have reworked our discussion, for example, including previous biochemical analysis done to reduce PR3 aggregation. We also replaced the "structure-based" views of CD177 and PR3 in Figure 5 with more abstract representations to avoid any confusion with regards to the detailed molecular mechanisms of ANCA-PR3 interaction. In response to the comments, we now provide a better context for the section on the future development of diagnostics and therapies. We maintain that our work is novel not just with regards to the structural and biophysical work (as recognised by all three reviewers), but also with regards to the ELISA study. In contrast to previous studies, we are able to compare the binding of patient samples to different PR3 variants (or complex), using an anti-His. We believe that the newly revised discussion now provides a balanced view of previous and novel results, within the limitations of a concise manuscript.

>>>

>>>

The authors write here: "We also replaced the "structure-based" views of CD177 and PR3 in Figure 5 with more abstract representations to avoid any confusion with regards to the detailed molecular mechanisms of ANCA-PR3 interaction."

To which "detailed molecular mechanisms" do they refer? Where - at all - in this manuscript is there any treatment of "detailed molecular mechanisms"? There is no meaningful consideration of the properties of wtPR3, an unsubstantiated assumption that the CD177 alone and the truncated CD177:PR3rec complex structures are interchangeable with each other and with their in vivo counterparts and no results that clearly map onto the in vivo relevance of the native complex (except some differential antibody binding with no insights into mechanisms).

The authors "maintain that our work is novel" but there is no demonstration that they have developed or invented anything unique. The identification of epitopes based on the

structure is claimed but there is no description of these epitopes. ELISA studies are not novel and the ones reported in the manuscript are nothing out of the ordinary. Solid biochemistry is just that and doesn't need to be advertised as being "pioneering" or "novel". There is entirely too much hype in the text for so little thoughtful analysis.

Authors: Please see our comments above. We also affirm that our work is novel, for example, we developed our independent purification strategies for PR3 and CD177, solved the structure of the complex, performed structure-based protein engineering, and developed a new ELISA test using a standard that binds to our protein tags so that we could compare responses between different PR3 versions and conditions (instead of using patient plasma/ANCAs as a standard). Finally, albeit with a small sample size, our ELISA data showed that CD177 can inhibit PR3-ANCA binding. That said, we acknowledge the limitations of this study and appreciate this reviewer's thorough analysis of our work, as well as the excellent work done in the field previously.

We have done our best to amend the style and language used in the paper to better reflect our intentions.

Additional comments:

- On page 4 of the manuscript, the authors write: "The precise mechanism on PR3-CD177 interaction remains to be shown." -grammar problems aside, the statement is accurate; one must therefore ask: what then was the topic of this manuscript supposed to be?

Authors: please see our comments above. We have also amended this sentence to be more clear: 'The precise mechanism of PR3-CD177 interactions *in vivo* still remains to be shown.'

- Further down on page 4, the authors state that neutrophils with more mPR3 are more strongly activated by ANCAs than neutrophils with low mPR3 levels - but their citation is incorrect; the correct reference for that data is:

Adrian Schreiber, Friedrich C Luft, Ralph Kettritz. *Kidney Int.* 2004 Jun;65(6):2172-83. doi: 10.1111/j.1523-1755.2004.00640.x.

Authors: we kindly thank the reviewer for highlighting this oversight. We have now added the correct reference: 'PR3-ANCA activate CD177^{pos}/PR3^{high} neutrophils more strongly compared to CD177^{neg}/PR3^{low} neutrophil populations (Schreiber, Luft and Kettritz, 2004).'

- The authors mark the binding site of antibody CLB12.8 in figure EV1E and state in the figure legend that this antibody is "mentioned in this report"...but this reviewer is unable to find any mention of this antibody anywhere else in the text (perhaps the authors have spent too much time looking at Fig. 3g in Jerke, et al.).

>>>

>>>

Authors: we thank the reviewer for highlighting this oversight, since we don't mention WGM2 or CLB12.8 in the main text we have removed them from Figure EV1E

Editorial requests

- Please add up to five keywords to the manuscript and order the manuscript sections like this, using only these names:

Title page - Abstract (max. 175 words) - Keywords (up to five) - Introduction - Results - Discussion - Methods - Data availability section - Acknowledgements (please include here all the funding information) - Disclosure and Competing Interests Statement - References - Figure legends - Expanded View Figure legends

Keywords: PR3, CD177, ANCA, granulomatosis with polyangiitis (GPA), ANCA-associated vasculitis

Manuscript sections have been ordered using only the names provided above. Abstract is 166 words.

- We now use CRediT to specify the contributions of each author in the journal submission system. CRediT replaces the author contribution section. Please use the free text box to provide more detailed descriptions and do NOT provide your final manuscript text file with an author contributions section. See also our guide to authors (section 'Author contributions'):

<https://link.springer.com/journal/44319/submission-guidelines#cms-Revised-submissions>

We have used the submission system to add author contributions and have not included an author contributions section in the final manuscript file as requested.

- Please add the email addresses of the 4 co-corresponding authors to the title page.

Done

- The email for Kamel El Omari (kamel.omari@some.ox.ac.uk) bounced. The author should thus either be removed from the author list in the system and then added back using the new/current email address, or please send us the new/current email address and we will update the author's account.

Apologies, there is a typo in his email, it's kamel.elomari@some.ox.ac.uk.

- Please upload individual production quality figure files as .eps, .tif, .jpg (one file per figure), of main figures and EV figures. Please upload these as separate, individual files upon re-submission.

Done

<https://link.springer.com/journal/44319/submission-guidelines#cms-Revised-submissions>

<https://media.springernature.com/original/springer-cms/rest/v1/content/27825798/data/v1>

- Please use our reference format:

<https://link.springer.com/journal/44319/submission-guidelines#cms-Reference-guidelines>

Please use 'et al' needs to be used after the 10th author name; DOIs should only be used for preprints and datasets that have not been published yet.

We have now amended our references

- Per journal policy, we do not allow 'data not shown', which is stated in the manuscript (page 9). All data referred to in the paper should be displayed in the main or Expanded View figures, or an Appendix. Thus, please add these data (or change the text accordingly if these data are not central to the study). See:

<https://link.springer.com/partners/embo-press/editorial-policies#Data%20deposition>

We have changed the text as this data is not central to the study.

- Please check again that the number "n" for how many independent experiments were performed, their nature (biological versus technical replicates), the bars and error bars (e.g. SEM, SD) and the test used to calculate p-values is indicated in the respective figure legends (main, EV and Appendix figures). Please also check that all the p-values are explained in the legend, and that these fit to those shown in the figure. Please provide statistical testing where applicable. Please avoid the phrase 'independent experiment' but clearly state if these were biological or technical replicates. Please also indicate (e.g. with n.s.) if testing was performed, but the differences are not significant. In case n=2, please show the data as separate datapoints without error bars and statistics. See also:

<https://link.springer.com/journal/44319/submission-guidelines#cms-Figure-and-data-presentation>

If n<5, please show single datapoints for diagrams. Moreover:

- Please note that the exact p values are not provided in the legends of figures 3C, D; 4B, C; EV4 A. We provide exact p values in Dataset EV1 and is mentioned in the

figure legend. We provide a legend for the asterisks used for p-values at the end of the figure legend that covers all panels.

- Please note that information related to n is missing in the legends of figures EV5 B-G thank you for spotting, info now has been added to figures EV5 B-G and Figure 4B+C

- Please note that the error bars are not defined in the legends of figures EV5 B-G – legend has been amended

- Please note that the asterisk is not defined in the legend of figures EV1 I, EV2 B. This needs to be rectified. – legends have now been amended

- Please upload the datasets as Dataset EV1, Dataset EV2 and Dataset EV3 and make sure they are all called out like this. Please add a legend for each on the first TAB of the respective excel file.

Legends have been added for each dataset and have been uploaded as requested

- Please label the tables in the manuscript as Table 1 and Table 2. Or should these be EV tables? If yes, please remove them from the manuscript text file and upload the files separately (as Table EV1 and Table EV2). Please add legends for both tables on the first TAB of the respective excel sheets and also update any callouts.

Tables S1+2 have been renamed to Table 1 and Table 2 and we have updated the callouts in the text

- Please make sure that all the funding information is also entered into the online submission system and that it is complete and similar to the one in the acknowledgement section of the manuscript text file. Presently, the grants Wellcome Trust (202827/Z/16/Z and 226647/Z/22/Z) and the EMBO Young Investigator Programme; Chang Gung Memorial Hospital-Linkou, Taiwan (CMRPD1M0033) and the National Science and Technology Council (NSTC), Taiwan (NSTC-113-2918-I-182-001 and NSTC-113-2320- B-182-009) are missing in the submission system. Please check.

we have now added the missing funding information on the submission system

- Thanks for providing the source data. Please upload the source data as one folder per main figure, grouping together all the files (separate files for each panel) for this figure (and ZIPed together), and one folder for all the EV Figures, grouping together all the files for each EV Figure in separate folders (and ZIPed together). Please also make sure that all links to externally deposited source data are updated and present in the final Data Availability Section.

Done

- Please also add a direct link for the deposited structural data to the Data Availability section.

Done

In addition, I would need from you uploaded separately:

- a short, two-sentence summary of the manuscript (not more than 35 words).

Crystal structures of CD177 ectodomain and a CD177-PR3 complex reveal binding between the first two Ly6/uPAR domains of CD177 and a hydrophobic surface on PR3. Binding competes with a subset of PR3-ANCA vasculitis patient autoantibodies.

- two to four short (!) bullet points highlighting the key findings of your study (two lines each).

- Recombinant human proteinase 3 (PR3) containing an artificial N-linked glycosylation site on I221N is effective in ELISA-based PR3-ANCA detection assays
- The CD177 ectodomain folds into two subdomains (LU1+LU2 and LU3+LU4), of which the N-terminal subdomain (LU1+LU2) binds to PR3.
- The PR3-CD177 binding interface overlaps with the previously identified hydrophobic patch of PR3 and can be mutated to abolish binding
- ELISA assays suggest that PR3-ANCA binding to PR3 is reduced when the CD177-binding site is masked or modified.

- a schematic summary figure as separate file that provides a sketch of the major findings (not a data image) in jpeg or tiff format (with the exact width of 550 pixels and a height of not more than 400 pixels) that can be used as a visual synopsis on our website.

Dr. Elena Seiradake
Oxford University
Biochemistry
South Parks Road
Oxford, OXON OX1 3QU
United Kingdom

Dear Dr. Seiradake,

Thank you for the submission of your final revised manuscript to our editorial offices. I now went through it and your final p-b-p-responses and consider the remaining concerns and suggestions of the two referees and the editorial requests as adequately addressed.

I thus am very pleased to accept your manuscript for publication in the next available issue of EMBO reports. Thank you for your contribution to our journal.

Please make sure that the two datasets deposited at the RCSB Protein Data Bank are public latest upon online publication of the manuscript.

You may qualify for financial assistance for your publication charges - either via a Springer Nature fully open access agreement or an EMBO initiative. Check your eligibility: <https://link.springer.com/journal/44319/how-to-publish-with-us>

Yours sincerely,

>>> Please note that it is EMBO Reports policy for the transcript of the editorial process (containing referee reports and your response letter) to be published as an online supplement to each paper. If you do NOT want this, you will need to inform the Editorial Office via email immediately. More information is available here: <https://link.springer.com/partners/embo-press/editorial-policies#Peer%20review>